# Revisiting Random Walks for Learning on Graphs

**Jinwoo Kim**[1]    **Olga Zaghen**[2][*][†]    **Ayhan Suleymanzade**[1][*]
**Youngmin Ryou**[1]    **Seunghoon Hong**[1]
[1]KAIST    [2]University of Amsterdam

## Abstract

We revisit a simple model class for machine learning on graphs, where a random walk on a graph produces a machine-readable record, and this record is processed by a deep neural network to directly make vertex-level or graph-level predictions. We call these stochastic machines **random walk neural networks (RWNNs)**, and through principled analysis, show that we can design them to be isomorphism invariant while capable of universal approximation of graph functions in probability. A useful finding is that almost any kind of record of random walks guarantees probabilistic invariance as long as the vertices are anonymized. This enables us, for example, to record random walks in plain text and adopt a language model to read these text records to solve graph tasks. We further establish a parallelism to message passing neural networks using tools from Markov chain theory, and show that over-smoothing in message passing is alleviated by construction in RWNNs, while over-squashing manifests as probabilistic under-reaching. We empirically demonstrate RWNNs on a range of problems, verifying our theoretical analysis and demonstrating the use of language models for separating strongly regular graphs where 3-WL test fails, and transductive classification on arXiv citation network[1].

## 1 Introduction

Message passing neural networks (MPNNs) are a popular class of neural networks on graphs where each vertex keeps a feature vector and updates it by propagating messages over neighbors (Battaglia et al., 2018). MPNNs have achieved success, with one reason being their respect for the natural symmetries of graph problems (i.e., invariance to graph isomorphism) (Chen et al., 2019). On the other hand, MPNNs in their basic form can be viewed as implementing color updates of the Weisfeiler-Lehman (1-WL) graph isomorphism test, and thus their expressive power is not stronger (Xu et al., 2019). Also, their inner workings are often tied to the topology of the input graph, which is related to over-smoothing, over-squashing, and under-reaching of features under mixing (Giraldo et al., 2023).

In this work, we revisit an alternative direction for learning on graphs, where a random walk on a graph produces a machine-readable record, and this record is processed by a deep neural network that directly makes graph-level or vertex-level predictions. We refer to these stochastic machines as **random walk neural networks (RWNNs)**. Pioneering works were done in this direction, often motivated by the compatibility of random walks with sequence learning methods. Examples include DeepWalk (Perozzi et al., 2014) and node2vec (Grover & Leskovec, 2016), which use skip-gram on walks, AWE (Ivanov & Burnaev, 2018), which uses embeddings of anonymized walks (Micali & Zhu, 2016), and CRaWl (Tönshoff et al., 2023), which uses 1D CNNs on sliding window descriptions of walks. These methods were proven powerful in various contexts, such as graph isomorphism learning that requires expressive powers surpassing 1-WL test (Tönshoff et al., 2023; Martinkus et al., 2023).

Despite the potential, unlike MPNNs, we have not yet reached a good principled understanding of RWNNs. First, from the viewpoint of geometric deep learning, it is unclear how we can systematically incorporate the symmetries of graph problems into RWNNs, as their neural networks are not necessarily isomorphism invariant. Second, the upper bound of their expressive power, along with the requirements on each component to reach it, is not clearly known. Third, whether and how the over-smoothing and over-squashing problems may occur in RWNNs is not well understood. Addressing these questions would improve our understanding of the existing methods and potentially allow us to design enhanced RWNNs by short-circuiting the search in their large design space.

---

[*]These authors contributed equally.
[†]Work done during an internship at School of Computing, KAIST.
[1]Code is available at `https://github.com/jw9730/random-walk`.

Table 1: An overview of prior methods in the context of RWNNs and new components originating from our analysis. NB is non-backtracking; MDLR is minimum degree local rule (Section 2.1).

| Method | Random walk | Recording function $q$ | Reader NN $f_\theta$ |
|---|---|---|---|
| DeepWalk | Uniform | Identity | Skip-gram |
| node2vec | Second-order $pq$-walk | Identity | Skip-gram |
| AWE | Uniform | Anonymization | Embedding table |
| CRaWl | Uniform + NB | Sliding window | 1D CNN |
| RW-AgentNet | Uniform | Neighborhoods | RNN |
| WalkLM | Uniform | Text attributes | Language model |
| Ours (§2, §3) | MDLR (+ NB) (§2.1) (+ restarts) (§2.2) | Anonymization (§2.1) (+ named neighbors) (§2.1) | Any universal (§3.1) |

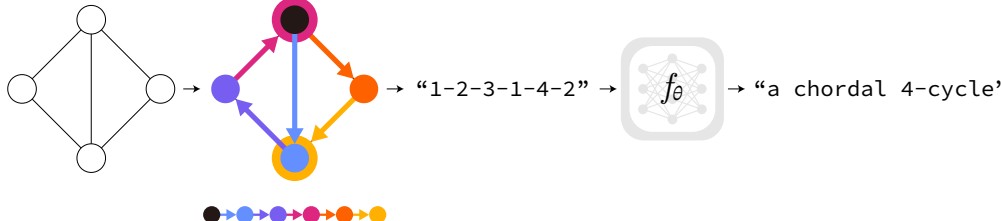

Figure 1: An RWNN that reads text record using a language model.

**Contributions** We aim to establish a principled understanding of RWNNs with a focus on the previously stated questions. Our starting point is a new formalization of RWNNs which decouples random walks, their records, and neural networks that process them. This abstracts a wide range of design choices including those of prior methods, as in Table 1. Our key idea for the analysis is then to understand RWNNs through a combination of two recent perspectives in geometric deep learning: probabilistic notions of invariance and expressive power (Bloem-Reddy & Teh, 2020; Abboud et al., 2021), and invariant projection of non-invariant neural networks (Puny et al., 2022; Dym et al., 2024).

This idea allows us to impose probabilistic invariance on RWNNs even if their neural networks lack symmetry, by instead requiring (probabilistic) invariance conditions on random walks and their records. This provides a justification for the anonymized recording of walks (Micali & Zhu, 2016) and our novel extension of it using named neighbors. As long as probabilistic invariance holds, this also enables recording walks in plain text and adopting a language model to process them (Figure 1).

We then upper-bound the expressive power of RWNNs as universal approximation in probability which surpasses the WL hierarchy of MPNNs, under the condition that random walk records the graph of interest with a high probability. This establishes a useful link to cover times of Markov chains, providing guidance on designing random walks to minimize or bound the cover times. From this we motivate a novel adaptation of minimum degree local rule (MDLR) walks (David & Feige, 2018) for RWNNs, and introduce restarts when working on a large and possibly infinite graph. We also provide a justification for the widespread use of non-backtracking (Tönshoff et al., 2023).

Continuing the link to Markov chain theory, we further analyze RWNNs and establish a parallelism to a linearized model of MPNNs. From this, we show that over-smoothing in MPNNs is inherently avoided in RWNNs, while over-squashing manifests as probabilistic under-reaching. This eliminates the typical trade-off between over-smoothing and over-squashing in MPNNs (Giraldo et al., 2023; Nguyen et al., 2023), and allows an RWNN to focus on overcoming under-reaching by scaling the walk length or using rapidly-mixing random walks such as non-backtracking walks.

We empirically demonstrate RWNNs on several graph problems. On synthetic setups, optionally with a small 1-layer transformer, we verify our claims on cover times, over-smoothing, and over-squashing. Then, we demonstrate adapting a language model (DeBERTa (He et al., 2021)) to solve the challenging task of isomorphism learning of strongly regular graphs with perfect accuracy whereas the 3-WL test fails, improving over the previously known best result of RWNNs. We further suggest that our approach can turn transductive classification problem on a large graph into an in-context learning problem by recording labeled vertices during random walks. To demonstrate this, we apply Llama 3 (Dubey et al., 2024) model on transductive classification on arXiv citation network with 170k vertices, and show that it can outperform a range of MPNNs as well as zero- and few-shot baselines. Our experiments show the utility of our approach in the analysis and development of RWNNs.

## 2 RANDOM WALK NEURAL NETWORKS

We start our discussion by formalizing random walk neural networks (RWNNs). We define an RWNN as a randomized function $X_\theta(\cdot)$ that takes a graph $G$ as input and outputs a random variable $X_\theta(G)$ on the output space $\mathbb{R}^d$.[1] In the case of vertex-level tasks, it is additionally queried with an input vertex $v$ which gives a random variable $X_\theta(G, v)$. An RWNN consists of the following components:

1. **Random walk algorithm** that produces $l$ steps of vertex transitions $v_0 \to \cdots \to v_l$ on the input graph $G$. If an input vertex $v$ is given, we fix the starting vertex by $v_0 = v$.

2. **Recording function** $q : (v_0 \to \cdots \to v_l, G) \mapsto \mathbf{z}$ that produces a machine-readable record $\mathbf{z}$ of the random walk. It may access the graph $G$ to record auxiliary information such as attributes.

3. **Reader neural network** $f_\theta : \mathbf{z} \mapsto \hat{\mathbf{y}}$ that processes record $\mathbf{z}$ and outputs a prediction $\hat{\mathbf{y}}$ in $\mathbb{R}^d$. It is the only trainable component and is not restricted to specific architectures.

For graph-level prediction, we sample $\hat{\mathbf{y}} \sim X_\theta(G)$ by running a random walk on $G$ and processing its record using the reader NN $f_\theta$. For vertex-level prediction $\hat{\mathbf{y}} \sim X_\theta(G, v)$, we query the input vertex $v$ by simply starting the walk from it. In practice, we ensemble several sampled predictions e.g. by averaging, which can be understood as Monte Carlo estimation of the mean predictor $(\cdot) \mapsto \mathbb{E}[X_\theta(\cdot)]$.

We now analyze each component for graph-level tasks on finite graphs, and then vertex-level tasks on possibly infinite graphs. The latter simulates the problem of scaling to large graphs such as in transductive classification. As our definition captures many known designs (Table 1), we discuss them as well. We leave the pseudocode in Appendix A.2 and notations and proofs in Appendix A.9.

### 2.1 GRAPH-LEVEL TASKS

Let $\mathbb{G}$ be the class of undirected, connected, and simple graphs[2]. Let $n \geq 1$ and $\mathbb{G}_n$ be the collection of graphs in $\mathbb{G}$ with at most $n$ vertices. Our goal is to model a graph-level function $\phi : \mathbb{G}_n \to \mathbb{R}^d$ using an RWNN $X_\theta(\cdot)$. Since $\phi$ is a graph function, it is reasonable to assume isomorphism invariance:

$$\phi(G) = \phi(H), \quad \forall G \simeq H, \tag{1}$$

Incorporating the invariance structure to our model class $X_\theta(\cdot)$ would offer generalization benefit. As $X_\theta(\cdot)$ is randomized, we accept the probabilistic notion of invariance (Bloem-Reddy & Teh, 2020):

$$X_\theta(G) \overset{d}{=} X_\theta(H), \quad \forall G \simeq H. \tag{2}$$

An intuitive justification is that, if $X_\theta(\cdot)$ is invariant in probability, its mean predictor $(\cdot) \mapsto \mathbb{E}[X_\theta(\cdot)]$ would be an invariant function (we leave further discussion in Section 4). We now claim that we can achieve probabilistic invariance of $X_\theta(\cdot)$ by properly choosing the random walk algorithm and recording function while not imposing any constraint on the reader NN $f_\theta$.

**Proposition 2.1.** $X_\theta(\cdot)$ *is invariant in probability, if its random walk algorithm is invariant in probability and its recording function is invariant.*

In other words, even if $f_\theta$ lacks symmetry, invariant random walk and recording function provably converts it into an invariant random variable $X_\theta(\cdot)$. This is an extension of invariant projection operators on functions (Puny et al., 2022; Dym et al., 2024) to a more general and probabilistic setup.

**Random walk algorithm** A random walk algorithm is invariant in probability if it satisfies:

$$\pi(v_0) \to \cdots \to \pi(v_l) \overset{d}{=} u_0 \to \cdots \to u_l, \quad \forall G \overset{\pi}{\simeq} H, \tag{3}$$

where $v_{[\cdot]}$ is a random walk on $G$, $u_{[\cdot]}$ is a random walk on $H$, and $\pi : V(G) \to V(H)$ specifies the isomorphism from $G$ to $H$. It turns out that many random walk algorithms in literature are already invariant. To see this, let us write the probability of walking from a vertex $u$ to its neighbor $x \in N(u)$:

$$\mathrm{Prob}[v_t = x | v_{t-1} = u] := \frac{c_G(u, x)}{\sum_{y \in N(u)} c_G(u, y)}, \tag{4}$$

where the function $c_G : E(G) \to \mathbb{R}_+$ assigns positive weights (conductances) to edges. If we set $c_G(\cdot) = 1$, we recover uniform random walk used in DeepWalk (Perozzi et al., 2014). We show:

---

[1]While any output type such as text is possible (Figure 1), we explain with vector output for simplicity.
[2]We assume this for simplicity but extending to directed or attributed graphs is possible.

**Proposition 2.2.** *The random walk in Equation 4 is invariant in probability if its conductance $c_{[\cdot]}(\cdot)$ is invariant:*

$$c_G(u, v) = c_H(\pi(u), \pi(v)), \quad \forall G \overset{\pi}{\simeq} H. \tag{5}$$

*It includes constant conductance, and any choice that only uses degrees of endpoints $\deg(u)$, $\deg(v)$.*

We favor using degrees since it is cheap while potentially improving the behaviors of walks. Among many possible choices, we find that the following conductance function called the minimum degree local rule (MDLR) (Abdullah et al., 2015; David & Feige, 2018) is particularly useful:

$$c_G(u, v) \coloneqq \frac{1}{\min[\deg(u), \deg(v)]}. \tag{6}$$

MDLR is special as its vertex cover time, i.e., the expected time of visiting all $n$ vertices of a graph, is $O(n^2)$, optimal among first-order random walks (Equation 4) that use degrees of endpoints.

A common practice is to add non-backtracking property that enforces $v_{t+1} \neq v_{t-1}$ (Tönshoff et al., 2023), and we also find this beneficial. In Appendix A.1 we extend Equation 3 and Proposition 2.2 to second-order walks that include non-backtracking and node2vec walks (Grover & Leskovec, 2016).

**Recording function** A recording function $q : (v_0 \rightarrow \cdots \rightarrow v_l, G) \mapsto \mathbf{z}$ takes a random walk and produces a machine-readable record $\mathbf{z}$. We let $q(\cdot, G)$ have access to the graph $G$ the walk is taking place. This allows recording auxiliary information such as vertex or edge attributes. A recording function is invariant if it satisfies the following for any given random walk $v_{[\cdot]}$ on $G$:

$$q(v_0 \rightarrow \cdots \rightarrow v_l, G) = q(\pi(v_0) \rightarrow \cdots \rightarrow \pi(v_l), H), \quad \forall G \overset{\pi}{\simeq} H. \tag{7}$$

Invariance requires that $q(\cdot, G)$ produces the same record $\mathbf{z}$ regardless of re-indexing of vertices of $G$ into $H$. For this, we have to be careful in how we represent each vertex in a walk $v_0 \rightarrow \cdots \rightarrow v_l$ as a machine-readable value, and which auxiliary information we record from $G$. For example, identity recording used in DeepWalk (Perozzi et al., 2014) and node2vec (Grover & Leskovec, 2016) are not invariant as the result is sensitive to vertex re-indexing. We highlight two choices which are invariant:

- **Anonymization.** We name each vertex in a walk with a unique integer, starting from $v_0 \mapsto 1$ and incrementing it based on their order of discovery. For instance, a random walk $a \rightarrow b \rightarrow c \rightarrow a$ translates to a sequence $1 \rightarrow 2 \rightarrow 3 \rightarrow 1$.

- **Anonymization + named neighbors.** While applying anonymization for each vertex $v$ in a walk, we record its neighbors $u \in N(v)$ if they are already named but the edge $v \rightarrow u$ has not been recorded yet. For instance, a walk $a \rightarrow b \rightarrow c \rightarrow d$ on a fully-connected graph translates to a sequence $1 \rightarrow 2 \rightarrow 3\langle 1 \rangle \rightarrow 4\langle 1, 2 \rangle$, where $\langle \cdot \rangle$ represents the named neighbors.

The pseudocode of both algorithms can be found in Appendix A.2. We now show the following:

**Proposition 2.3.** *A recording function $q : (v_0 \rightarrow \cdots \rightarrow v_l, G) \mapsto \mathbf{z}$ that uses anonymization, optionally with named neighbors, is invariant.*

While anonymization was originally motivated by privacy concerns in Micali & Zhu (2016) (see also Appendix A.10), Proposition 2.3 offers a new justification based on invariance. Our design of recording named neighbors is novel, and is inspired by sublinear algorithms that probe previously discovered neighbors in a walk (Dasgupta et al., 2014). In our context, it is useful since whenever a walk visits a set of vertices $S \subseteq V(G)$ it automatically records the entire induced subgraph $G[S]$. As a result, to record all edges of a graph, a walk only has to visit all vertices. While traversing all edges, i.e. edge cover time, is $O(n^3)$ (Zuckerman, 1991) in general, it is possible to choose a walk algorithm that takes only $O(n^2)$ time to visit all $n$ vertices. MDLR in Equation 6 exactly achieves this.

**Reader neural network** A reader neural network $f_\theta : \mathbf{z} \mapsto \hat{\mathbf{y}}$ processes the record $\mathbf{z}$ of the random walk and outputs a prediction $\hat{\mathbf{y}}$ in $\mathbb{R}^d$. As in Proposition 2.1, there is no invariance constraint imposed on $f_\theta$, and any neural network that accepts the recorded walks and has a sufficient expressive power can be used (we will make this precise in Section 3.1). This is in contrast to MPNNs where invariance is hard-coded in feature mixing operations. Also, our record $\mathbf{z}$ can take any format, such as a matrix, byte sequence, or plain text, as long as $f_\theta$ accepts it. Thus, it is possible (while not required) to choose the record to be plain text, such as `"1-2-3-1"`, and choose $f_\theta$ to be a pre-trained language model. This offers expressive power (Yun et al., 2020) and has a potential benefit of knowledge transfer from the language domain (Lu et al., 2022; Rothermel et al., 2021).

## 2.2 VERTEX-LEVEL TASKS

We now consider vertex-level tasks. In case of finite (small) graphs, we may simply frame a vertex-level task $(G, v) \mapsto \mathbf{y}$ as a graph-level task $G' \mapsto \mathbf{y}$ where $G'$ is $G$ with its vertex $v$ marked. Then, we can solve $G' \mapsto \mathbf{y}$ by querying an RWNN $X_\theta(G, v)$ to start its walk at $v_0 = v$.

A more interesting case is when $G$ is an infinite graph that is only locally finite, i.e. has finite degrees. This simulates problems on a large graph such as transductive classification. In this case, we may assume that our target function depends on finite local structures (Tahmasebi et al., 2023).

Let $r \geq 1$, and $\mathbb{B}_r := \{B_r(v)\}$ be the collection of local balls in $G$ of radius $r$ centered at $v \in V(G)$. We would like to model a vertex-level function $\phi : \mathbb{B}_r \to \mathbb{R}^d$ on $G$ using an RWNN $X_\theta(\cdot)$ by querying the vertex of interest, $X_\theta(G, v)$. We assume that $\phi$ is isomorphism invariant:

$$\phi(B_r(v)) = \phi(B_r(u)), \quad \forall B_r(v) \simeq B_r(u). \tag{8}$$

The probabilistic invariance of $X_\theta(\cdot)$ is defined as follows:

$$X_\theta(G, v) \stackrel{d}{=} X_\theta(G, u), \quad \forall B_r(v) \simeq B_r(u). \tag{9}$$

We can achieve probabilistic invariance by choosing the walk algorithm and recording function similar to the graph-level case[3]. As a modification, we query $X_\theta(G, v)$ with the starting vertex $v_0 = v$ of the random walk. Anonymization informs $v$ to the reader NN by always naming it as 1.

Then, we make a key choice of localizing random walks with restarts. That is, we reset a walk to its starting vertex $v_0$ either with a probability $\alpha \in (0, 1)$ at each step or periodically every $k$ steps. Restarting walks tend to stay more around starting vertex $v_0$, and were used to implement locality bias in personalized PageRank algorithm for search (Page et al., 1999). This localizing effect is crucial in our context since a walk can drift away from $B_r(v_0)$ before recording all necessary information, which may take an infinite time to return as $G$ is infinite (Janson & Peres, 2012). Restarts make the return to $v_0$ mandatory, ensuring that $B_r(v_0)$ can be recorded in a finite expected time. We show:

**Theorem 2.4.** *For a uniform random walk on an infinite graph $G$ starting at $v$, the vertex and edge cover times of the finite local ball $B_r(v)$ are not always finitely bounded.*

**Theorem 2.5.** *In Theorem 2.4, if the random walk restarts at $v$ with any nonzero probability $\alpha$ or any period $k \geq r + 1$, the vertex and edge cover times of $B_r(v)$ are always finite.*

## 3 ANALYSIS

In Section 2, we have described the design of RWNNs, primarily relying on the principle of (probabilistic) invariance. In this section, we provide in-depth analysis on their expressive power and relations to the issues in MPNNs such as over-smoothing, over-squashing, and under-reaching.

### 3.1 EXPRESSIVE POWER

Intuitively, if the records of random walks contain enough information such that the structures of interest, e.g., graph $G$ or local ball $B_r(v)$, can be fully recovered, a powerful reader NN such as an MLP (Hornik et al., 1989) or a transformer (Yun et al., 2020) on these records would be able to approximate any function of interest. Our analysis formalizes this intuition.

We first consider using an RWNN $X_\theta(\cdot)$ to universally approximate graph-level functions $\phi(\cdot)$ in probability, as defined in Abboud et al. (2021).

**Definition 3.1.** *$X_\theta(\cdot)$ is a universal approximator of graph-level functions in probability if, for all invariant functions $\phi : \mathbb{G}_n \to \mathbb{R}$ for a given $n \geq 1$, and $\forall \epsilon, \delta > 0$, there exist choices of length $l$ of the random walk and network parameters $\theta$ such that the following holds:*

$$\mathrm{Prob}[|\phi(G) - X_\theta(G)| < \epsilon] > 1 - \delta, \quad \forall G \in \mathbb{G}_n. \tag{10}$$

We show that, if the random walk is long enough and the reader NN $f_\theta$ is universal, an RWNN $X_\theta(\cdot)$ is capable of graph-level universal approximation. The length of the walk $l$ controls the confidence $> 1 - \delta$, with edge cover time $C_E(G)$ or vertex cover time $C_V(G)$ playing a central role.

---

[3]This is assuming that the random walks are localized in $B_r(v)$ and $B_r(u)$, e.g. with restarts.

**Theorem 3.2.** *An RWNN $X_\theta(\cdot)$ with a sufficiently powerful $f_\theta$ is a universal approximator of graph-level functions in probability (Definition 3.1) if it satisfies either of the below:*

- *It uses anonymization to record random walks of lengths $l > C_E(G)/\delta$.*

- *It uses anonymization + named neighbors to record walks of lengths $l > C_V(G)/\delta$.*

While both cover times are $O(n^3)$ for uniform random walks (Aleliunas et al., 1979; Zuckerman, 1991), we can use MDLR in Equation 6 to achieve an $O(n^2)$ vertex cover time, in conjunction with named neighbors recording. While universality can be in principle achieved with uniform random walks, our design reduces the worst-case length $l$ required for the desired reliability $> 1 - \delta$.

We now show an analogous result for the universal approximation of vertex-level functions.

**Definition 3.3.** *$X_\theta(\cdot)$ is a universal approximator of vertex-level functions in probability if, for all invariant functions $\phi : \mathbb{B}_r \to \mathbb{R}$ for a given $r \geq 1$, and $\forall \epsilon, \delta > 0$, there exist choices of length $l$ and restart probability $\alpha$ or period $k$ of the random walk and network parameters $\theta$ such that:*

$$\text{Prob}[|\phi(B_r(v)) - X_\theta(G, v)| < \epsilon] > 1 - \delta, \quad \forall B_r(v) \in \mathbb{B}_r. \tag{11}$$

Accordingly, using local vertex cover time $C_V(B_r(v))$ and edge cover time $C_E(B_r(v))$, we show:

**Theorem 3.4.** *An RWNN $X_\theta(\cdot)$ with a sufficiently powerful $f_\theta$ and any nonzero restart probability $\alpha$ or restart period $k \geq r + 1$ is a universal approximator of vertex-level functions in probability (Definition 3.3) if it satisfies either of the below for all $B_r(v) \in \mathbb{B}_r$:*

- *It uses anonymization to record random walks of lengths $l > C_E(B_r(v))/\delta$.*

- *It uses anonymization + named neighbors to record walks of lengths $l > C_V(B_r(v))/\delta$.*

Since restarts are required to finitely bound the local cover times on an infinite graph (Section 2.2), non-restarting walks cannot support vertex-level universality in general, and our design is obligatory.

### 3.2 OVER-SMOOTHING, OVER-SQUASHING, AND UNDER-REACHING

Often, in MPNNs, each layer operates by passing features over edges and mixing them using weights deduced from e.g., adjacency matrix. This ties them to the topology of the input graph, and a range of prior work has shown how this relates to the well-known issues of over-smoothing, over-squashing, and under-reaching. We connect RWNNs to these results to verify if similar issues may take place.

Let $G$ be a connected non-bipartite graph with row-normalized adjacency matrix $P$. We consider a linearized MPNN, where the vertex features $\mathbf{h}^{(0)}$ are initialized as some probability vector $\mathbf{x}$, and updated by $\mathbf{h}^{(t+1)} = \mathbf{h}^{(t)} P$. This simplification is often useful in understanding the aforementioned issues (Giraldo et al., 2023; Zhao & Akoglu, 2019). Specifically, in this model:

- Over-smoothing happens as the features exponentially converge to a stationary vector $\mathbf{h}^{(l)} \to \boldsymbol{\pi}$ as $l \to \infty$, smoothing out the input $\mathbf{x}$ (Giraldo et al., 2023).

- Over-squashing and under-reaching occur when a feature $\mathbf{h}_u^{(l)}$ becomes insensitive to distant input $\mathbf{x}_v$. While under-reaching refers to insufficient depth $l < \text{diam}(G)$ (Barceló et al., 2020), over-squashing refers to features getting overly compressed at bottlenecks of $G$, even with sufficient depth $l$. The latter is described by the Jacobian $|\partial \mathbf{h}_u^{(l)} / \partial \mathbf{x}_v| \leq [\sum_{t=0}^{l} P^t]_{uv}$,[4] as the bound often decays exponentially with $l$ (Topping et al., 2022; Black et al., 2023).

What do these results tell us about RWNNs? We can see that, while $P$ drives feature mixing in the message passing schema, it can be also interpreted as the transition probability matrix of uniform random walk where $P_{uv}$ is the probability of walking from $u$ to $v$. This parallelism motivates us to design an analogous, simplified RWNN and study its behavior.

We consider a simple RWNN that runs a uniform random walk $v_0 \to \cdots \to v_l$, reads the record $\mathbf{x}_{v_0} \to \cdots \to \mathbf{x}_{v_l}$ by averaging, and outputs it as $\mathbf{h}^{(l)}$. Like linear MPNN, the model involves $l$ steps

---

[4]This bound is obtained by applying Lemma 3.2 of Black et al. (2023) to our linearized MPNN.

of time evolution through $P$. However, while MPNN uses $P$ to process features, this model uses $P$ only to obtain a record of input, with feature processing decoupled. We show that in this model, over-smoothing does not occur as in simple MPNNs[5]:

**Theorem 3.5.** *The simple RWNN outputs* $\mathbf{h}^{(l)} \to \mathbf{x}^\top \boldsymbol{\pi}$ *as* $l \to \infty$.

Even if the time evolution through $P$ happens fast (i.e. $P$ is rapidly mixing) or with many steps $l$, the model is resistant to over-smoothing as the input $\mathbf{x}$ always affects the output. In fact, if $P$ is rapidly mixing, we may expect improved behaviors based on Section 3.1 as the cover times could reduce.

On the other hand, we show that over-squashing manifests as probabilistic under-reaching:

**Theorem 3.6.** *Let* $\mathbf{h}_u^{(l)}$ *be output of the simple RWNN queried with* $u$. *Then:*

$$\mathbb{E}\left[\left\|\frac{\partial \mathbf{h}_u^{(l)}}{\partial \mathbf{x}_v}\right\|\right] = \frac{1}{l+1}\left[\sum_{t=0}^{l} P^t\right]_{uv} \to \boldsymbol{\pi}_v \quad as \quad l \to \infty. \tag{12}$$

The equation shows that the feature Jacobians are bounded by the sum of powers of $P$, same as in simple MPNN. Both models are subject to over-squashing phenomenon that is similarly formalized, but manifests through different mechanisms. While in message passing the term is related to over-compression of features at bottlenecks (Topping et al., 2022), in RWNNs it is related to exponentially decaying probability of reaching a distant vertex $v$, i.e., probabilistic under-reaching.

In many MPNNs, it is understood that the topology of the input graph inevitably induces a trade-off between over-smoothing and over-squashing (Nguyen et al., 2023; Giraldo et al., 2023). Our results suggest that RWNNs avoid the trade-off, and we can focus on overcoming under-reaching e.g., with long or rapidly-mixing walks, while not worrying much about over-smoothing. Design choices such as MDLR (Equation 6) and non-backtracking (Alon et al., 2007) can be understood as achieving this.

## 4 RELATED WORK

We briefly review the related work. An extended discussion can be found in Appendix A.10.

**Random walks for learning on graphs** In graph learning, random walks have received interest due to their compatibility with sequence learning methods (Table 1). DeepWalk (Perozzi et al., 2014) and node2vec (Grover & Leskovec, 2016) used skip-gram models on walks. CRaWl (Tönshoff et al., 2023) used 1D CNN on sliding window-based walk records, with expressive power bounded by the window size. We discuss the relation between CRaWl and our approach in depth in Appendix A.10. AgentNet (Martinkus et al., 2023) learns agents that walk on a graph while recurrently updating their features. While optimizing the walk strategy, this may trade off speed as training requires recurrent roll-out of the network. Our method allows pairing simple and fast walkers, such as MDLR, with parallelizable NNs such as transformers. WalkLM (Tan et al., 2023) proposed a fine-tuning method for language models on walks on text-attributed graphs. By utilizing anonymization, our approach is able to process graphs even if no text attribute is given. Lastly, a concurrent work (Wang & Cho, 2024) has arrived at a similar use of anonymization combined with RNNs.

**Probabilistic invariant neural networks** Whenever a learning problem is compatible with symmetry, incorporating the associated invariance structure to the hypothesis class often leads to generalization benefit (Bronstein et al., 2021; Elesedy, 2022; 2023). This is also the case for probabilistic invariant NNs (Lyle et al., 2020; Bloem-Reddy & Teh, 2020), which includes our approach. Probabilistic invariant NNs have recently gained interest due to their potential of achieving higher expressive powers compared to deterministic counterparts (Cotta et al., 2023; Cornish, 2024). In graph learning, this is often achieved with stochastic symmetry breaking between vertices using randomized features (Loukas, 2020; Puny et al., 2020; Abboud et al., 2021; Kim et al., 2022), vertex orderings (Murphy et al., 2019; Kim et al., 2023), or dropout (Papp et al., 2021). Our approach can be understood as using random walk as a symmetry-breaking mechanism for probabilistic invariance, which provides an additional benefit of natural compatibility with sequence learning methods.

---

[5]While we show not forgetting $\mathbf{x}$ for brevity, we may extend to initial vertex $v_0$ using its anonymization as 1.

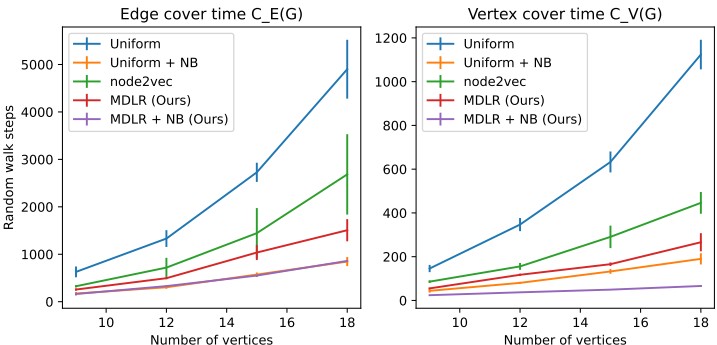

Figure 2: Cover times of random walks on varying sizes of lollipop graphs. NB is non-backtracking.

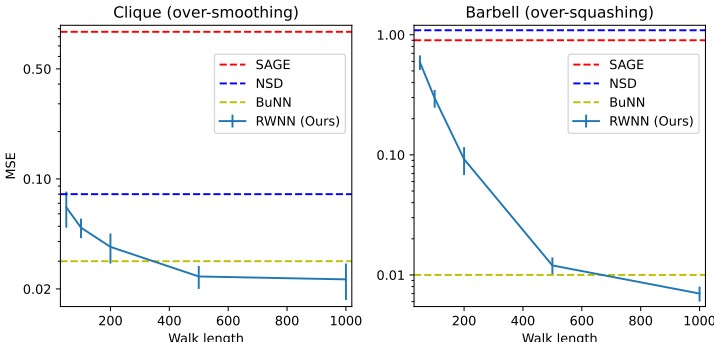

Figure 3: Over-smoothing and over-squashing (MSE ↓) for various walk lengths $l$.

Table 2: Over-smoothing and over-squashing results in comparison to baselines from Bamberger et al. (2024). We report test mean squared error (MSE ↓) aggregated for 4 randomized runs.

| Method | Clique (over-smoothing) | Barbell (over-squashing) |
|---|---|---|
| Baseline 1 | 30.94 ± 0.42 | 30.97 ± 0.42 |
| Baseline 2 | 0.99 ± 0.08 | 1.00 ± 0.07 |
| MLP | 1.10 ± 0.08 | 1.08 ± 0.07 |
| GCN | 29.65 ± 0.34 | 1.05 ± 0.08 |
| SAGE | 0.86 ± 0.10 | 0.90 ± 0.29 |
| GAT | 20.97 ± 0.40 | 1.07 ± 0.09 |
| NSD | 0.08 ± 0.02 | 1.09 ± 0.15 |
| BuNN | 0.03 ± 0.01 | 0.01 ± 0.07 |
| RWNN-transformer (Ours) | **0.023 ± 0.006** | **0.007 ± 0.001** |

## 5 EXPERIMENTS

We perform a set of experiments to demonstrate RWNNs. We implement random walks in C++ based on Chenebaux (2020), which produces good throughput (<0.1 seconds for 10k steps) without using GPUs as in Tönshoff et al. (2023); pseudocode is in Appendix A.2. We implement training and inference pipelines in PyTorch with NVIDIA GPUs and Intel Gaudi 2 compatibility. Supplementary experiments and analysis can be found in the Appendix, from A.4 to A.8.

### 5.1 SYNTHETIC EXPERIMENTS

On synthetic setups, we first verify our claims on cover times in Section 2.1, focusing on the utility of MDLR (Equation 6), non-backtracking, and named neighbor recording. We measure the edge cover times $C_E(G)$ and vertex cover times $C_V(G)$ on varying sizes of lollipop graphs $G$ (Feige, 1995) which are commonly used to establish the upper bounds of cover times. While the strict definition of edge cover requires traversing each edge in both directions, our measurements only require traversing one direction, which suffices for the universality of RWNNs. The results are in Figure 2. We find that **(1)** MDLR achieves a significant speed-up compared to uniform and node2vec walks, **(2)** adding non-backtracking significantly improves all first-order walks, and **(3)** edge cover times are often much larger than vertex cover times, strongly justifying the use of named neighbor recording (Theorem 3.2).

Table 3: Isomorphism learning results. We report accuracy of classifying training data rounded to the first decimal point. *, †, ‡, §, and ◇ are from Papp et al. (2021), Murphy et al. (2019), Zhao et al. (2022b), Martinkus et al. (2023), and Alvarez-Gonzalez et al. (2024), respectively.

| Method | CSL
1, 2-WL fails | SR16
3-WL fails | SR25
3-WL fails |
|---|---|---|---|
| GIN | 10.0%† | 50.0%§ | 6.7%‡ |
| PPGN | 100.0%§ | 50.0%§ | 6.7%‡ |
| GIN-AK+ | - | - | 6.7%‡ |
| PPGN-AK+ | - | - | 100.0%‡ |
| GIN+ELENE | - | - | 6.7%◇ |
| GIN+ELENE-L (ED) | - | - | 100.0%◇ |
| GIN-RNI | 16.0%* | - | - |
| GIN-RP | 37.6%† | - | - |
| GIN-Dropout | 82.0%* | - | - |
| RW-AgentNet | 100.0%§ | 50.5%§ | - |
| AgentNet | 100.0%§ | 100.0%§ | 6.7% |
| CRaWl | 100.0%§ | 100.0%§ | 46.6% |
| RWNN-DeBERTa (Ours) | 100.0% | 100.0% | 100.0% |

Then, we verify our claims in Section 3.2 using Clique and Barbell datasets of Bamberger et al. (2024) that explicitly test for over-smoothing and over-squashing, respectively. Our RWNN uses MDLR walks with non-backtracking, and the reader NN is a 1-layer transformer encoder with width 128, matching the baselines. Figure 3 shows the results for varying walk lengths $l$. The model performs well on Clique overall, while Barbell requires scaling the walk length. This agrees with our claims in Section 3.2 that RWNNs in general avoid over-smoothing, but over-squashing manifests as probabilistic under-reaching (and is therefore mitigated by sufficiently long walks). With $l = 1000$, our model in Table 2 achieves the best performance among the considered methods. More details on the experiment and additional baseline comparisons can be found in Appendix A.4.

## 5.2 GRAPH ISOMORPHISM LEARNING

We now verify the claims on expressive power in Section 3.1 using three challenging datasets where the task is recognizing the isomorphism type of input graph where certain WL test fails.

The Circular Skip Links (CSL) graphs dataset (Murphy et al., 2019) contains 10 non-isomorphic regular graphs with 41 vertices of degree 4. Distinguishing these graphs requires computing lengths of skip links, and 1-WL and 2-WL tests fail. The $4 \times 4$ rook's and Shrikhande graphs (Alvarez-Gonzalez et al., 2024), which we call SR16, are a pair of strongly regular graphs with 16 vertices of degree 6. Distinguishing the pair requires detecting 4-cliques, and 3-WL test fails. The SR25 dataset (Balcilar et al., 2021) contains 15 strongly regular graphs with 25 vertices of degree 12, on which 3-WL test fails. Examples of the graphs and random walks on them can be found in Appendix A.2.

Our RWNN uses MDLR walks with non-backtracking, and recording function with anonymization and named neighbors that produces plain text (Algorithm 3). We use pre-trained DeBERTa-base language model as the reader NN to leverage its capacity (see Appendix A.5 regarding pre-training), and fine-tune it with cross-entropy loss for ≤100k steps using AdamW optimizer with 2e-5 learning rate and 0.01 weight decay following Kim et al. (2023). We truncate the input to 512 tokens due to memory constraints. We use batch size 256, and accumulate gradients for 8 steps for SR25, which we found sufficient to stabilize the training. At test time, we ensemble 4 predictions by averaging logits.

The results are in Table 3. MPNNs that align with certain WL tests fail when asked to solve harder problems, e.g., GIN aligns with 1-WL and fails in CSL. A limited set of the state-of-the art NNs solve SR25, but at the cost of introducing specialized structural features (Alvarez-Gonzalez et al., 2024). Alternative approaches based on stochastic symmetry-breaking, e.g., random node identification, often fail on CSL although universal approximation is possible in theory (Abboud et al., 2021), possibly due to learning difficulties. For algorithms based on walks, AgentNet and CRaWl solve CSL and SR16, while failing on SR25. This is because learning a policy to walk in AgentNet can be challenging in complex tasks especially at the early stage of training, and the expressiveness of CRaWl is limited by the receptive field of the 1D CNN. Our approach based on DeBERTa language model overcomes the problems, demonstrating as the first RWNN that solves SR25.

In Appendix A.3, we further provide visualizations of learned attentions by mapping attention weights on text records of walks to input graphs. We find that the models often focus on sparse, connected substructures, which presumably provide discriminative information on the isomorphism types.

Table 4: Test accuracy on ogbn-arxiv. † denotes using validation labels for label propagation or in-context learning following Huang et al. (2020). For Llama 3, we ensemble 5 predictions by voting.

| Method | Accuracy |
|---|---|
| MLP | 55.50% |
| node2vec | 70.07% |
| GCN | 71.74% |
| GraphSAGE | 71.49% |
| GAT | 73.91% |
| RevGAT | 74.26% |
| Label propagation | 68.50% |
| C&S | 72.62% |
| C&S† | 74.02% |
| Llama2-13b zero-shot | 44.23% |
| GPT-3.5 zero-shot | 73.50% |
| DeBERTa, fine-tuned | 74.13% |
| Llama3-8b zero-shot | 51.31% |
| Llama3-8b one-shot | 52.81% |
| Llama3-8b one-shot† | 52.82% |
| Llama3-70b zero-shot | 65.30% |
| Llama3-70b one-shot† | 67.65% |
| RWNN-Llama3-8b (Ours) | 71.06% |
| RWNN-Llama3-8b (Ours)† | 73.17% |
| RWNN-Llama3-70b (Ours)† | **74.85%** |

## 5.3 REAL-WORLD TRANSDUCTIVE CLASSIFICATION

Previous results considered undirected, unattributed, and relatively small graphs. We now show that RWNN based on Llama 3 (Dubey et al., 2024) language models can solve real-world problem on a large graph with directed edges and textual attributes. We use ogbn-arxiv (Hu et al., 2020), a citation network of 169,343 arXiv papers with title and abstract. The task is transductive classification into 40 areas such as `"cs.AI"` using a set of labeled vertices. Additional results are in Appendix A.6.

We consider two representative types of baselines. The first reads title and abstract of each vertex using a language model and solves vertex-wise classification problem, ignoring graph structure. The second initializes vertex features as language model embeddings and trains an MPNN, at the risk of over-compressing the text. To take the best of both worlds, we design the recording function so that, not only it does basic operations such as anonymization, it records a complete information of the local subgraph including title and abstract, edge directions, and notably, **labels in case of labeled vertices** (Appendix A.2) (Sato, 2024). The resulting record naturally includes a number of input-label pairs of the classification task at hand, implying that we can frame transductive classification problem as a simple in-context learning problem (Brown et al., 2020). This allows training-free application of Llama 3 (Dubey et al., 2024) language model for transductive classification. We choose restart rate $\alpha = 0.7$ (Section 2.2) by hyperparameter search on a 100-sample subset of the validation data.

The results are in Table 4. Our models based on frozen Llama 3 perform competitively against a range of previous MPNNs on text embeddings, as well as outperforming language models that perform vertex-wise predictions ignoring graph structures such as GPT-3.5 and fine-tuned DeBERTa. Especially, our model largely outperforms one-shot baselines, which are given 40 randomly chosen labeled examples (one per class). This is surprising as our model observes fewer vertices, 31.13 in average, due to other recorded information. This is presumably since the record produced by our algorithm informs useful graph structure to the language model to quickly learn the task in-context compared to randomly chosen shots. In Appendix A.3, we visualize the attention weights, verifying that the model makes use of the graph structure recorded by random walks to make predictions.

As a final note, our approach is related to label propagation algorithms (Zhu & Ghahramani, 2002; Zhu, 2005; Grady, 2006) for transductive classification, which makes predictions by running random walks and probing the distribution of visited labeled vertices. The difference is that, in our approach, the reader NN i.e. the language model can appropriately use other information such as attributes, as well as do meaningful non-linear processing rather than simply probing the input. As shown in Table 4, our approach outperforms label propagation, verifying our intuition.

## 6 CONCLUSION

We contributed a principled understanding of RWNNs, where random walks on graphs are recorded and processed by NNs to make predictions. We analyzed invariance, expressive power, and information propagation, showing that we can design RWNNs to be invariant and universal without constraining its neural network. Experiments support the utility of our approach.

ACKNOWLEDGMENTS

This work was in part supported by the National Research Foundation of Korea (RS-2024-00351212 and RS-2024-00436165), the Institute of Information & communications Technology Planning & Evaluation (IITP) (RS-2022-II220926, RS-2024-00509279, RS-2021-II212068, RS-2022-II220959, and RS-2019-II190075) funded by the Korea government (MSIT), Artificial intelligence industrial convergence cluster development project funded by the Korea government (MSIT) & Gwangju Metropolitan City, and NAVER-Intel Co-Lab. We would like to thank Zeyuan Allen-Zhu for helpful discussion regarding anonymous random walks.

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

# A APPENDIX

## A.1 SECOND-ORDER RANDOM WALKS (SECTION 2)

In second-order random walks, the probability of choosing $v_{t+1}$ depends not only on $v_t$ but also on $v_{t-1}$. We consider non-backtracking (Alon et al., 2007; Fitzner & van der Hofstad, 2013) and node2vec (Grover & Leskovec, 2016) random walks as representatives.

A non-backtracking random walk is defined upon a first-order random walk algorithm, e.g. one in Equation 4, by enforcing $v_{t+1} \neq v_{t-1}$:

$$\text{Prob}[v_{t+1} = x | v_t = j, v_{t-1} = i] := \begin{cases} \dfrac{\text{Prob}[v_{t+1} = x | v_t = j]}{\sum_{y \in N(j) \setminus \{i\}} \text{Prob}[v_{t+1} = y | v_t = j]} & \text{for } x \neq i, \\ 0 & \text{for } x = i, \end{cases} \quad (13)$$

where $\text{Prob}[v_{t+1} = x | v_t = j]$ is the probability of walking $j \to x$ given by the underlying first-order random walk. Notice that the probabilities are renormalized over $N(j) \setminus \{i\}$. This is ill-defined in the case the walk traverses $i \to j$ and reaches a dangling vertex $j$ which has $i$ as its only neighbor, since $N(j) \setminus \{i\} = \emptyset$. In such cases, we allow the random walk to "begrudgingly" backtrack (Rappaport et al., 2017), $i \to j$ and then $j \to i$, given that it is the only possible choice due to the dangling of $j$.

In case of node2vec random walk (Grover & Leskovec, 2016), a weighting term $\alpha(v_{t-1}, v_{t+1})$ with two (hyper)parameters, return $p$ and in-out $q$, is introduced to modify the behavior of a first-order random walk:

$$\text{Prob}[v_{t+1} = x | v_t = j, v_{t-1} = i] := \frac{\alpha(i, x) \, \text{Prob}[v_{t+1} = x | v_t = j]}{\sum_{y \in N(j)} \alpha(i, y) \, \text{Prob}[v_{t+1} = y | v_t = j]}, \quad (14)$$

where the probability $\text{Prob}[v_{t+1} = x | v_t = j]$ is given by the underlying first-order random walk and $\alpha(v_{t-1}, v_{t+1})$ is defined as follows using the shortest path distance $d(v_{t-1}, v_{t+1})$:

$$\alpha(v_{t-1}, v_{t+1}) := \begin{cases} 1/p & \text{for } d(v_{t-1}, v_{t+1}) = 0, \\ 1 & \text{for } d(v_{t-1}, v_{t+1}) = 1, \\ 1/q & \text{for } d(v_{t-1}, v_{t+1}) = 2. \end{cases} \quad (15)$$

Choosing a large return parameter $p$ reduces backtracking since it decreases the probability of walking from $v_t$ to $v_{t-1}$. Choosing a small in-out parameter $q$ has a similar effect of avoiding $v_{t-1}$, with a slight difference that it avoids the neighbors of $v_{t-1}$ as well.

We now show an extension of Proposition 2.2 to the above second-order random walks:

**Proposition A.1.** *The non-backtracking random walk in Equation 13 and the node2vec random walk in Equation 14 are invariant if their underlying first-order random walk algorithm is invariant.*

*Proof.* The proof is given in Appendix A.9.3. $\qquad\square$

## A.2 MAIN ALGORITHM

We outline the main algorithms for the random walk and the recording function described in Section 2 and used in Section 5. Algorithm 1 shows the random walk algorithm, and Algorithms 2, 3, and 4 show the recording functions. For the random walk algorithm, we use non-backtracking described in Appendix A.1 and the minimum degree local rule conductance $c_G(\cdot)$ given in Equation 6 by default.

Based on the algorithms, in Figures 4, 5, and 6 we illustrate the task formats used for graph separation experiments in Section 5.2 by providing examples of input graphs, text records of random walks on them, and prediction targets for the language model that processes the records. Likewise, in Figures 7, 8, and 9 we show task formats for transductive classification on arXiv citation network in Section 5.3.

---

**Algorithm 1:** Random walk algorithm

---

**Data:** Input graph $G$, optional input vertex $v$, conductance function $c_G : E(G) \to \mathbb{R}_+$, walk
     length $l$, optional restart probability $\alpha \in (0, 1)$ or period $k > 1$
**Result:** Random walk $v_0 \to \cdots \to v_l$, restart flags $r_1, ..., r_l$
    `/* transition probabilities `$p : E(G) \to \mathbb{R}_+$` */`
1 **for** $(u, x) \in E(G)$ **do**
2     $p(u, x) \leftarrow c_G(u, x) / \sum_{y \in N(u)} c_G(u, y)$              `// Equation 4`
    `/* starting vertex `$v_0$` */`
3 **if** $v$ *is given* **then**
        `/* query starting vertex */`
4     $v_0 \leftarrow v$
5 **else**
        `/* sample starting vertex */`
6     $v_0 \sim \text{Uniform}(V(G))$
    `/* random walk `$v_1 \to \cdots \to v_l$`, restart flags `$r_1, ..., r_l$` */`
7 $v_1 \sim \text{Categorical}(\{p(v_0, x) : x \in N(v_0)\})$
8 $r_1 \leftarrow 0$
9 **for** $t \leftarrow 2$ **to** $l$ **do**
    `/* restart flag `$r_t$` */`
10     $r_t \leftarrow 0$
11     **if** $\alpha$ *is given* **then**
            `/* if restarted at `$t - 1$`, do not restart */`
12         **if** $r_{t-1} = 0$ **then**
13             $r_t \sim \text{Bernoulli}(\alpha)$
14     **else if** $k$ *is given* **then**
15         **if** $t \equiv 0 \pmod{k}$ **then**
16             $r_t \leftarrow 1$
    `/* random walk `$v_t$` */`
17     **if** $r_t = 0$ **then**
18         $S \leftarrow N(v_{t-1}) \setminus \{v_{t-2}\}$
19         **if** $S \neq \emptyset$ **then**
            `/* non-backtracking */`
20             $v_t \sim \text{Categorical}(\{p(v_{t-1}, x) / \sum_{y \in S} p(v_{t-1}, y) : x \in S\})$     `// Equation 13`
21         **else**
            `/* begrudgingly backtracking */`
22             $v_t \leftarrow v_{t-2}$
23     **else**
        `/* restart */`
24         $v_t \leftarrow v_0$

---

---

**Algorithm 2:** Recording function, using anonymization

**Data:** Random walk $v_0 \to \cdots \to v_l$, restart flags $r_1, ..., r_l$

**Result:** Text record $\mathbf{z}$

```
/* named vertices S, namespace id(·) */
```
1   $S \leftarrow \{v_0\}$
2   $\mathrm{id}(v_0) \leftarrow 1$
```
/* text record z */
```
3   $\mathbf{z} \leftarrow \mathrm{id}(v_0)$
4   **for** $t \leftarrow 1$ **to** $l$ **do**
```
    /* anonymization vₜ ↦ id(vₜ) */
```
5     **if** $v_t \notin S$ **then**
6       $S \leftarrow S \cup \{v_t\}$
7       $\mathrm{id}(v_t) \leftarrow |S|$
```
    /* text record z */
```
8     **if** $r_t = 0$ **then**
```
       /* record walk vₜ₋₁ → vₜ */
```
9       $\mathbf{z} \leftarrow \mathbf{z} + \texttt{"-"} + \mathrm{id}(v_t)$
10    **else**
```
       /* record restart vₜ = v₀ */
```
11       $\mathbf{z} \leftarrow \mathbf{z} + \texttt{";"} + \mathrm{id}(v_t)$

---

**Algorithm 3:** Recording function, using anonymization and named neighbors

**Data:** Random walk $v_0 \to \cdots \to v_l$, restart flags $r_1, ..., r_l$, input graph $G$

**Result:** Text record $\mathbf{z}$

```
/* named vertices S, namespace id(·), recorded edges T */
```
1   $S \leftarrow \{v_0\}$
2   $\mathrm{id}(v_0) \leftarrow 1$
3   $T \leftarrow \emptyset$
```
/* text record z */
```
4   $\mathbf{z} \leftarrow \mathrm{id}(v_0)$
5   **for** $t \leftarrow 1$ **to** $l$ **do**
```
    /* anonymization vₜ ↦ id(vₜ) */
```
6     **if** $v_t \notin S$ **then**
7       $S \leftarrow S \cup \{v_t\}$
8       $\mathrm{id}(v_t) \leftarrow |S|$
```
    /* text record z */
```
9     **if** $r_t = 0$ **then**
```
       /* record walk vₜ₋₁ → vₜ */
```
10      $\mathbf{z} \leftarrow \mathbf{z} + \texttt{"-"} + \mathrm{id}(v_t)$
11      $T \leftarrow T \cup \{(v_{t-1}, v_t), (v_t, v_{t-1})\}$
```
       /* named neighbors U */
```
12      $U \leftarrow N(v_t) \cap S$
13      **if** $U \neq \emptyset$ **then**
```
          /* sort in ascending order id(u₁) < ⋯ < id(u_|U|) */
```
14        $[u_1, ..., u_{|U|}] \leftarrow \mathrm{SortByKey}(U, \mathrm{id})$
15        **for** $u \leftarrow u_1$ **to** $u_{|U|}$ **do**
```
             /* record named neighbors vₜ → u */
```
16          **if** $(v_t, u) \notin T$ **then**
17            $\mathbf{z} \leftarrow \mathbf{z} + \texttt{"#"} + \mathrm{id}(u)$
18            $T \leftarrow T \cup \{(v_t, u), (u, v_t)\}$

19     **else**
```
       /* record restart vₜ = v₀ */
```
20      $\mathbf{z} \leftarrow \mathbf{z} + \texttt{";"} + \mathrm{id}(v_t)$

---

**Algorithm 4:** Recording function for transductive classification on arXiv network (Section 5.3)

---

**Data:** Directed input graph $G$, random walk $v_0 \to \cdots \to v_l$ and restart flags $r_1, ..., r_l$ on the undirected copy of $G$, paper titles $\{\mathbf{t}_v\}_{v \in V(G)}$ and abstracts $\{\mathbf{a}_v\}_{v \in V(G)}$, target category labels $\{\mathbf{y}_v\}_{v \in L}$ for labeled vertices $L \subset V(G)$

**Result:** Text record $\mathbf{z}$

```
/* named vertices S, namespace id(·), recorded edges T */
```
1   $S \leftarrow \{v_0\}$
2   $\mathrm{id}(v_0) \leftarrow 1$
3   $T \leftarrow \emptyset$
```
/* text record z */
/* record starting vertex */
```
4   $\mathbf{z} \leftarrow$ `"Paper 1 - Title: {`$\mathbf{t}_{v_0}$`}", Abstract: {`$\mathbf{a}_{v_0}$`}"`
5   **for** $t \leftarrow 1$ **to** $l$ **do**
```
        /* anonymization vt ↦ id(vt) */
```
6      **if** $v_t \notin S$ **then**
7          $S \leftarrow S \cup \{v_t\}$
8          $\mathrm{id}(v_t) \leftarrow |S|$
```
        /* text record z */
```
9      **if** $r_t = 0$ **then**
```
            /* record walk vt−1 → vt with direction */
```
10          **if** $(v_{t-1}, v_t) \in E(G)$ **then**
11              $\mathbf{z} \leftarrow \mathbf{z} +$ `" Paper {`$\mathrm{id}(v_{t-1})$`} cites Paper {`$\mathrm{id}(v_t)$`}"`
12          **else**
13              $\mathbf{z} \leftarrow \mathbf{z} +$ `" Paper {`$\mathrm{id}(v_{t-1})$`} is cited by Paper {`$\mathrm{id}(v_t)$`}"`
14          $T \leftarrow T \cup \{(v_{t-1}, v_t), (v_t, v_{t-1})\}$
```
            /* record title tv, abstract av, and label yv if v is labeled */
```
15          **if** $\mathrm{id}(v_t) = |S|$ **then**
16              $\mathbf{z} \leftarrow \mathbf{z} +$ `" - {`$\mathbf{t}_v$`}"`
17              **if** $v \in L$ **then**
18                  $\mathbf{z} \leftarrow \mathbf{z} +$ `", Category: {`$\mathbf{y}_v$`}"`
19              $\mathbf{z} \leftarrow \mathbf{z} +$ `", Abstract: {`$\mathbf{a}_v$`}"`
20          **else**
21              $\mathbf{z} \leftarrow \mathbf{z} +$ `"."`
```
            /* named neighbors U */
```
22          $U \leftarrow N(v_t) \cap S$
23          **if** $U \neq \emptyset$ **then**
```
                /* sort in ascending order id(u1) < · · · < id(u|U|) */
```
24              $[u_1, ..., u_{|U|}] \leftarrow \mathrm{SortByKey}(U, \mathrm{id})$
25              **for** $u \leftarrow u_1$ **to** $u_{|U|}$ **do**
```
                    /* record named neighbors vt → u with directions */
```
26                  **if** $(v_t, u) \notin T$ **then**
27                      **if** $(v_t, u) \in E(G)$ **then**
28                          $\mathbf{z} \leftarrow \mathbf{z} +$ `" Paper {`$\mathrm{id}(v_t)$`} cites Paper {`$\mathrm{id}(u)$`}."`
29                      **else**
30                          $\mathbf{z} \leftarrow \mathbf{z} +$ `" Paper {`$\mathrm{id}(v_t)$`} is cited by Paper {`$\mathrm{id}(u)$`}."`
31                      $T \leftarrow T \cup \{(v_t, u), (u, v_t)\}$

32      **else**
```
            /* record restart */
```
33          $\mathbf{z} \leftarrow \mathbf{z} +$ `" Restart at Paper 1."`

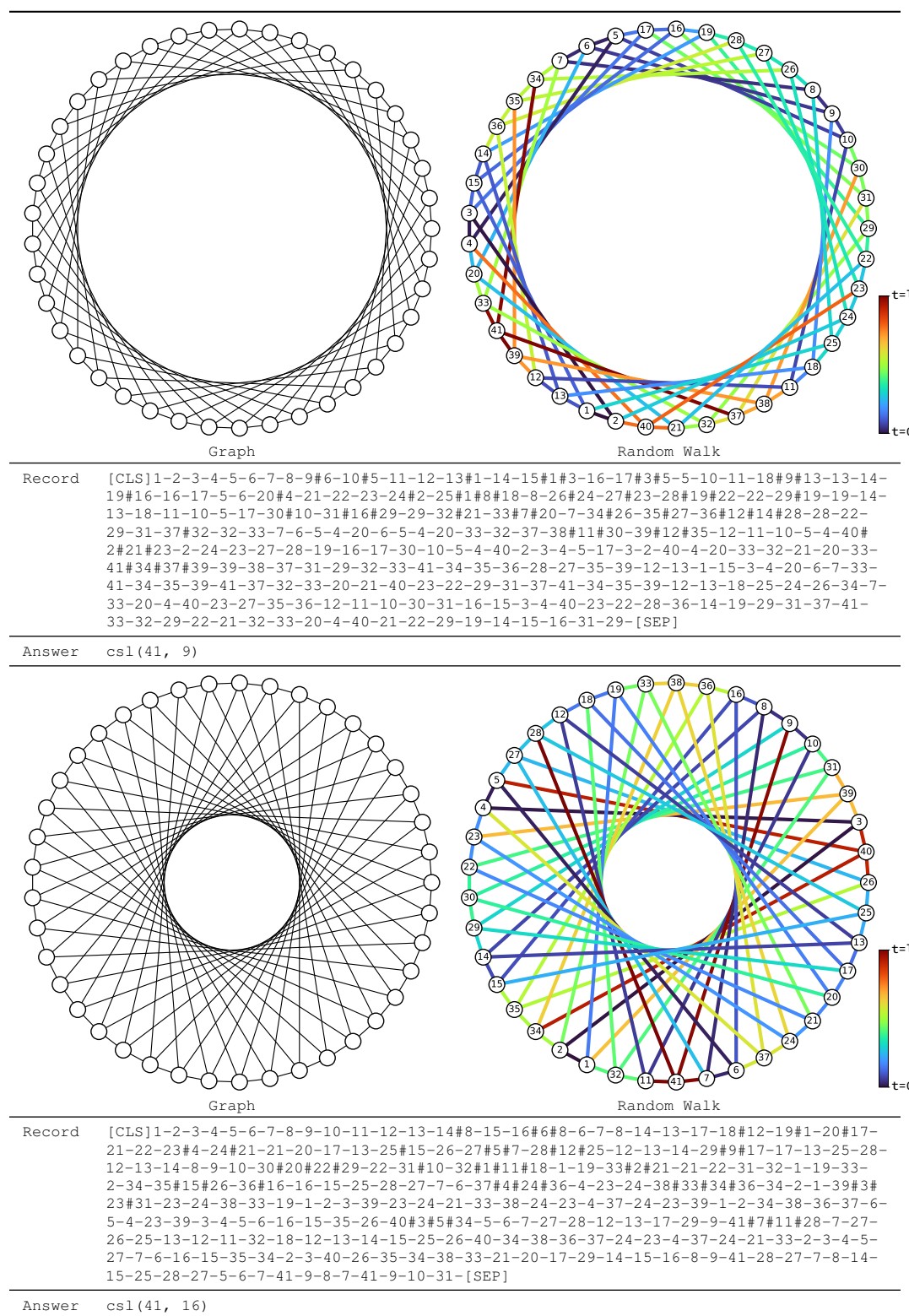

| | |
|---|---|
| Record | [CLS]1-2-3-4-5-6-7-8-9#6-10#5-11-12-13#1-14-15#1#3-16-17#3#5-5-10-11-18#9#13-13-14-19#16-16-17-5-6-20#4-21-22-23-24#2-25#1#8#18-8-26#24-27#23-28#19#22-22-29#19-19-14-13-18-11-10-5-17-30#10-31#16#29-29-32#21-33#7#20-7-34#26-35#27-36#12#14#28-28-22-29-31-37#32-32-33-7-6-5-4-20-6-5-4-20-33-32-37-38#11#30-39#12#35-12-11-10-5-4-40#2#21#23-2-24-23-27-28-19-16-17-30-10-5-4-40-2-3-4-5-17-3-2-40-4-20-33-32-21-20-33-41#34#37#39-39-38-37-31-29-32-33-41-34-35-36-28-27-35-39-12-13-1-15-3-4-20-6-7-33-41-34-35-39-41-37-32-33-20-21-40-23-22-29-31-37-41-34-35-39-12-13-18-25-24-26-34-7-33-20-4-40-23-27-35-36-12-11-10-30-31-16-15-3-4-40-23-22-28-36-14-19-29-31-37-41-33-32-29-22-21-32-33-20-4-40-21-22-29-19-14-15-16-31-29-[SEP] |
| Answer | csl(41, 9) |

| | |
|---|---|
| Record | [CLS]1-2-3-4-5-6-7-8-9-10-11-12-13-14#8-15-16#6#8-6-7-8-14-13-17-18#12-19#1-20#17-21-22-23#4-24#21-21-20-17-13-25#15-26-27#5#7-28#12#25-12-13-14-29#9#17-17-13-25-28-12-13-14-8-9-10-30#20#22#29-22-31#10-32#1#11#18-1-19-33#2#21-21-22-31-32-1-19-33-2-34-35#15#26-36#16-16-15-25-28-27-7-6-37#4#24#36-4-23-24-38#33#34#36-34-2-1-39#3#23#31-23-24-38-33-19-1-2-3-39-23-24-21-33-38-24-23-4-37-24-23-39-1-2-34-38-36-37-6-5-4-23-39-3-4-5-6-16-15-35-26-40#3#5#34-5-6-7-27-28-12-13-17-29-9-41#7#11#28-7-27-26-25-13-12-11-32-18-12-13-14-15-25-26-40-34-38-36-37-24-23-4-37-24-21-33-2-3-4-5-27-7-6-16-15-35-34-2-3-40-26-35-34-38-33-21-20-17-29-14-15-16-8-9-41-28-27-7-8-14-15-25-28-27-5-6-7-41-9-8-7-41-9-10-31-[SEP] |
| Answer | csl(41, 16) |

Figure 4: Two CSL graphs and text records of random walks from Algorithms 1 and 3. Task is graph classification into 10 isomorphism types $\mathrm{csl}(41, s)$ for skip length $s \in \{2, 3, 4, 5, 6, 9, 11, 12, 13, 16\}$. In random walks, we label vertices by anonymization and color edges by their time of discovery.

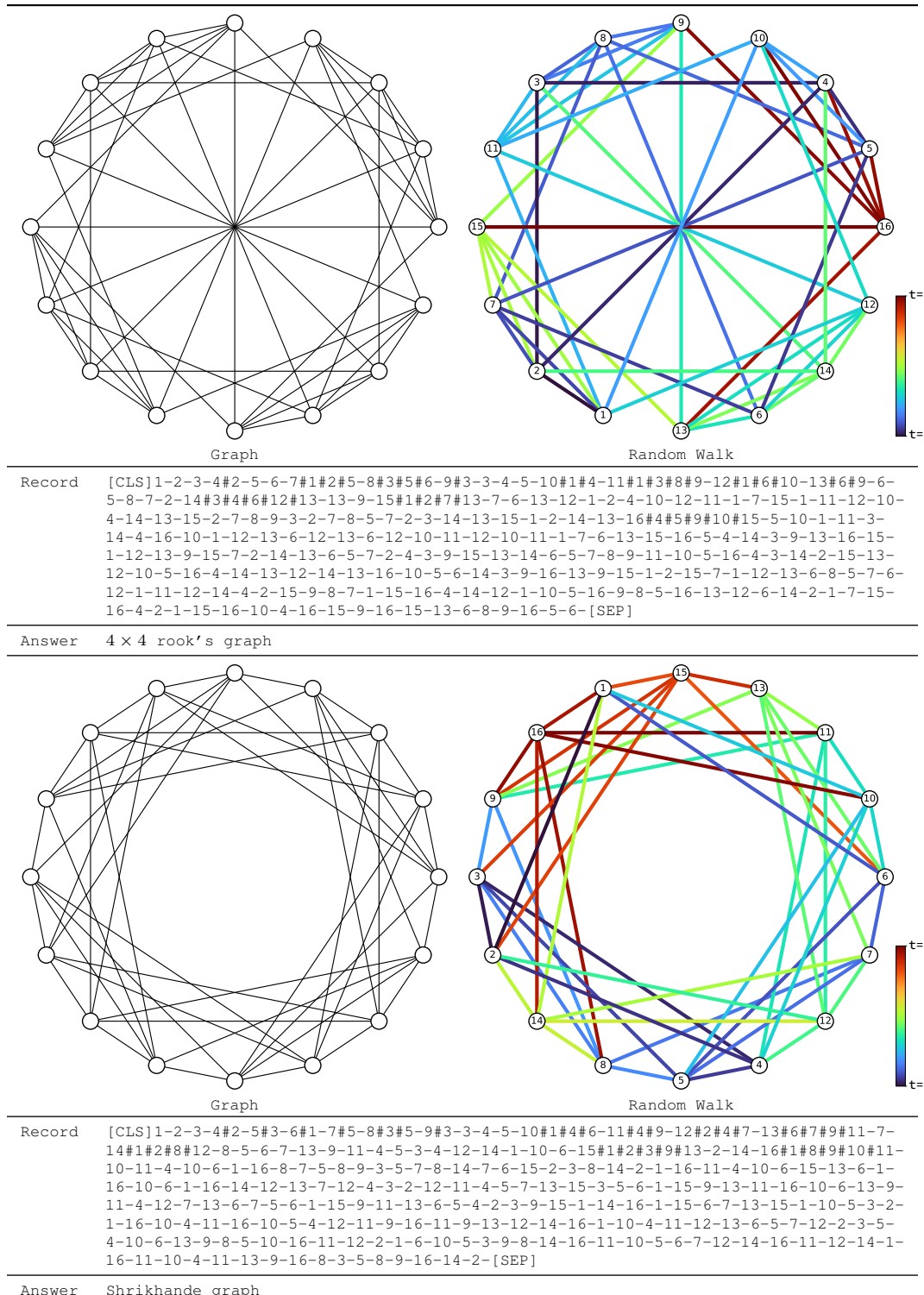

Figure 5: SR16 graphs and text records of random walks from Algorithms 1 and 3. The task is graph classification into two isomorphism types {4 × 4 rook's graph, Shrikhande graph}. In random walks, we label vertices by anonymization and color edges by their time of discovery.

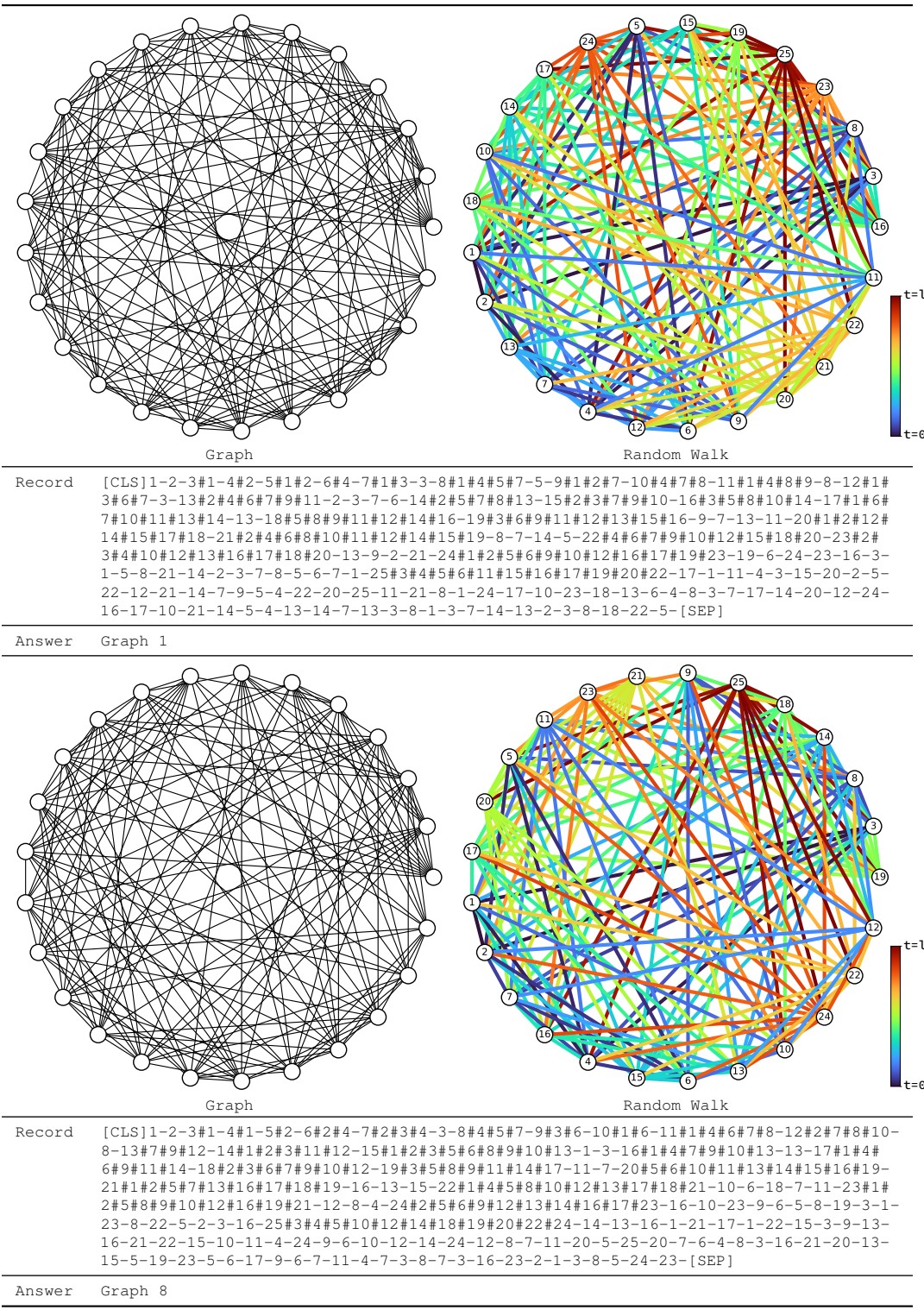

Figure 6: Two SR25 graphs and text records of random walks from Algorithms 1 and 3. The task is graph classification into 15 isomorphism types. In random walks, we label vertices by anonymization and color edges by their time of discovery.

| System | A walk on the arXiv citation network will be given. Predict the arXiv CS sub-category Paper 1 belongs to. It is one of the following: Numerical Analysis (cs.NA), Multimedia (cs.MM), Logic in Computer Science (cs.LO), ... Discrete Mathematics (cs.DM). Only respond with the answer, do not say any word or explain. |
|---|---|
| Record | A walk on arXiv citation network is as follows: Paper 1 – Title: Safety guided deep reinforcement learning via online gaussian process estimation, Abstract: An important facet of reinforcement learning (rl) has to do with how the agent goes about exploring the environment. traditional exploration strategies typically focus on efficiency and ignore safety. however, for practical applications, ensuring safety of the agent during exploration is crucial since performing an unsafe action or reaching an unsafe state could result in irreversible damage to the agent. the main challenge of safe exploration is that characterizing the unsafe states and actions... Restart at 1. Paper 1 is cited by 2 – Learning to walk in the real world with minimal human effort, Abstract: Reliable and stable locomotion has been one of the most fundamental challenges for legged robots. deep reinforcement learning (deep rl) has emerged as a promising method for developing such control policies... Restart at 1. Paper 1 cites 3 – Deep q learning from demonstrations, Category: Artificial Intelligence (cs.AI), Abstract: Deep reinforcement learning (rl) has achieved several high profile successes in difficult decision-making problems. however, these algorithms typically require a huge amount of data before they reach... Restart at 1. Paper 1 cites 4 – Constrained policy optimization, Category: Machine Learning (cs.LG), Abstract: For many applications of reinforcement learning it can be more convenient to specify both a reward function and constraints, rather than trying to design behavior through the reward function. for example,... Paper 4 is cited by 2. Paper 4 is cited by 5 – Artificial intelligence values and alignment, Abstract: This paper looks at philosophical questions that arise in the context of ai alignment. it defends three propositions. first, normative and technical aspects of the ai alignment problem are interrelated,... Paper 5 cites 6 – Agi safety literature review, Category: Artificial Intelligence (cs.AI), Abstract: The development of artificial general intelligence (agi) promises to be a major event. along with its many potential benefits, it also raises serious safety concerns (bostrom, 2014). the intention of... Paper 6 is cited by 7 – Modeling agi safety frameworks with causal influence diagrams, Abstract: Proposals for safe agi systems are typically made at the level of frameworks, specifying how the components of the proposed system should be trained and interact with each other. in this paper, we model... Restart at 1. Paper 1 cites 8 – Safe learning of regions of attraction for uncertain nonlinear systems with gaussian processes, Category: Systems and Control (cs.SY), Abstract: Control theory can provide useful insights into the properties of controlled, dynamic systems. one important property of nonlinear systems is the region of attraction (roa), a safe subset of the state... Paper 8 is cited by 9 – Control theory meets pomdps a hybrid systems approach, Abstract: Partially observable markov decision processes (pomdps) provide a modeling framework for a variety of sequential decision making under uncertainty scenarios in artificial intelligence (ai). since the... Restart at 1.

...

Which arXiv CS sub-category does Paper 1 belong to? Only respond with the answer, do not say any word or explain. |
| Answer | Machine Learning (cs.LG) |

Figure 7: Transductive classification on arXiv citation network using text record of random walk from Algorithms 1 and 4. Colors indicate task instruction, walk information, and label information.

| System | Title and abstract of an arXiv paper will be given. Predict the arXiv CS sub-category the paper belongs to. It is one of the following: Numerical Analysis (cs.NA), Multimedia (cs.MM), Logic in Computer Science (cs.LO), ... Discrete Mathematics (cs.DM). Only respond with the answer, do not say any word or explain. |
|---|---|
| Context | Title: Safety guided deep reinforcement learning via online gaussian process estimation
Abstract: An important facet of reinforcement learning (rl) has to do with how the agent goes about exploring the environment. traditional exploration strategies typically focus on efficiency and ignore safety. however, for practical applications, ensuring safety of the agent during exploration is crucial since performing an unsafe action or reaching an unsafe state could result in irreversible damage to the agent. the main challenge of safe exploration is that characterizing the unsafe states and actions... Which arXiv CS sub-category does this paper belong to? |
| Answer | Machine Learning (cs.LG) |

Figure 8: Zero-shot format for vertex classification on arXiv citation network. Task instruction and label information are colored.

| System | Title and abstract of an arXiv paper will be given. Predict the arXiv CS sub-category the paper belongs to. It is one of the following: Numerical Analysis (cs.NA), Multimedia (cs.MM), Logic in Computer Science (cs.LO), ... Discrete Mathematics (cs.DM). |
|---|---|
| Context | Title: Deeptrack learning discriminative feature representations online for robust visual tracking
Abstract: Deep neural networks, albeit their great success on feature learning in various computer vision tasks, are usually considered as impractical for online visual tracking, because they require very long...
Category: cs.CV

Title: Perceived audiovisual quality modelling based on decison trees genetic programming and neural networks...
Abstract: Our objective is to build machine learning based models that predict audiovisual quality directly from a set of correlated parameters that are extracted from a target quality dataset. we have used the...
Category: cs.MM

...

Title: Safety guided deep reinforcement learning via online gaussian process estimation
Abstract: An important facet of reinforcement learning (rl) has to do with how the agent goes about exploring the environment. traditional exploration strategies typically focus on efficiency and ignore safety. however, for practical applications, ensuring safety of the agent during exploration is crucial since performing an unsafe action or reaching an unsafe state could result in irreversible damage to the agent. the main challenge of safe exploration is that characterizing the unsafe states and actions... Which arXiv CS sub-category does this paper belong to? |
| Answer | cs.LG |

Figure 9: One-shot format for transductive classification on arXiv citation network. 40 labeled examples are given in form of multi-turn dialogue. Task instruction and label information are colored.

```
[...]  A walk on arXiv citation network is as follows:
Paper 1 - Title: Unsupervised and interpretable scene discovery with discrete attend infer
repeat, Abstract: In this work we present discrete attend infer repeat (discrete-air), a
recurrent auto-encoder with structured latent distributions containing discrete categori
cal distributions, continuous attribute distributions, and factorised spatial attention.
while inspired by the original air model andretaining air model's capability in identify
ing objects in an image, discrete-air provides direct interpretability of the latent codes.
we show that for multi-mnist and a multiple-objects version of dsprites... Restart at 1.
[...]  Restart at 1. Paper 1 cites 12 - Attend infer repeat fast scene understanding with
generative models, Category: Computer Vision and Pattern Recognition (cs.CV), Abstract: We
present a framework for efficient inference in structured image models that explicitly rea
son about objects. we achieve this by performing probabilistic inference using a recurrent
neural network that... Paper 12 cites 3. Paper 12 cites 7. Paper 12 is cited by 9. Paper
12 is cited by 10. Paper 12 cites 11. Restart at 1. Paper 1 cites 11. Restart at 1. Paper
1 cites 12. Paper 12 is cited by 13 - An architecture for deep hierarchical generative mod
els, Category: Machine Learning (cs.LG), Abstract: We present an architecture which lets us
train deep, directed generative models with many layers of latent variables. we include de
terministic paths between all latent variables and the generated output,... Paper 13 cites
3. Restart at 1.  [...]
```

|  | Attention (Layer 13/30) |
| --- | --- |
| Answer | cs.CV |
| Prediction | cs.CV |

Figure 10: Example of attention in a frozen Llama 3 8B applied on arXiv transductive classification. Attention weights are averaged over all 30 heads.

### A.3   ATTENTION VISUALIZATIONS

In Figure 10, we show an example of self-attention in the frozen Llama 3 8B model applied to arXiv transductive classification (Section 5.3). We show attention weights on text record of random walk from the generated cs token as query, which is just before finishing the prediction, e.g., cs.CV. We color the strongest activation with orange, and do not color values below $1\%$ of it. The model invests a nontrivial amount of attention on walk information such as  Paper 1 cites 12, while also making use of titles and labels of labeled vertices. This indicates the model is utilizing the graph structure recorded by the random walk in conjunction with other information to make predictions.

In Figures 11, 12, and 13, we show examples of self-attention in DeBERTa models trained for graph separation (Section 5.2). We first show attention weights on text record of random walk from the [CLS] query token. Then, we show an alternative visualization where the attention weights are mapped onto the original input graph. For example, for each attention on $v_{t+1}$ where the walk has traversed $v_t \rightarrow v_{t+1}$, we regard it as an attention on the edge $(v_t, v_{t+1})$ in the input graph. We ignore [CLS] and [SEP] tokens since they do not appear on the input graph. We observe that the models often focus on sparse, connected substructures, which presumably provide discriminative information on the isomorphism types. In particular, for CSL graphs (Figure 11) we observe an interpretable pattern of approximate cycles composed of skip links. This is presumably related to measuring the lengths of skip links, which provides sufficient information of isomorphism types of CSL graphs.

```
1-2-3-4-5-6-7-8#5-9#1#4-1-10-11#2-2-1-12#3#8-3-4-9-8-5-13-14#7-15#6-16-17-18-19#14#16-14-7-
8-12-1-9-20#3#10-21#2-22#11-23-24#10#21-10-1-12-25#4#7#13-7-6-5-4-25-12-8-5-4-25-12-8-7-25-
13-26#6#19-27#15#18-18-17-28-29-30-31-32#28-33#29-34#23#31-31-32-35#17#30-17-16-19-14-15-6-
7-8-5-6-7-8-9-4-3-12-1-10-20-9-8-7-14-19-16-15-6-5-4-9-8-5-4-9-20-21-22-11-36#24-37#23#31-
31-34-33-32-28-38#16#27-27-15-16-17-18-27-26-13-25-7-14-19-26-6-15-16-38-27-18-17-35-30-31-
34-23-24-21-2-1-10-11-22-39#34#36-40#30#33#37-33-29-30-35-32-33-40-30-35-32-31-37-36-39-34-
31-30-29-41#18#35#38-18-27-38-16-17-28-38-27-26-13-14-15-27-26-13-14-15-6-26-13-5-6-15-14-
13-25-7-14-15-6-26-27-38-28-29-33-34-39-36-24-10-1-2-
```

Attention (Head 12/12, Layer 12/12)

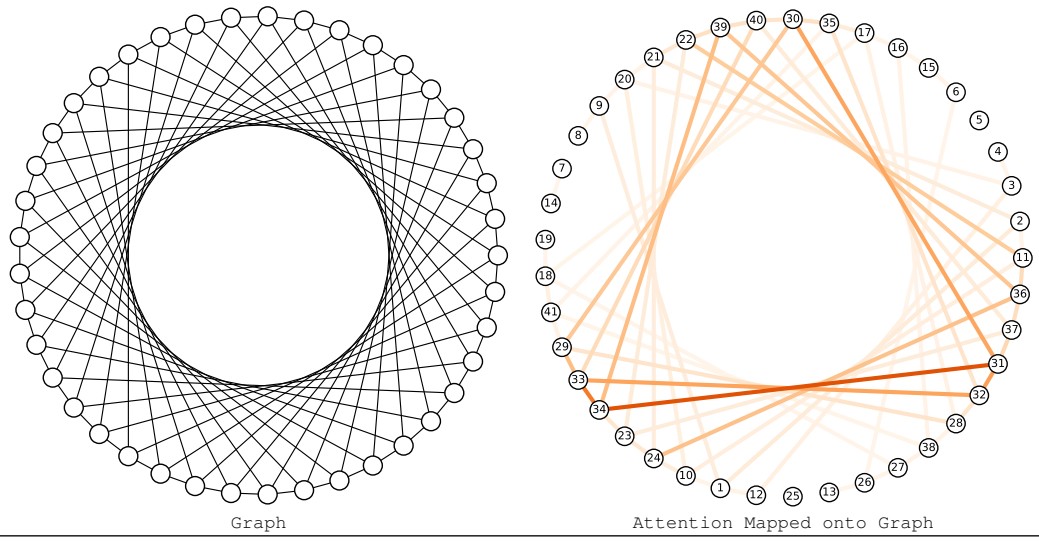

Graph        Attention Mapped onto Graph

```
1-2-3-4-5-6#3-7-8-9#1-10#7-7-11#5-5-6-3-4-12-13#5-14#11-15-16#8#11-11-5-13-12-4-5-6-3-4-12-
17#8-18#16-19#15-20-21-22-23#1-24-25#1#10-26-27-28-29#24#26-24-25-10-9-1-25-24-30#20#22-31#
19#29-29-28-27-32#22-33#21#28-21-34#13-35#12#18#20-12-17-36#2#4#9-9-10-25-26-27-32-22-23-24-
25-10-9-8-16-18-19-20-21-34-37#14#33-38#15#28#31-15-14-13-34-21-22-32-39#2#23-2-1-23-24-30-
31-38-15-19-31-30-24-29-31-38-15-19-20-35-18-19-15-14-37-34-35-20-21-34-37-33-32-22-21-34-
37-33-32-22-30-20-35-18-16-15-14-13-12-35-20-30-22-32-33-37-14-11-5-4-36-2-1-25-26-29-24-23-
39-32-22-23-24-25-1-2-36-9-10-25-24-30-31-19-15-16-11-14-37-38-15-14-11-7-10-40#6#26-6-7-10-
9-8-17-18-16-11-7-8-16-11-7-8-17-18-16-15-38-37-14-11-5-4-36-17-18-
```

Attention (Head 5/12, Layer 12/12)

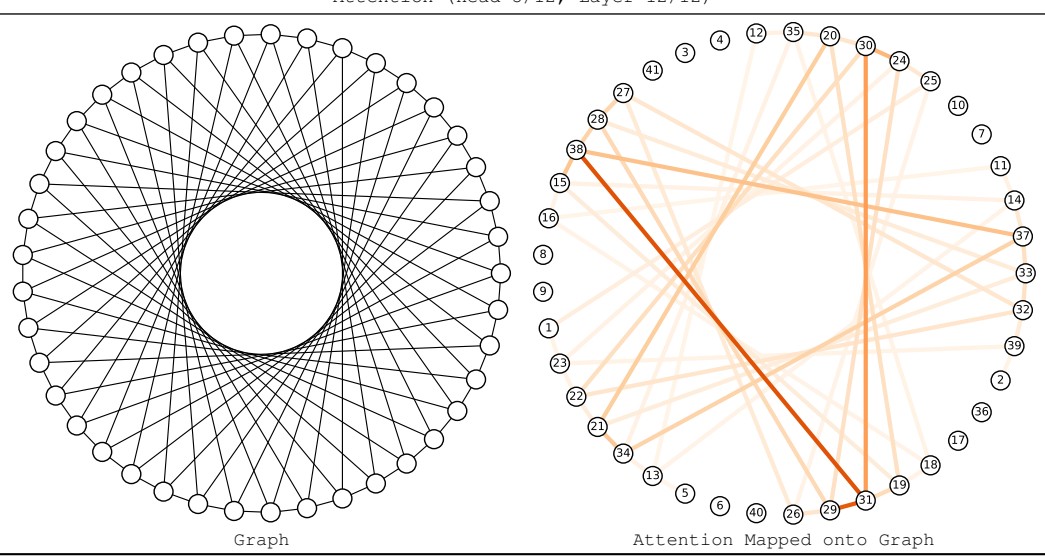

Graph        Attention Mapped onto Graph

Figure 11: Examples of attention from [CLS] token in a DeBERTa trained on CSL graph separation, forming approximate cycles using skip links. This is presumably related to measuring the lengths of skip links, which provides sufficient information to infer isomorphism types of CSL graphs.

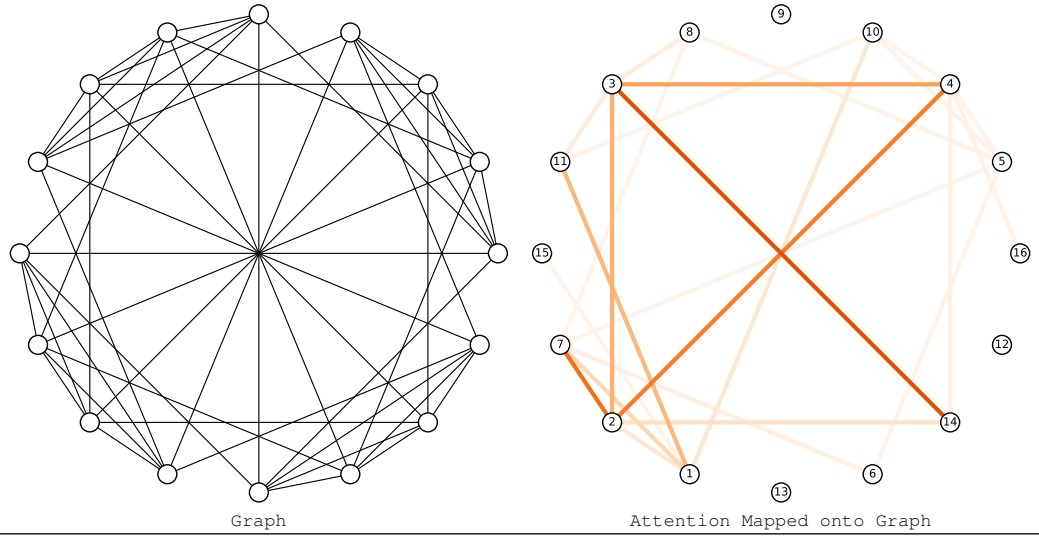

Figure 12: Examples of attention from [CLS] token in a DeBERTa trained on SR16 graph separation. The model tend to focus on named neighbor records that form a sparse connected substructure, which we conjecture to provide discriminative information on the isomorphism type of SR16 graphs.

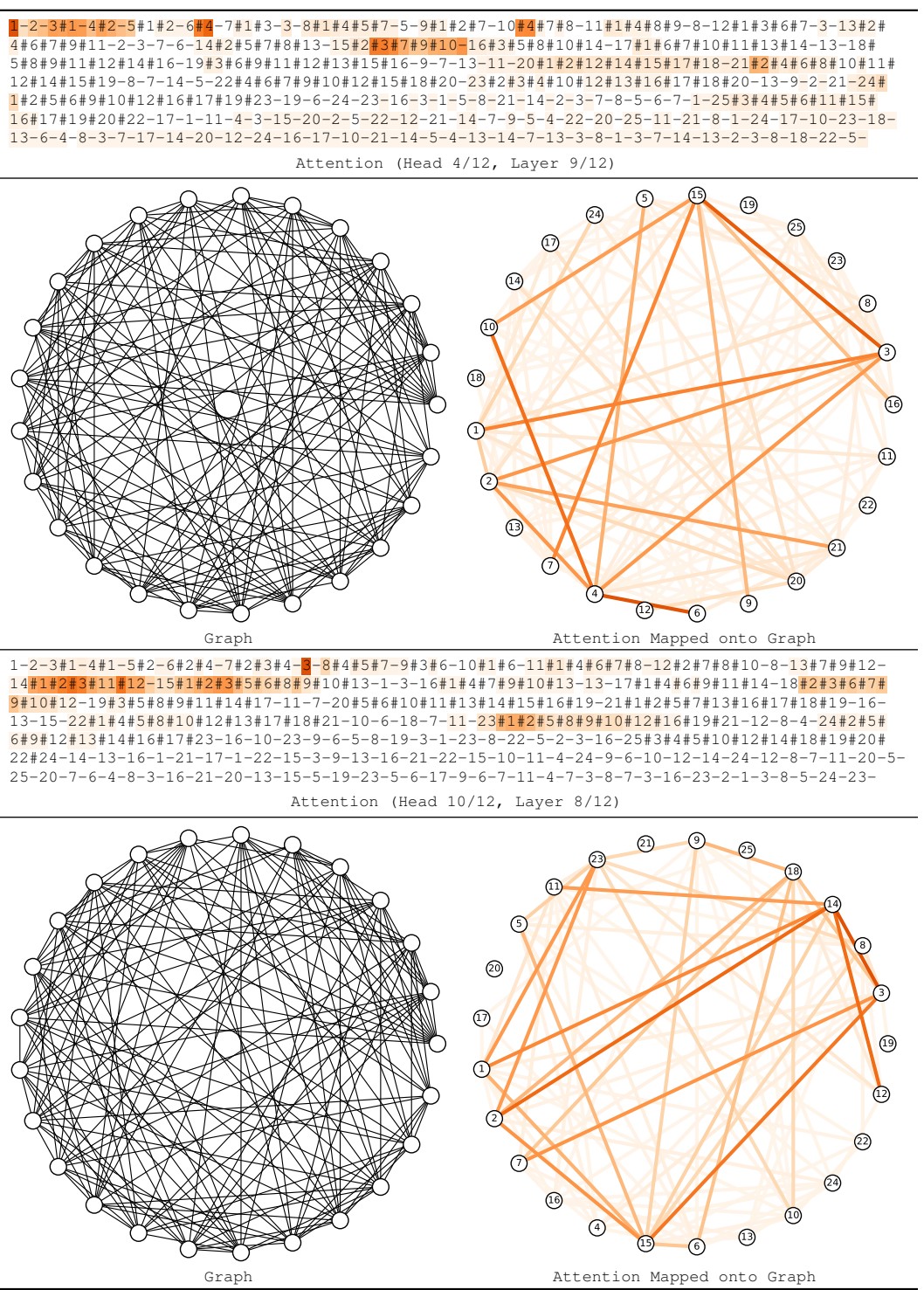

Figure 13: Examples of attention from `[CLS]` token in a DeBERTa trained on SR25 graph separation. The model tend to focus on named neighbor records that form a sparse connected substructure, which we conjecture to provide discriminative information on the isomorphism type of SR25 graphs.

Table 5: Over-smoothing and over-squashing results, extending Table 2 (Section 5.1). We report test MSE aggregated for 4 randomized runs, except for CRaWl which took >3 days per run.

| Method | Clique (over-smoothing) | Barbell (over-squashing) |
|---|---|---|
| GCN | 29.65 ± 0.34 | 1.05 ± 0.08 |
| SAGE | 0.86 ± 0.10 | 0.90 ± 0.29 |
| GAT | 20.97 ± 0.40 | 1.07 ± 0.09 |
| NSD | 0.08 ± 0.02 | 1.09 ± 0.15 |
| BuNN | 0.03 ± 0.01 | 0.01 ± 0.07 |
| $l = 100$ | | |
| RWNN-MLP | 0.082 ± 0.018 | 0.326 ± 0.035 |
| WalkLM-transformer | 0.096 ± 0.018 | 0.388 ± 0.048 |
| RWNN-transformer (Ours) | 0.049 ± 0.007 | 0.297 ± 0.050 |
| $l = 1000$ | | |
| CRaWl | 0.103 ± N/A | 0.289 ± N/A |
| CRaWl* | 0.121 ± N/A | 0.103 ± N/A |
| RWNN-MLP | 0.028 ± 0.026 | 0.015 ± 0.005 |
| WalkLM-transformer | **0.023 ± 0.003** | 0.037 ± 0.009 |
| RWNN-transformer (Ours) | **0.023 ± 0.006** | **0.007 ± 0.001** |

## A.4 SUPPLEMENTARY MATERIALS FOR SYNTHETIC EXPERIMENTS (SECTION 5.1)

We provide experimental details and additional baseline comparisons for the Clique and Barbell experiments in Section 5.1 we could not include in the main text due to space constraints.

**Dataset** In Clique and Barbell (Bamberger et al., 2024), each vertex of a graph $G$ is given a random scalar value. The task at each vertex $v$ is to regress the mean of all vertices with an opposite sign. In Clique, $G$ is a 20-clique, and its vertices are randomly partitioned into two sets of size 10, and one set is given positive values and the other is given negative values. In Barbell, two 10-cliques are linked to form $G$ and one clique is given positive values and the other is given negative values.

**Models** Our RWNN uses MDLR walks with non-backtracking, and since the task is vertex-level, for each $v$ we fix the starting vertex as $v_0 = v$. The recording function $q : (v_0 \to \cdots \to v_l, G) \mapsto \mathbf{z}$ takes a walk and produces a record $\mathbf{z} \coloneqq (\mathrm{id}(v_0), \mathbf{x}_{v_0}, \mathbf{x}_{v_0}) \to \cdots \to (\mathrm{id}(v_l), \mathbf{x}_{v_l}, \mathbf{x}_{v_0})$, where $\mathrm{id}(v)$ is the anonymization of $v$ and $\mathbf{x}_v$ is the scalar value of $v$. We find that recording $\mathbf{x}_{v_0}$ at each step improves training. The reader NN is a 1-layer transformer encoder, with 128 feature dimensions, sinusoidal positional embedding, and average pooling readout. To map the recording at each step $(\mathrm{id}(v_t), \mathbf{x}_{v_t}, \mathbf{x}_{v_0})$ to a 128-dimensional feature, we use a learnable vocabulary for $\mathrm{id}(v_t)$, a learnable linear layer for $\mathbf{x}_{v_t}$, and a 3-layer MLP with 32 hidden dimensions and tanh activation for $\mathbf{x}_{v_0}$.

We test additional baselines that use random walks. We test CRaWl (Tönshoff et al., 2023) using the official code, and only remove batch normalization since it led to severe overfitting for Clique and Barbell. Since CRaWl uses vertex-level average pooling, while RWNN-transformer uses walk-level, we also test a variant that uses walk-level pooling and denote it by CRaWl*. We also test RWNN-MLP, which uses an MLP on the concatenated walk features instead of a transformer. The MLP has 3 layers with 128 hidden dimensions and ReLU activation. We also test an adaptation of WalkLM (Tan et al., 2023). While the work entails two ideas, representation of walks and fine-tuning by masked auto-encoding, we focus on testing its choice of walks and their representation since the fine-tuning method is orthogonal to our work. We do this by modifying RWNN-transformer to use uniform walks and omit anonymization $\mathrm{id}(v_t)$ from the walk record, and denote the model by WalkLM-transformer.

We train all models with Adam optimizer with batch size 1 and learning rate 1e-4 for 500 epochs, closely following Bamberger et al. (2024) except for the lower learning rate which we found necessary for stabilizing training due to the stochasticity of walk based methods. For all RWNN variants, we perform 8 predictions per vertex during training, and average 32 predictions per vertex at test-time.

**Results** The results are in Table 5. All random walk based methods consistently perform better than GCN, SAGE, and GAT, and RWNN-transformer achieves the lowest test MSE. This supports our hypothesis that RWNNs can, in general, be resistant to over-smoothing and over-squashing (for the latter, if the walk is long), but also shows that proper design choices are important to obtain the best performance. For example, CRaWl and CRaWl* are limited by their window size of information processing (see Appendix A.10), and RWNN-MLP is bottlenecked by the fixed-sized hidden feature. While WalkLM-transformer performs on par with RWNN-transformer on Clique, its error is higher than RWNN-MLP on Barbell, showing the benefits of our design of walks and their records.

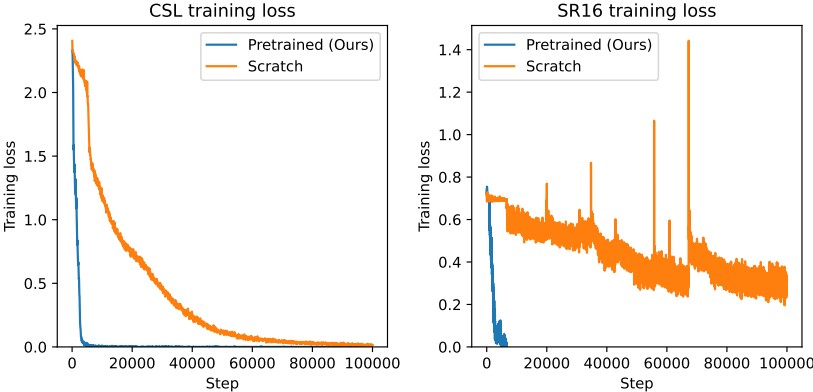

Figure 14: The impact of language pre-training on graph isomorphism learning (Section 5.2).

Table 6: Fine-tuning for arXiv, extending Table 4 (Section 5.3). † uses validation labels as in Table 4.

| Method | Training-free? | Accuracy |
|---|---|---|
| Llama3-8b zero-shot | ◯ | 51.31% |
| Llama3-8b one-shot | ◯ | 52.81% |
| Llama3-8b one-shot† | ◯ | 52.82% |
| Llama3-70b zero-shot | ◯ | 65.30% |
| Llama3-70b one-shot | ◯ | 67.65% |
| RWNN-Llama3-8b (Ours) | ◯ | 71.06% |
| RWNN-Llama3-8b (Ours)† | ◯ | 73.17% |
| RWNN-Llama3-70b (Ours)† | ◯ | 74.85% |
| RWNN-Llama3-8b (Ours), fine-tuned | ✕ | **74.95%** |

Table 7: Transductive classification test accuracy on homophilic (Cora, Citeseer) and heterophilic (Amazon Ratings) datasets. The baselines scores are from Sato (2024); Zhao et al. (2023); Liu et al. (2024); Chen et al. (2024a;b); Platonov et al. (2023b). † denotes using validation labels as in Table 4.

| Method | Training-free? | Cora 20-shot | Cora | Citeseer | Amazon Ratings |
|---|---|---|---|---|---|
| Training-free GNN | ◯ | 60.00% | - | - | - |
| GCN | ✕ | 81.40% | 89.13% | 74.92% | 48.70% |
| GAT | ✕ | 80.80% | 89.68% | 75.39% | 49.09% |
| GraphText | ◯ | 68.30% | 67.77% | 68.98% | - |
| LLaGA | ◯ | - | 59.59% | - | - |
| GraphText | ✕ | - | 87.11% | 74.77% | - |
| LLaGA | ✕ | - | 88.86% | - | 28.20% |
| OFA | ✕ | 75.90% | - | - | 51.44% |
| RWNN-Llama3-8b (Ours) | ◯ | 72.29% | 85.42% | 79.62% | 42.71% |
| RWNN-Llama3-8b (Ours)† | ◯ | 79.84% | 87.08% | 80.41% | 38.66% |
| RWNN-Llama3-8b (Ours), fine-tuned | ✕ | 79.45% | 86.72% | 80.72% | 55.76% |

## A.5 SUPPLEMENTARY MATERIALS FOR GRAPH ISOMORPHISM LEARNING (SECTION 5.2)

**The impact of language pre-training** In Section 5.2, we have used language pre-trained DeBERTa for graph isomorphism learning, even though the records of random walks do not resemble natural language (Appendix A.2 and A.3). To test if language pre-training is useful, we additionally trained RWNN-DeBERTa on CSL and SR16 datasets using the same configurations to our reported models but from random initialization. The training curves are shown in Figure 14. We find that language pre-training has significant benefits over random initialization for isomorphism learning tasks, as randomly initialized models converge very slowly for CSL, and fails to converge for SR16.

We conjecture that certain representations or computational circuits acquired in language pre-training are useful for processing text records of walks to recognize graphs. For example, some circuits specialized in recognizing certain syntactic structures in natural language could be repurposed during fine-tuning to detect skip links of CSL graphs from text records of walks. Our results are also consistent with prior observations on cross-model transfer of pre-trained language models to different domains such as image recognition (Lu et al., 2022; Rothermel et al., 2021).

Table 8: Peptides-func graph classification. The baseline scores are from Tönshoff et al. (2024).

| Method | Test AP |
|---|---|
| GCN | 0.6860 |
| GINE | 0.6621 |
| GatedGCN | 0.6765 |
| Transformer | 0.6326 |
| SAN | 0.6439 |
| CRaWl | 0.7074 |
| RWNN-DeBERTa (Ours) | **0.7123 ± 0.0016** |

```
Record  1[O,sp2]=π2[C,d:3,sp2,r]-3[C,cw,d:4,H:1,sp3,r]-4[N,d:3,H:1,sp2,r]-
        π5[C,d:3,sp2,r]=π6[O,sp2]=π5-π4-3-2=π1=π2-3-7[C,d:4,H:2,sp3,r]-8[S,d:2,sp3,r]-
        9[S,d:2,sp3,r]-10[C,d:4,H:2,sp3,r]-11[C,ccw,d:4,H:1,sp3,r]-12[N,d:3,H:1,sp2,r]-
        π13[C,d:3,sp2,r]=π14[O,sp2]=π13-π12-11-15[C,d:3,sp2,r]-π16[N,d:3,H:1,sp2,r]-
        17[C,cw,d:4,H:1,sp3,r]-18[C,d:3,sp2,r]=π19[O,sp2]=π18-17-20[C,d:4,H:2,sp3]-
        21[C,d:4,H:2,sp3]-22[C,d:3,sp2]=π23[O,sp2]=π22-π24[O,d:2,H:1,sp2]-π22=π23=
        π22-π24-π22=π23=π22-21-20-17-16-π15=π25[O,sp2]=π15-π16-17-20-21-22-π24-π22=
        π23=π22-21-20-17-16-π15=π25=π15-π16-17-20-21-22-π24-π22=π23=π22-π24-π22-21-
        20-17-18=π19=π18-π26[N,d:3,H:1,sp2,r]-27[C,ccw,d:4,H:1,sp3,r]-28[C,d:3,sp2]=
        π29[O,sp2]=π28-π30[N,d:3,H:1,sp2]-31[C,ccw,d:4,H:1,sp3]-32[C,d:4,H:1,sp3]-
        33[C,d:4,H:3,sp3]-32-34[C,d:4,H:3,sp3]-32-33-32-34-32-33-34-32-33-32-31-30-
        π28=π29=π28-π30-31-32-34-32-31-30-π28=π29=π28-π30-31-32-33-32-34-32-33-32-34-
        32-31-30-π28=π29=π28-π30-31-35[C,d:3,sp2]=36[O,sp2]=π35-π37[N,d:3,H:1,sp2]-
        38[C,cw,d:4,H:1,sp3]-39[C,d:3,sp2]-π40[N,d:3,H:1,sp2]-41[C,cw,d:4,H:1,sp3]-
        42[C,d:4,H:2,sp3]-43[C,d:3,sp2,*,r]*π44[C,d:3,H:1,sp2,*,r]*π45[N,d:3,H:1,sp2,*
        ,r]*π46[C,d:3,H:1,sp2,*,r]*π47[N,d:2,sp2,*,r]#*π43*π43*π44*π45*π46*π47*π43-42-41-
        40-π39=π48[O,sp2]=π39-38-37-π35=π36=π35-π37-38-49[C,d:4,H:2,sp3]-50[C
```

Figure 15: An example text record for Peptides-func graph classification.

## A.6 SUPPLEMENTARY MATERIALS FOR TRANSDUCTIVE CLASSIFICATION (SECTION 5.3)

**Fine-tuning** In Section 5.3, our RWNNs are training-free. We demonstrate fine-tuning the Llama 3 8b reader NN using the training labels with quantized low-rank adaptation (QLoRA) (Hu et al., 2022; Dettmers et al., 2023). The result is in Table 6, showing a promising performance.

**Additional real-world transductive classification** We provide additional experiments on real-world transductive classification datasets Cora, Citeseer, and Amazon Ratings. Our RWNNs are designed similarly to the ones used for arXiv. We compile the baseline scores from Sato (2024); Zhao et al. (2023); Liu et al. (2024); Chen et al. (2024a;b); Platonov et al. (2023b), which includes MPNNs and language models on graphs. The results are in Table 7. Llama 3 based RWNN is competitive among training-free methods, and performs reasonably well when fine-tuned with QLoRA. Especially, on Amazon Ratings which is heterophilic (Platonov et al., 2023b), our fine-tuned RWNN-Llama3-8b achieves the highest accuracy (55.76%). This supports our results in Section 3.2 that RWNNs avoid over-smoothing, as avoiding it is known to be important for handling heterophily (Yan et al., 2022).

## A.7 SUPPLEMENTARY TASKS

We provide supplementary experimental results on real-world graph classification, substructure counting, and link prediction tasks we could not include in the main text due to space constraints.

**Real-world graph classification** We conduct a preliminary demonstration of RWNN on real-world graph classification, using the Peptides-func dataset (Dwivedi et al., 2022) used in Tönshoff et al. (2023). Our model is a pre-trained DeBERTa-base (identical to Section 5.2) fine-tuned on text records of non-backtracking MDLR walks with anonymization and named neighbor recording. The recording function is designed to properly incorporate the vertex and edge attributes provided in the dataset, including the atom and bond types. An example text is provided in Figure 15.

In Table 8, we report the test average precision (AP) at best validation accuracy, with 40 random predicted logits averaged at test time. We report the mean and standard deviation for five repeated tests. The baseline scores, including CRaWl, are from Tönshoff et al. (2023); Tönshoff et al. (2024). We see that RWNN-DeBERTa, which has been successful in graph isomorphism learning (Section 5.2), also shows a promising result in real-world protein graph classification, despite its simplicity of recording random walks in plain text and processing them with a fine-tuned language model.

Table 9: Substructure (8-cycle) counting. We report normalized test mean absolute error (MAE) at best validation. We trained MP, Subgraph GNN, Local 2-GNN, and Local 2-FGNN using configurations in Zhang et al. (2024), and trained MP-RNI, MP-Dropout, AgentNet, and CRaWl as in Table 3.

| Method | Graph-level | Vertex-level |
|---|---|---|
| MP | 0.0917 | 0.1156 |
| Subgraph GNN | 0.0515 | 0.0715 |
| Local 2-GNN | 0.0433 | 0.0478 |
| Local 2-FGNN | 0.0422 | 0.0438 |
| Local 2-FGNN (large) | 0.0932 | 0.1121 |
| MP-RNI | 0.3610 | 0.3650 |
| MP-Dropout | 0.1359 | 0.1393 |
| AgentNet | 0.1107 | N/A |
| CRaWl | 0.0526 | 0.0725 |
| RWNN-DeBERTa (Ours) | 0.0459 | 0.0465 |

Table 10: Real-world link prediction. The baseline scores are from Table 5 of Cai et al. (2021) and we reproduced LGLP using the official code. We report the aggregated test AP for 3 randomized runs.

| Method | YST | KHN | ADV |
|---|---|---|---|
| Katz | 81.63 ± 0.41 | 83.04 ± 0.38 | 91.76 ± 0.15 |
| PR | 82.08 ± 0.46 | 87.18 ± 0.26 | 92.43 ± 0.17 |
| SR | 76.02 ± 0.49 | 75.87 ± 0.66 | 83.22 ± 0.20 |
| node2vec | 76.61 ± 0.94 | 80.60 ± 0.74 | 76.70 ± 0.82 |
| SEAL | 86.45 ± 0.25 | 90.37 ± 0.16 | 93.52 ± 0.13 |
| LGLP | 88.54 ± 0.77 | 90.80 ± 0.33 | 93.51 ± 0.32 |
| RWNN-transformer (Ours) | 88.13 ± 0.45 | 90.90 ± 0.35 | 93.70 ± 0.21 |

**Substructure counting**  To supplement Section 5.2, we demonstrate RWNNs on challenging tasks that require both expressive power and generalization to unseen graphs. We use a synthetic dataset of 5,000 random regular graphs from Chen et al. (2020) and consider the task of counting substructures. We choose 8-cycles since they require a particularly high expressive power (Zhang et al., 2024). We test both graph- and vertex-level counting (i.e., counting 8-cycles containing the queried vertex) following Zhang et al. (2024). We use the data splits from Chen et al. (2020); Zhao et al. (2022b); Zhang et al. (2024), while excluding disconnected graphs following our assumption in Section 2.1. Our RWNN uses the same design to Section 5.2, and we train them with L1 loss for ≤250k steps using batch size of 128 graphs for graph-level counting and 8 graphs for vertex-level. At test-time, we ensemble 64 and 32 predictions by averaging for graph- and vertex-level tasks, respectively.

The results are in Table 9, and overall in line with Section 5.2. While MPNNs show a low performance, variants with higher expressive powers such as local 2-GNN and local 2-FGNN achieve high performances. This is consistent with the observation of Zhang et al. (2024). On the other hand, MPNNs with stochastic symmetry-breaking often show learning difficulties and does not achieve good performance. AgentNet was not directly applicable for vertex-level counting since its vertex encoding depends on which vertices the agent visits, and a learning difficulty was observed for graph-level counting. Our approach based on DeBERTa demonstrates competitive performance, outperforming random walk baselines and approaching performances of local 2-GNN and 2-FGNN.

**Link prediction**  We demonstrate an application of RWNNs to real-world link prediction, based on Cai et al. (2021) where link prediction is cast as vertex classification on line graphs. We follow the experimental setup of Cai et al. (2021) and only change their 3-layer 32-dim DGCNN with an RWNN that uses a 3-layer 32-dim transformer encoder. As the task is vertex-level, we use periodic restarts with $k = 5$. The results in Table 10 shows that RWNN performs reasonably well.

## A.8    SUPPLEMENTARY ANALYSIS (SECTION 5 AND APPENDIX A.7)

We provide supplementary analysis and discussion on the impact of cover times, test-time ensembling, and the effective walk lengths, which we could not include in the main text due to space constraints.

Table 11: The performance of RWNN-DeBERTa on SR16 graph separation (Table 3, Section 5.2) for different required cover times controlled by the use of named neighbor recording.

|  | Test accuracy | Cover time |
|---|---|---|
| w/o named neighbor recording | 50% | $C_E(G) = 490.00$ |
| w/ named neighbor recording | **100%** | $C_V(G) = \mathbf{48.25}$ |

Table 12: The performance of RWNN-Llama3-70b† on arXiv transductive classification (Table 4, Section 5.3) for different local cover times controlled by the restart probability $\alpha$ of the random walk.

|  | Test accuracy | Cover time $C_V(B_{r=1}(v))$ | Unique vertices observed per walk |
|---|---|---|---|
| $\alpha = 0.3$ | 74.40% | 161.2 | **45.28** |
| $\alpha = 0.7$ | **74.85%** | **96.84** | 31.13 |

**The impact of cover times of walks**    In Section 2, we use cover times as a key tool for the analysis and comparison of different random walk algorithms. In the analysis, cover times provide the worst-case upper bounds of walk length required to achieve universality, which is in line with the literature on the cover times of Markov chains (Feige, 1995). To demonstrate how the differences in cover times may translate to different model behaviors, we provide two examples where performing interventions that change cover times leads to substantial differences in task performances.

First, in SR16 graph separation (Section 5.2), we test the use of named neighbor recording, which changes the required cover time from vertex cover time $C_V(G)$ (when used) to edge cover time $C_E(G)$ (when not used). The result is in Table 11. We can see that not using named neighbor recording has a drastic effect on the required cover time (vertex cover time 48.25 → edge cover time[6] 490.00), and the test performance deteriorates from perfect to random accuracy (100% → 50%).

Second, in arXiv transductive classification (Section 5.3), due to the high homophily (Platonov et al., 2023a) we can assume that the cover time of the local neighborhoods, i.e., $B_r(v)$ for small $r$, has an important influence on performance. Thus, we set $r = 1$ and check if the walk strategy that lowers the cover time of $B_r(v)$ leads to a better performance. To test different local cover times, we use two choices of the restart probability $\alpha = 0.3$ and 0.7. Since we use named neighbor recording, we measure vertex cover times. Due to the large size of $G$, we estimate the cover times by fixing a random set of 100 test vertices and running 100 walks from each of them. The result is in Table 12. Increasing the restart probability (0.3 → 0.7) simultaneously results in a lower cover time (161.2 → 96.84) and a higher test accuracy (74.40% → 74.85%), even though the average number of observed vertices decreases (45.28 → 31.13). This implies that lower cover time is correlated with better performance, while the number of observed vertices itself is not directly correlated with better performance.

While the above shows cases where interventions in cover times lead to different task performances, in general, there are factors that make it not always trivial to equate task performances and cover times. This is because a substantial amount of useful information in practice could be recovered from possibly non-covering walks. We discuss this aspect in depth in Appendix A.11.

**The impact of test-time ensembling**    For each test input to an RWNN $X_\theta$, we often sample several randomized predictions $\hat{\mathbf{y}} \sim X_\theta(\cdot)$ and ensemble them by averaging (or voting, for Llama 3 models where we cannot directly obtain continuous logits). As discussed in Section 2, this can be viewed as MC estimation of the underlying invariant mean predictor $(\cdot) \mapsto \mathbb{E}[X_\theta(\cdot)]$ (or the invariant limiting classifier in the case of voting (Cotta et al., 2023)). From another perspective, we can also interpret this as a type of test-time scaling with repeated sampling, which has been shown beneficial in language models (Brown et al., 2024) and randomized algorithms (called amplification (Cotta et al., 2023; Sieradzki et al., 2022)). To see if similar perspectives can be held for RWNNs, we measure the model performances with increased numbers of samples for averaging or voting. The results are in Table 13. We see that increasing sample size consistently leads to better performances and reduced variances overall. The point of plateau seems to differ per application and model architecture, and we leave obtaining a better understanding of the scaling behavior for future work.

---

[6]As in Section 5.1, while the strict definition of edge cover requires traversing each edge in both directions, our measurements only require traversing one direction, which suffices for the universality of RWNNs.

Table 13: Test performances of RWNN-Llama3-70b† (Table 4, Section 5.3) and RWNN-DeBERTa (Table 8, Appendix A.7) for different numbers of randomized predictions for ensembling by voting and averaging, respectively. For *, we were unable to run repeated experiments due to sampling costs.

| # samples | arXiv test accuracy |
|---|---|
| 1 | 74.45 ± 0.049% |
| 3 | 74.75 ± 0.035% |
| 5 | 74.83 ± 0.028% |
| 7 | 74.86% ± N/A* |
| 10 | **74.87% ± N/A*** |

| # samples | Peptides-func test AP |
|---|---|
| 1 | 0.5300 ± 0.0092 |
| 10 | 0.6937 ± 0.0039 |
| 20 | 0.7056 ± 0.0026 |
| 40 | 0.7123 ± 0.0016 |
| 60 | 0.7145 ± 0.0015 |
| 80 | 0.7155 ± 0.0014 |
| 100 | 0.7161 ± 0.0009 |
| 160 | 0.7176 ± 0.0013 |
| 320 | **0.7177 ± N/A*** |

Table 14: Effective random walk lengths for the experiments in Sections 5.2 and 5.3.

| Dataset | Length |
|---|---|
| CSL (Section 5.2) | 214.03 ± 2.48 |
| SR16 (Section 5.2) | 222.00 ± 0.00 |
| SR25 (Section 5.2) | 129.54 ± 2.26 |
| arXiv (Section 5.3) | 194.52 ± 10.59 |

**Effective walk lengths**  For Sections 5.2 and 5.3, the effective lengths of walks are affected by the language model tokenizers since text records exceeding the maximum number of tokens are truncated. The maximum number of tokens is a hyperparameter which we determine from memory constraints; for the DeBERTa tokenizer used in Section 5.2, we set it to 512, and for the Llama 3 tokenizer used in Section 5.3, we set it to 4,096. The effective lengths of random walks for each dataset are measured and provided in Table 14. The effective length is shorter (129.54 on average) for SR25 graphs compared to CSL and SR16 (214.03 and 222.00 on average, respectively) although using the same tokenizer configurations, which is because the higher average degrees of SR25 graphs (Figure 6) compared to CSL and SR16 graphs (Figures 4 and 5, respectively) has led to a higher number of text characters invested in recording named neighbors ("#" symbols in the figures).

## A.9 PROOFS

We recall that an isomorphism between two graphs $G$ and $H$ is a bijection $\pi : V(G) \to V(H)$ such that any two vertices $u$ and $v$ are adjacent in $G$ if and only if $\pi(u)$ and $\pi(v)$ are adjacent in $H$. If an isomorphism $\pi$ exists between $G$ and $H$, the graphs are isomorphic and written $G \simeq H$ or $G \overset{\pi}{\simeq} H$. By $X \overset{d}{=} Y$ we denote the equality of two random variables $X$ and $Y$ in distributions.

In the proofs, we write by $\mathrm{id}(\cdot)$ the namespace obtained by anonymization of a walk $v_0 \to \cdots \to v_l$ (Section 2) that maps each vertex $v_t$ to its unique integer name $\mathrm{id}(v_t)$ starting from $\mathrm{id}(v_0) = 1$.

Let us define some notations related to cover times. These will be used from Appendix A.9.5. We denote by $H(u, v)$ the expected number of steps a random walk starting from $u$ takes until reaching $v$. This is called the hitting time between $u$ and $v$. For a graph $G$, let $C_V(G, u)$ be the expected number of steps a random walk starting from $u$ takes until visiting every vertices of $G$. The vertex cover time $C_V(G)$, or the cover time, is this quantity given by the worst possible $u$:

$$C_V(G) := \max_{u \in V(G)} C_V(G, u). \tag{16}$$

Likewise, let $C_E(G, u)$ be the expected number of steps a random walk starting from $u$ takes until traversing every edge of $G$. The edge cover time $C_E(G)$ is given by the worst possible $u$:

$$C_E(G) := \max_{u \in V(G)} C_E(G, u). \tag{17}$$

For local balls $B_r(u)$, we always set the starting vertex of the random walk to $u$ (Section 2.2). Thus, we define their cover times as follows:

$$C_V(B_r(u)) := C_V(B_r(u), u), \tag{18}$$
$$C_E(B_r(u)) := C_E(B_r(u), u). \tag{19}$$

### A.9.1 PROOF OF PROPOSITION 2.1 (SECTION 2)

**Proposition 2.1.** $X_\theta(\cdot)$ *is invariant in probability, if its random walk algorithm is invariant in probability and its recording function is invariant.*

*Proof.* We recall the probabilistic invariance of random walk algorithm in Equation 3:

$$\pi(v_0) \to \cdots \to \pi(v_l) \overset{d}{=} u_0 \to \cdots \to u_l, \quad \forall G \overset{\pi}{\simeq} H, \tag{20}$$

where $v_{[\cdot]}$ is a random walk on $G$ and $u_{[\cdot]}$ is a random walk on $H$. We further recall the invariance of recording function $q : (v_0 \to \cdots \to v_l, G) \mapsto \mathbf{z}$ in Equation 7:

$$q(v_0 \to \cdots \to v_l, G) = q(\pi(v_0) \to \cdots \to \pi(v_l), H), \quad \forall G \overset{\pi}{\simeq} H, \tag{21}$$

for any given random walk $v_{[\cdot]}$ on $G$. Combining Equation 20 and equation 21, we have the following:

$$q(v_0 \to \cdots \to v_l, G) \overset{d}{=} q(u_0 \to \cdots \to u_l, H), \quad \forall G \simeq H. \tag{22}$$

Then, since the reader neural network $f_\theta$ is a deterministic map, we have:

$$f_\theta(q(v_0 \to \cdots \to v_l, G)) \overset{d}{=} f_\theta(q(u_0 \to \cdots \to u_l, H)), \quad \forall G \simeq H, \tag{23}$$

which leads to:

$$X_\theta(G) \overset{d}{=} X_\theta(H), \quad \forall G \simeq H. \tag{24}$$

This shows Equation 2 and completes the proof. □

### A.9.2 PROOF OF PROPOSITION 2.2 (SECTION 2)

We first show a useful lemma.

**Lemma A.2.** *The random walk in Equation 4 is invariant in probability, if the probability distributions of the starting vertex $v_0$ and each transition $v_{t-1} \to v_t$ are invariant.*

*Proof.* We prove by induction. Let us write probabilistic invariance of the starting vertex $v_0$ as:

$$\pi(v_0) \stackrel{d}{=} u_0, \quad \forall G \stackrel{\pi}{\simeq} H, \tag{25}$$

and probabilistic invariance of each transition $v_{t-1} \to v_t$ as follows, by fixing $v_{t-1}$ and $u_{t-1}$:

$$\pi(v_{t-1}) \to \pi(v_t) \stackrel{d}{=} u_{t-1} \to u_t, \quad \forall G \stackrel{\pi}{\simeq} H, v_{t-1} \in V(G), u_{t-1} := \pi(v_{t-1}). \tag{26}$$

Let us assume the following for some $t \geq 1$:

$$\pi(v_0) \to \cdots \to \pi(v_{t-1}) \stackrel{d}{=} u_0 \to \cdots \to u_{t-1}, \quad G \stackrel{\pi}{\simeq} H. \tag{27}$$

Then, from the chain rule of joint distributions, the first-order Markov property of random walk in Equation 4, and probabilistic invariance in Equation 26, we obtain the following:

$$\pi(v_0) \to \cdots \to \pi(v_t) \stackrel{d}{=} u_0 \to \cdots \to u_t, \quad G \stackrel{\pi}{\simeq} H. \tag{28}$$

By induction from the initial condition $\pi(v_0) \stackrel{d}{=} u_0$ in Equation 25, the following holds $\forall l > 0$:

$$\pi(v_0) \to \cdots \to \pi(v_l) \stackrel{d}{=} u_0 \to \cdots \to u_l, \quad \forall G \stackrel{\pi}{\simeq} H. \tag{29}$$

This shows Equation 3 and completes the proof. $\qquad \square$

We now prove Proposition 2.2.

**Proposition 2.2.** *The random walk in Equation 4 is invariant in probability if its conductance $c_{[\cdot]}(\cdot)$ is invariant:*

$$c_G(u, v) = c_H(\pi(u), \pi(v)), \quad \forall G \stackrel{\pi}{\simeq} H. \tag{30}$$

*It includes constant conductance, and any choice that only uses degrees of endpoints $\deg(u)$, $\deg(v)$.*

*Proof.* We recall Equation 4 for a given conductance function $c_G : E(G) \to \mathbb{R}_+$:

$$\mathrm{Prob}[v_t = x | v_{t-1} = u] := \frac{c_G(u, x)}{\sum_{y \in N(u)} c_G(u, y)}. \tag{31}$$

From Lemma A.2, it is sufficient to show the probabilistic invariance of each transition $v_{t-1} \to v_t$ as we sample $v_0$ from invariant distribution $\mathrm{Uniform}(V(G))$ (Algorithm 1). We rewrite Equation 26 as:

$$\mathrm{Prob}[v_t = x | v_{t-1} = u] = \mathrm{Prob}[u_t = \pi(x) | u_{t-1} = \pi(u)], \quad \forall G \stackrel{\pi}{\simeq} H. \tag{32}$$

If the conductance function $c_{[\cdot]}(\cdot)$ is invariant (Equation 30), we can show Equation 32 by:

$$
\begin{aligned}
\mathrm{Prob}[v_t = x | v_{t-1} = u] &:= \frac{c_G(u, x)}{\sum_{y \in N(u)} c_G(u, y)}, \\
&= \frac{c_H(\pi(u), \pi(x))}{\sum_{y \in N(u)} c_H(\pi(u), \pi(y))}, \\
&= \frac{c_H(\pi(u), \pi(x))}{\sum_{\pi(y) \in N(\pi(u))} c_H(\pi(u), \pi(y))}, \\
&= \mathrm{Prob}[u_t = \pi(x) | u_{t-1} = \pi(u)], \quad \forall G \stackrel{\pi}{\simeq} H. \tag{33}
\end{aligned}
$$

In the third equality, we used the fact that isomorphism $\pi(\cdot)$ preserves adjacency. It is clear that any conductance function $c_{[\cdot]}(\cdot)$ with a constant conductance such as $c_G(\cdot) = 1$ is invariant. Furthermore, any conductance function that only uses degrees of endpoints $\deg(u)$, $\deg(v)$ is invariant as isomorphism $\pi(\cdot)$ preserves the degree of each vertex. This completes the proof. $\qquad \square$

### A.9.3 Proof of Proposition A.1 (Appendix A.1)

We extend the proof of Proposition 2.2 given in Appendix A.9.2 to second-order random walks discussed in Appendix A.1. We first show a useful lemma:

**Lemma A.3.** *A second-order random walk algorithm is invariant in probability, if the probability distributions of the starting vertex $v_0$, the first transition $v_0 \to v_1$, and each transition $(v_{t-2}, v_{t-1}) \to v_t$ for $t > 1$ are invariant.*

*Proof.* We prove by induction. The probabilistic invariance of starting vertex $v_0$ and first transition $v_0 \to v_1$ are:

$$\pi(v_0) \stackrel{d}{=} u_0, \quad \forall G \stackrel{\pi}{\simeq} H, \tag{34}$$

$$\pi(v_0) \to \pi(v_1) \stackrel{d}{=} u_0 \to u_1, \quad \forall G \stackrel{\pi}{\simeq} H, v_0 \in V(G), u_0 := \pi(v_0), \tag{35}$$

and probabilistic invariance of each transition $(v_{t-2}, v_{t-1}) \to v_t$ for $t > 1$ can be written as follows by fixing $(v_{t-2}, v_{t-1})$ and $(u_{t-2}, u_{t-1})$:

$$\pi(v_{t-2}) \to \pi(v_{t-1}) \to \pi(v_t) \stackrel{d}{=} u_{t-2} \to u_{t-1} \to u_t, \quad \forall G \stackrel{\pi}{\simeq} H,$$
$$(v_{t-2}, v_{t-1}) \in E(G),$$
$$(u_{t-2}, u_{t-1}) := (\pi(v_{t-2}), \pi(v_{t-1})). \tag{36}$$

Let us assume the following for some $t \geq 2$:

$$\pi(v_0) \to \cdots \to \pi(v_{t-2}) \to \pi(v_{t-1}) \stackrel{d}{=} u_0 \to \cdots \to u_{t-2} \to u_{t-1}, \quad G \stackrel{\pi}{\simeq} H. \tag{37}$$

Then, from the chain rule of joint distributions, the second-order Markov property of second-order random walks, and probabilistic invariance in Equation 36, we obtain the following:

$$\pi(v_0) \to \cdots \to \pi(v_{t-1}) \to \pi(v_t) \stackrel{d}{=} u_0 \to \cdots \to u_{t-1} \to u_t, \quad G \stackrel{\pi}{\simeq} H. \tag{38}$$

By induction from the initial condition $\pi(v_0) \to \pi(v_1) \stackrel{d}{=} u_0 \to u_1$ given from Equations 34 and 35, the following holds $\forall l > 0$:

$$\pi(v_0) \to \cdots \to \pi(v_l) \stackrel{d}{=} u_0 \to \cdots \to u_l, \quad \forall G \stackrel{\pi}{\simeq} H. \tag{39}$$

This shows Equation 3 and completes the proof. $\qquad\square$

We now prove Proposition A.1.

**Proposition A.1.** *The non-backtracking random walk in Equation 13 and the node2vec random walk in Equation 14 are invariant in probability, if their underlying first-order random walk algorithm is invariant in probability.*

*Proof.* We assume that the first transition $v_0 \to v_1$ is given by the underlying first-order random walk algorithm. This is true for non-backtracking since there is no $v_{t-2}$ to avoid, and true for the official implementation of node2vec (Grover & Leskovec, 2016). Then from Lemma A.3, it is sufficient to show probabilistic invariance of each transition $(v_{t-2}, v_{t-1}) \to v_t$ for $t > 1$ as we sample $v_0$ from the invariant distribution $\text{Uniform}(V(G))$ (Algorithm 1) and the first transition $v_0 \to v_1$ is given by the first-order random walk which is assumed to be invariant in probability. Rewrite Equation 36 as:

$$\text{Prob}[v_t = x | v_{t-1} = j, v_{t-2} = i] = \text{Prob}[u_t = \pi(x) | u_{t-1} = \pi(j), u_{t-2} = \pi(i)], \quad \forall G \stackrel{\pi}{\simeq} H. \tag{40}$$

For non-backtracking, we first handle the case where $i \to j$ reaches a dangling vertex $j$ which has $i$ as its only neighbor. In this case the walk begrudgingly backtracks $i \to j \to i$ (Appendix A.1). Since isomorphism $\pi(\cdot)$ preserves adjacency, $\pi(i) \to \pi(j)$ also reaches a dangling vertex $\pi(j)$ and must begrudgingly backtrack $\pi(i) \to \pi(j) \to \pi(i)$. By interpreting the distributions on $v_t$ and $u_t$ as one-hot at $i$ and $\pi(i)$, respectively, we can see that Equation 40 holds. If $i \to j$ does not reach a

dangling vertex, the walk $j \to x$ follows the invariant probability of the underlying first-order random walk $\text{Prob}[v_{t+1} = x|v_t = j]$ renormalized over $N(j) \setminus \{i\}$. Then we can show Equation 40 by:

$$\text{Prob}[v_t = x|v_{t-1} = j, v_{t-2} = i]$$

$$:= \begin{cases} \dfrac{\text{Prob}[v_t = x|v_{t-1} = j]}{\sum_{y \in N(j) \setminus \{i\}} \text{Prob}[v_t = y|v_{t-1} = j]} & \text{for } x \neq i, \\ \\ 0 & \text{for } x = i, \end{cases}$$

$$= \begin{cases} \dfrac{\text{Prob}[u_t = \pi(x)|u_{t-1} = \pi(j)]}{\sum_{\pi(y) \in N(\pi(j)) \setminus \{\pi(i)\}} \text{Prob}[u_t = \pi(y)|u_{t-1} = \pi(j)]} & \text{for } \pi(x) \neq \pi(i), \\ \\ 0 & \text{for } \pi(x) = \pi(i), \end{cases}$$

$$= \text{Prob}[u_t = \pi(x)|u_{t-1} = \pi(j), u_{t-2} = \pi(i)], \quad \forall G \overset{\pi}{\simeq} H. \tag{41}$$

In the second equality, we have used the fact that isomorphism $\pi(\cdot)$ is a bijection and preserves adjacency, and the assumption that the first-order random walk algorithm is invariant in probability. For node2vec walk, we first show the invariance of the weighting term $\alpha(i, x)$ in Equation 15 using the fact that isomorphism preserves shortest path distances between vertices $d(i, x) = d(\pi(i), \pi(x))$:

$$\alpha(i, x) := \begin{cases} 1/p & \text{for } d(i, x) = 0, \\ 1 & \text{for } d(i, x) = 1, \\ 1/q & \text{for } d(i, x) = 2. \end{cases}$$
$$= \begin{cases} 1/p & \text{for } d(\pi(i), \pi(x)) = 0, \\ 1 & \text{for } d(\pi(i), \pi(x)) = 1, \\ 1/q & \text{for } d(\pi(i), \pi(x)) = 2. \end{cases}$$
$$= \alpha(\pi(i), \pi(x)). \tag{42}$$

Then we can show Equation 40 by:

$$\text{Prob}[v_t = x|v_{t-1} = j, v_{t-2} = i] := \frac{\alpha(i, x) \text{Prob}[v_t = x|v_{t-1} = j]}{\sum_{y \in N(j)} \alpha(i, y) \text{Prob}[v_t = y|v_{t-1} = j]},$$

$$= \frac{\alpha(\pi(i), \pi(x)) \text{Prob}[u_t = \pi(x)|u_{t-1} = \pi(j)]}{\sum_{\pi(y) \in N(\pi(j))} \alpha(\pi(i), \pi(y)) \text{Prob}[u_t = \pi(y)|u_{t-1} = \pi(j)]},$$

$$= \text{Prob}[u_t = \pi(x)|u_{t-1} = \pi(j), u_{t-2} = \pi(i)], \quad \forall G \overset{\pi}{\simeq} H. \tag{43}$$

In the second equality, we have used the fact that isomorphism $\pi(\cdot)$ preserves adjacency, and the assumption that the first-order walk algorithm is invariant in probability. This completes the proof. $\square$

### A.9.4 Proof of Proposition 2.3 (Section 2)

**Proposition 2.3.** *A recording function $q : (v_0 \to \cdots \to v_l, G) \mapsto \mathbf{z}$ that uses anonymization, optionally with named neighbors, is invariant.*

*Proof.* Given a walk $v_0 \to \cdots \to v_l$ on $G$, we can write the anonymized namespace $\text{id}(\cdot)$ as follows:

$$\text{id}(v_t) = 1 + \arg\min_i [v_i = v_t]. \tag{44}$$

Consider any walk $\pi(v_0) \to \cdots \to \pi(v_l)$ on some $H$ which is isomorphic to $G$ via $\pi$. Let us denote the anonymization of this walk as $\text{id}'(\cdot)$. Then we have:

$$\text{id}'(\pi(v_t)) = 1 + \arg\min_i [\pi(v_i) = \pi(v_t)],$$

$$= 1 + \arg\min_i [v_i = v_t],$$

$$= \text{id}(v_t), \quad \forall G \overset{\pi}{\simeq} H. \tag{45}$$

The second equality is due to the fact that $\pi$ is a bijection. This completes the proof for anonymization. For the named neighbors, we prove by induction. Let us fix some $t \geq 1$ and assume:

$$q(v_0 \to \cdots \to v_{t-1}, G) = q(\pi(v_0) \to \cdots \to \pi(v_{t-1}), H), \quad \forall G \overset{\pi}{\simeq} H. \tag{46}$$

Let $T \subseteq E(G)$ be the set of edges recorded by $\mathbf{z} := q(v_0 \to \cdots \to v_{t-1}, G)$, and $T' \subseteq E(H)$ be the set of edges recorded by $\mathbf{z}' := q(\pi(v_0) \to \cdots \to \pi(v_{t-1}), H)$. Then, $T$ and $T'$ are isomorphic via $\pi$:

$$T' = \{(\pi(u), \pi(v)) \forall (u, v) \in T\}. \tag{47}$$

The equality is shown as follows. ($\supseteq$) Any $(u, v) \in T$ is recorded in $\mathbf{z}$ as $(\mathrm{id}(u), \mathrm{id}(v))$. From Equations 46 and 45, we have $\mathbf{z} = \mathbf{z}'$ and $(\mathrm{id}(u), \mathrm{id}(v)) = (\mathrm{id}(\pi(u)), \mathrm{id}(\pi(v)))$. Thus, $(\pi(u), \pi(v))$ is recorded in $\mathbf{z}'$ as $(\mathrm{id}(\pi(u)), \mathrm{id}(\pi(v)))$, i.e., $(\pi(u), \pi(v)) \in T'$. ($\subseteq$) This follows from symmetry. Now, we choose some $v_t \in N(v_{t-1})$ and consider the set $F \subseteq E(G)$ of all unrecorded edges from $v_t$:

$$F := \{(v_t, u) : u \in N(v_t)\} \setminus T. \tag{48}$$

Then, the set $D$ of unrecorded neighbors of $v_t$ is given as:

$$D := \{u : (v_t, u) \in F\}. \tag{49}$$

Since $\pi(\cdot)$ preserves adjacency, the set $F' \subseteq E(H)$ of all unrecorded edges from $\pi(v_t)$ is given by:

$$\begin{aligned} F' &:= \{(\pi(v_t), \pi(u)) : \pi(u) \in N(\pi(v_t))\} \setminus T', \\ &= \{(\pi(v_t), \pi(u)) : u \in N(v_t)\} \setminus \{(\pi(u'), \pi(v')) \forall (u', v') \in T\}, \\ &= \{(\pi(v_t), \pi(u)) : u \in N(v_t), (v_t, u) \notin T\}, \\ &= \{(\pi(v_t), \pi(u)) : (v_t, u) \in F\}, \end{aligned} \tag{50}$$

and consequently:

$$\begin{aligned} D' &:= \{\pi(u) : (\pi(v_t), \pi(u)) \in F'\}, \\ &= \{\pi(u) : u \in D\}. \end{aligned} \tag{51}$$

While $D$ and $D'$ may contain vertices not yet visited by the walks and hence not named by $\mathrm{id}(\cdot)$, what we record are named neighbors. Let $S$ be the set of all named vertices in $G$. It is clear that the set $S'$ of named vertices in $H$ is given by $S' = \{\pi(v) : v \in S\}$. Then, the named neighbors in $G$ to be newly recorded for $v_{t-1} \to v_t$ is given by $U = D \cap S$, and for $\pi(v_{t-1}) \to \pi(v_t)$ in $H$ the set is given by $U' = D' \cap S'$. Since $\pi(\cdot)$ is a bijection, we can see that $U' = \{\pi(u) : u \in U\}$. From the invariance of anonymization in Equation 45, $U$ and $U'$ are named identically $\{\mathrm{id}(u) : u \in U\} = \{\mathrm{id}(\pi(u)) : \pi(u) \in U'\}$, and therefore will be recorded identically. As a result, the information to be added to the records at time $t$ are identical, and we have:

$$q(v_0 \to \cdots \to v_t, G) = q(\pi(v_0) \to \cdots \to \pi(v_t), H), \quad \forall G \overset{\pi}{\simeq} H. \tag{52}$$

Then, by induction from the initial condition:

$$q(v_0, G) = q(\pi(v_0), H) = \mathrm{id}(v_0) = \mathrm{id}(\pi(v_0)) = 1, \tag{53}$$

the recording function using anonymization and named neighbors is invariant. $\qquad\square$

### A.9.5 Proof of Theorem 2.4 (Section 2.2)

**Theorem 2.4.** *For a uniform random walk on an infinite graph $G$ starting at $v$, the vertex and edge cover times of the finite local ball $B_r(v)$ are not always finitely bounded.*

*Proof.* We revisit the known fact that hitting times on the infinite line $\mathbb{Z}$ is infinite (Dumitriu et al., 2003; Janson & Peres, 2012; carnage, 2012; McNew, 2013). Consider a local ball $B_1(0)$ of radius 1 from the origin 0 such that $V(B_1(0)) = \{-1, 0, 1\}$. Let us assume that the expected number of steps for a random walk starting at 0 to visit 1 for the first time is finite, and denote it by $T$. In the first step of the walk, we have to go either to $-1$ or 1. If we go to $-1$, we have to get back to 0 and then to 1 which would take $2T$ in expectation (by translation symmetry). This leads to $T = 1/2 \cdot 1 + 1/2 \cdot (1 + 2T) = 1 + T$, which is a contradiction. Since visiting all vertices or traversing all edges in in $B_1(0)$ clearly involves visiting 1, the vertex and edge cover times cannot be finitely bounded. $\qquad\square$

A.9.6 PROOF OF THEOREM 2.5 (SECTION 2.2)

Let us recall that our graph $G$ is locally finite (Section 2.2), and denote its maximum degree as $\Delta$. We further denote by $d(u, v)$ the shortest path distance between $u$ and $v$. We show some useful lemmas. We first show that $H(u, x)$, the expected number of steps of a random walk starting at $u$ takes until reaching $x$, is finite for any $u, x$ in $B_r(v)$ for any nonzero restart probability $\alpha$ or restart period $k \geq r$.

**Lemma A.4.** *For any $u$ and $x$ in $B_r(v)$, $H(u, x)$ is bounded by the following:*

$$H(u, x) \leq \frac{1}{\alpha} + \frac{1}{\alpha} \left( \frac{\Delta}{1-\alpha} \right)^r + \frac{1}{\alpha} \left( \frac{1}{\alpha} - 1 \right) \left( \frac{\Delta}{1-\alpha} \right)^{2r}, \tag{54}$$

*if the random walk restarts at $v$ with any probability $\alpha \in (0, 1)$, and:*

$$H(u, x) \leq k + k\Delta^r, \tag{55}$$

*if the random walk restarts at $v$ with any period $k \geq r$.*

*Proof.* We first prove for random restarts. The proof is inspired by Theorem 1.1 of McNew (2013) and Lemma 2 of Zuckerman (1991). We clarify some measure-theoretic details. Let $W$ be the set of all (infinite) random walks on $G$. We denote by $P$ the probability measure defined on the space $W$, equipped with its cylinder $\sigma$-algebra, by the random walk restarting at $v$ with probability $\alpha \in (0, 1)$. For a vertex $x$ in $G$, we define the hitting time function $|\cdot|_x : W \to \mathbb{N} \cup \{\infty\}$ as follows:

$$|w|_x := \arg\min_i [(w)_i = x], \quad \forall w \in W. \tag{56}$$

Let $W(a)$ be the set of all random walks that start at $a$, $W(a, b)$ be the set of random walks that start at $a$ and visit $b$, and $W(a, b^-)$ be the set of random walks that start at $a$ and do not visit $b$. Then, the measurability of $|\cdot|_x$ follows from the measurability of the sets $W(a)$, $W(a, b)$, and $W(a, b^-)$, which we show as follows. $\forall a, b \in V(G)$, $W(a)$ is clearly measurable, $W(a, b)$ is measurable as it equals $\bigcup_{n \in \mathbb{N}} \{w \in W | (w)_1 = a, (w)_n = b\}$, and $W(a, b^-)$ is measurable as it is $W(a) \setminus W(a, b)$.

Consider $W(u, x^-)$, the set of random walks that start at $u$ and never visit $x$. We start by showing that this set is of measure zero:

$$P(W(u, x^-)) = 0. \tag{57}$$

For this, we use the fact that $W(u, x^-) = W(u, v^-, x^-) \cup W(u, v, x^-)$, where $W(u, v^-, x^-)$ is the set of random walks starting at $u$ that do not visit $v$ nor $x$, and $W(u, v, x^-)$ is the set of walks starting at $u$ that visit $v$ and do not visit $x$. We have the following:

$$P(W(u, x^-)) = P(W(u, v^-, x^-)) + P(W(u, v, x^-)),$$
$$\leq P(W(u, v^-)) + P(W(u, v, x^-)). \tag{58}$$

We show Equation 57 by showing $P(W(u, v^-)) = 0$ and $P(W(u, v, x^-)) = 0$ in Equation 58.

1. $(P(W(u, v^-)) = 0)$ Consider $W(u, v^-)$, the set of all random walks that start at $u$ and never visit $v$. Since restart sends a walk to $v$, every walk in this set never restarts. The probability of a walk not restarting until step $t$ is $(1-\alpha)^t$. Denote this probability by $p_t$, and let $p$ be the probability of a random walk to never restart. Then $p_t \downarrow p = 0$ and $P(W(u, v^-)) \leq p = 0$.

2. $(P(W(u, v, x^-)) = 0)$ Assume $P(W(u, v, x^-)) > 0$. Then $P(W(v, x^-)) > 0$ since each walk step is independent. Let $W_N(v, x^-)$ be the set of walks that start at $v$ and do not reach $x$ within $N$ restarts. Then we have $W_N(v, x^-) \downarrow W(v, x^-)$. If a walk restarts at $v$, the probability of reaching $x$ before the next restart is at least the probability of exactly walking the shortest path from $v$ to $x$, which is $\geq (\frac{1-\alpha}{\Delta})^{d(v,x)} \in (0, 1)$. Then $P(W_N(v, x^-)) \leq (1 - (\frac{1-\alpha}{\Delta})^{d(v,x)})^N \downarrow 0$, leading to $P(W(v, x^-)) = 0$. This is a contradiction, so we have $P(W(u, v, x^-)) = 0$.

We are now ready to bound $H(u, x)$. Using the hitting time function $|\cdot|_x$ in Equation 56, we have:

$$H(u, x) = \mathbb{E}\left[ |w|_x | w \in W(u) \right]. \tag{59}$$

Since $W(u) = W(u,x) \cup W(u,x^-)$, and $P(W(u,x^-)) = 0$ from Equation 57, we have:

$$H(u,x) = \mathbb{E}[|w|_x | w \in W(u,x)],$$

$$= \int_{W(u,x)} |w|_x \, dP(w | w \in W(u,x)). \tag{60}$$

Each walk in $W(u,x)$ starts at $u$ and, when it reaches $x$, it either has or has not visited $v$. We treat the two cases separately. Let $W(a,b,c)$ be the set of walks that start at $a$ and reach $c$ after $b$, and let $W(a,b^-,c)$ be the set of walks that start at $a$ and reach $c$ before $b$. Then we have:

$$H(u,x) = \int_{W(u,v,x) \cup W(u,v^-,x)} |w|_x \, dP(w|W(u,x)),$$

$$= \int_{W(u,v,x)} |w|_x \, dP(w|W(u,x)) + \int_{W(u,v^-,x)} |w|_x \, dP(w|W(u,x)). \tag{61}$$

Let $\hat{H}(a,b,c)$ (or $\hat{H}(a,b^-,c)$) be the expected number of steps for a walk starting at $a$ to reach $c$ after reaching $b$ (or before $b$), given that it is in $W(a,b,c)$ (or in $W(a,b^-,c)$). If a walk from $u$ reaches $v$ before $x$, the expected steps is given by $\hat{H}(u,x^-,v) + H(v,x)$. If the walk reaches $x$ before $v$, the expected steps is $\hat{H}(u,v^-,x)$. Then, if $W(u,v^-,x) \neq \emptyset$, we can write $H(u,x)$ as follows:

$$H(u,x) = \left[\hat{H}(u,x^-,v) + H(v,x)\right] P(W(u,v,x)|W(u,x))$$

$$+ \hat{H}(u,v^-,x) \, P(W(u,v^-,x)|W(u,x)). \tag{62}$$

On the other hand, if $W(u,v^-,x) = \emptyset$, we simply have:

$$H(u,x) = \hat{H}(u,x^-,v) + H(v,x). \tag{63}$$

We first consider $\hat{H}(u,x^-,v)$, the expected number of steps from $u$ to reach $v$ before $x$. We show:

$$\hat{H}(u,x^-,v) \leq \frac{1}{\alpha}. \tag{64}$$

To see this, note that $\hat{H}(u,x^-,v)$ is equivalent to the expectation $\mathbb{E}[T]$ of the number of steps $T$ to reach $v$ from $u$, on the graph $G$ with the vertex $x$ deleted and transition probabilities renormalized. If $u$ is isolated on this modified graph, the only walk in $W(u,x^-,v)$ is the one that immediately restarts at $v$, giving $\mathbb{E}[T] = 1$. Otherwise, we have $\mathbb{E}[T] \leq 1/\alpha$ due to the following. Let $T'$ be the number of steps until the first restart at $v$. Since restart can be treated as a Bernoulli trial with probability of success $\alpha$, $T'$ follows geometric distribution with expectation $1/\alpha$. Since it is possible for the walk to reach $v$ before the first restart, we have $\mathbb{E}[T] \leq \mathbb{E}[T'] = 1/\alpha$, which gives Equation 64.

We now consider $H(v,x)$, the expectation $\mathbb{E}[T]$ of the number of steps $T$ to reach $x$ from $v$. Let $T'$ be the steps until the walk restarts at $v$, then exactly walks the shortest path from $v$ to $x$ for the first time, and then restarts at $v$. Since $T'$ walks until restart after walking the shortest path to $x$, and it is possible for a walk to reach $x$ before walking the shortest path to it, we have $\mathbb{E}[T] \leq \mathbb{E}[T']$. Then, we split the walk of length $T'$ into $N$ trials, where each trial consists of restarting at $v$ and walking until the next restart. A trial is successful if it immediately walks the shortest path from $v$ to $x$. Then $N$ is the number of trials until we succeed, and it follows geometric distribution with probability of success at least $(\frac{1-\alpha}{\Delta})^{d(v,x)}$ due to bounded degrees. Its expectation is then bounded as:

$$\mathbb{E}[N] \leq \left(\frac{\Delta}{1-\alpha}\right)^{d(v,x)}. \tag{65}$$

Let $S_i$ be the length of the $i$-th trial. Since each $S_i$ is i.i.d. with finite mean $\mathbb{E}[S_i] = 1/\alpha$, and $N$ is stopping time, we can apply Wald's identity (Hein) to compute the expectation of $T'$:

$$\mathbb{E}[T'] = \mathbb{E}[S_1 + \cdots + S_N],$$

$$= \mathbb{E}[N]\mathbb{E}[S_1],$$

$$\leq \frac{1}{\alpha}\left(\frac{\Delta}{1-\alpha}\right)^{d(v,x)}. \tag{66}$$

We remark that $H(v, x) \leq \mathbb{E}[T']$. Combining this result with Equation 64, we have:

$$\hat{H}(u, x^-, v) + H(v, x) \leq \frac{1}{\alpha} + \frac{1}{\alpha}\left(\frac{\Delta}{1-\alpha}\right)^{d(v,x)}. \tag{67}$$

If $W(u, v^-, x) = \emptyset$, we have $H(u, x) = \hat{H}(u, x^-, v) + H(v, x)$ from Equation 63 and it is finitely bounded for any $\alpha \in (0, 1)$ by the above. If $W(u, v^-, x) \neq \emptyset$, Equations 62 and 67 lead to:

$$H(u, x) \leq \hat{H}(u, x^-, v) + H(v, x) + \hat{H}(u, v^-, x)\, P(W(u, v^-, x)|W(u, x)),$$

$$\leq \frac{1}{\alpha} + \frac{1}{\alpha}\left(\frac{\Delta}{1-\alpha}\right)^{d(v,x)} + \hat{H}(u, v^-, x)\, P(W(u, v^-, x)|W(u, x)), \tag{68}$$

and it suffices to bound $\hat{H}(u, v^-, x)\, P(W(u, v^-, x)|W(u, x))$. We show the following:

$$\hat{H}(u, v^-, x) = \int_{W(u,v^-,x)} |w|_x \, dP(w|W(u, v^-, x)),$$

$$= \sum_{k=1}^{\infty} \int_{\{w \in W(u,v^-,x) \wedge |w|_x = k\}} |w|_x \, dP(w|W(u, v^-, x)),$$

$$= \sum_{k=1}^{\infty} k \int_{\{w \in W(u,v^-,x) \wedge |w|_x = k\}} dP(w|W(u, v^-, x)),$$

$$= \sum_{k=1}^{\infty} k\, P(\{w \in W(u, v^-, x) \wedge |w|_x = k\}|W(u, v^-, x)),$$

$$\leq \sum_{k=1}^{\infty} k\, P(\{w : |w|_x = k\}|W(u, v^-, x)),$$

$$= \sum_{k=1}^{\infty} k\, \frac{P(\{w : |w|_x = k \wedge w \in W(u, v^-, x)\})}{P(W(u, v^-, x))},$$

$$\leq \sum_{k=1}^{\infty} k(1-\alpha)^k \frac{1}{P(W(u, v^-, x))},$$

$$= \frac{1}{\alpha}\left(\frac{1}{\alpha} - 1\right) \frac{1}{P(W(u, v^-, x))}. \tag{69}$$

We have used Fubini's theorem for the second equality. Then we have:

$$\hat{H}(u, v^-, x)\, P(W(u, v^-, x)|W(u, x)) \leq \frac{1}{\alpha}\left(\frac{1}{\alpha} - 1\right)\frac{P(W(u, v^-, x)|W(u, x))}{P(W(u, v^-, x))},$$

$$= \frac{1}{\alpha}\left(\frac{1}{\alpha} - 1\right)\frac{1}{P(W(u, x))}. \tag{70}$$

$P(W(u, x))$ is at least the probability of precisely walking the shortest path from $u$ to $x$, which has length $d(u, x) \leq 2r$ since $u$ and $x$ are both in $B_r(v)$. This gives us the following:

$$P(W(u, x)) \geq \left(\frac{1-\alpha}{\Delta}\right)^{d(u,x)},$$

$$\geq \left(\frac{1-\alpha}{\Delta}\right)^{2r}. \tag{71}$$

Combining this with Equation 70, we have:

$$\hat{H}(u, v^-, x)\, P(W(u, v^-, x)|W(u, x)) \leq \frac{1}{\alpha}\left(\frac{1}{\alpha} - 1\right)\left(\frac{\Delta}{1-\alpha}\right)^{2r}. \tag{72}$$

Combining with Equations 62 and 67, we have:

$$H(u, x) \leq \frac{1}{\alpha} + \frac{1}{\alpha}\left(\frac{\Delta}{1-\alpha}\right)^{d(v,x)} + \frac{1}{\alpha}\left(\frac{1}{\alpha} - 1\right)\left(\frac{\Delta}{1-\alpha}\right)^{2r}, \tag{73}$$

for any $\alpha \in (0, 1)$. Notice that, while this bound is for the case $W(u, v^-, x) \neq \emptyset$, it subsumes the bound for the case $W(u, v^-, x) = \emptyset$ (Equation 67). Then, using $d(v, x) \leq r$, we get Equation 54. This completes the proof for random restarts.

We now prove for periodic restarts. If the walk starting at $u$ reaches $x$ before restarting at $v$, the steps taken is clearly less than $k$. If the walk starting at $u$ restarts at $v$ at step $k$ before reaching $x$, it now needs to reach $x$ from $v$, while restarting at $v$ at every $k$ steps. Let $T$ be the steps taken to reach $x$ from $v$. Let $T'$ be the number of steps until the walk restarts at $v$, then exactly follows the shortest path from $v$ to $x$ for the first time, and then restarts at $v$. It is clear that $\mathbb{E}[T] \leq \mathbb{E}[T']$. Then, we split the walk of length $T'$ into $N$ trials, where each trial consists of restarting at $v$ and walking $k$ steps until the next restart. A trial is successful if it immediately walks the shortest path from $v$ to $x$. Then $N$ is the number of trials until we get a success, and it follows geometric distribution with probability of success at least $(1/\Delta)^{d(v,x)}$ only for $k \geq d(v, x)$, and zero for $k < d(v, x)$ since the walk cannot reach $x$ before restart. Hence, its expectation is at most $\Delta^{d(v,x)}$ for $k \geq d(v, x)$, and we have $\mathbb{E}[T] \leq \mathbb{E}[T'] = k\mathbb{E}[N] \leq k\Delta^{d(v,x)}$. Adding the $k$ steps until the first restart at $v$, we have:

$$H(u, x) \leq k + k\Delta^{d(v,x)}, \tag{74}$$

for any $k \geq d(v, x)$. Using $d(v, x) \leq r$, we get Equation 55. This completes the proof. $\qquad \square$

We now extend Lemma A.4 to edges. Let $H(u, (x, y))$ be the expected number of steps of a random walk starting at $u$ takes until traversing an edge $(x, y)$ by $x \to y$. We show that $H(u, (x, y))$ is finite for any $u$ and adjacent $x, y$ in $B_r(v)$ for any nonzero restart probability $\alpha$ or restart period $k \geq r + 1$.

**Lemma A.5.** *For any $u$ and adjacent $x, y$ in $B_r(v)$, $H(u, (x, y))$ is bounded by the following:*

$$H(u, (x, y)) \leq \frac{1}{\alpha} + \frac{1}{\alpha}\left(\frac{\Delta}{1-\alpha}\right)^{r+1} + \frac{1}{\alpha}\left(\frac{1}{\alpha} - 1\right)\left(\frac{\Delta}{1-\alpha}\right)^{2r+1}, \tag{75}$$

*if the random walk restarts at $v$ with any probability $\alpha \in (0, 1)$, and:*

$$H(u, (x, y)) \leq k + k\Delta^{r+1}, \tag{76}$$

*if the random walk restarts at $v$ with any period $k \geq r + 1$.*

*Proof.* The proof is almost identical to Lemma A.4, except the target $x$ of reaching is substituted by $(x, y)$ in the direction of $x \to y$, and all arguments that use the shortest path from $u$ or $v$ to $x$ instead use the shortest path to $x$ postfixed by $x \to y$, which adds $+1$ to several terms in the bounds. $\qquad \square$

We are now ready to prove Theorem 2.5.

**Theorem 2.5.** *In Theorem 2.4, if the random walk restarts at $v$ with any nonzero probability $\alpha$ or any period $k \geq r + 1$, the vertex and edge cover times of $B_r(v)$ are always finite.*

*Proof.* The proof is inspired by the spanning tree argument of Aleliunas et al. (1979). Let us consider a depth first search of $B_r(v)$ starting from $v$. We denote by $T$ the resulting spanning tree with vertices $V(T) = V(B_r(v))$. We consider the expected time for a random walk starting at $v$ to visit every vertex in the precise order visited by the depth first search by traversing each edge twice. It is clear that this upper-bounds the vertex cover time of $B_r(v)$ starting at $v$ (Equation 18):

$$C_V(B_r(v)) \leq \sum_{(x,y) \in E(T)} [H(x, y) + H(y, x)]. \tag{77}$$

Then, using the bounds from Lemma A.4, the property of spanning trees $|E(T)| = |V(T)| - 1$, and the fact that $|V(T)| = |V(B_r(v))| \leq \Delta^r$ from bounded degree, we obtain:

$$C_V(B_r(v)) \leq 2(\Delta^r - 1)\left(\frac{1}{\alpha} + \frac{1}{\alpha}\left(\frac{\Delta}{1-\alpha}\right)^r + \frac{1}{\alpha}\left(\frac{1}{\alpha} - 1\right)\left(\frac{\Delta}{1-\alpha}\right)^{2r}\right), \tag{78}$$

if the random walk restarts at $v$ with any probability $\alpha \in (0, 1)$, and:

$$C_V(B_r(v)) \leq 2(\Delta^r - 1)(k + k\Delta^r), \tag{79}$$

if the random walk restarts at $v$ with any period $k \geq r$. This completes the proof for the vertex cover time. For the edge cover time, we consider the expected time for a random walk starting at $v$ to visit every edge in the precise order discovered[7] by the depth first search by traversing each edge twice. It is clear that this upper-bounds the edge cover time of $B_r(v)$ starting at $v$ (Equation 19):

$$C_E(B_r(v)) \leq \sum_{(x,y) \in E(B_r(v))} [H(x, (x,y)) + H(y, (y,x))]. \tag{80}$$

Then, using Lemma A.4 and the fact that $|E(B_r(v))| \leq \Delta^{2r} - 1$ from bounded degree, we obtain:

$$C_E(B_r(v)) \leq 2(\Delta^{2r} - 1)\left(\frac{1}{\alpha} + \frac{1}{\alpha}\left(\frac{\Delta}{1-\alpha}\right)^{r+1} + \frac{1}{\alpha}\left(\frac{1}{\alpha} - 1\right)\left(\frac{\Delta}{1-\alpha}\right)^{2r+1}\right), \tag{81}$$

if the random walk restarts at $v$ with any probability $\alpha \in (0, 1)$, and:

$$C_E(B_r(v)) \leq 2(\Delta^{2r} - 1) \cdot (k + k\Delta^{r+1}), \tag{82}$$

if the random walk restarts at $v$ with any period $k \geq r + 1$. This completes the proof. $\square$

While our proof shows finite bounds for the cover times, it is possible that they can be made tighter, for instance based on Zuckerman (1991). We leave improving the bounds as a future work.

### A.9.7 Proof of Theorem 3.2 (Section 3.1)

We recall universal approximation of graph-level functions in probability (Definition 3.1):

**Definition 3.1.** *We say $X_\theta(\cdot)$ is a universal approximator of graph-level functions in probability if, for all invariant functions $\phi : \mathbb{G}_n \to \mathbb{R}$ for a given $n \geq 1$, and $\forall \epsilon, \delta > 0$, there exist choices of length $l$ of the random walk and network parameters $\theta$ such that the following holds:*

$$\text{Prob}[|\phi(G) - X_\theta(G)| < \epsilon] > 1 - \delta, \quad \forall G \in \mathbb{G}_n. \tag{83}$$

We remark that an RWNN $X_\theta(\cdot)$ is composed of a random walk algorithm, a recording function $q : (v_0 \to \cdots \to v_l, G) \mapsto \mathbf{z}$, and a reader neural network $f_\theta : \mathbf{z} \mapsto \hat{\mathbf{y}} \in \mathbb{R}$.

Intuitively, if the record $\mathbf{z}$ of the random walk always provides complete information of the input graph $G$, we may invoke universal approximation of $f_\theta$ to always obtain $|\phi(G) - f_\theta(\mathbf{z})| < \epsilon$, and thus $|\phi(G) - X_\theta(G)| < \epsilon$. However, this is not always true as the random walk may e.g. fail to visit some vertices of $G$, in which case the record $\mathbf{z}$ would be incomplete. As we show below, this uncertainty leads to the probabilistic bound $> 1 - \delta$ of the approximation.

Let us denote the collection of all possible random walk records as $\{\mathbf{z}\} := \text{Range}(q)$, and consider a decoding function $\psi : \{\mathbf{z}\} \to \mathbb{G}_n$ that takes the record $\mathbf{z} := q(v_0 \to \cdots \to v_l, G)$ of a given random walk $v_{[\cdot]}$ and outputs the graph $\psi(\mathbf{z}) \in \mathbb{G}_n$ composed of all recorded vertices $V(H) := \{\text{id}(v_t) : v_t \in \{v_0, ..., v_l\}\}$ and all recorded edges $E(H) \subset V(H) \times V(H)$. We show the following lemma:

**Lemma A.6.** *Let $G_\mathbf{z}$ be the subgraph of $G$ whose vertices and edges are recorded by $\mathbf{z}$. Then the graph $\psi(\mathbf{z})$ decoded from the record $\mathbf{z}$ is isomorphic to $G_\mathbf{z}$ through the namespace $\text{id}(\cdot)$:*

$$G_\mathbf{z} \overset{\text{id}}{\simeq} \psi(\mathbf{z}). \tag{84}$$

*Furthermore, the decoded graph $\psi(\mathbf{z})$ reconstructs $G$ up to isomorphism, that is,*

$$G \overset{\text{id}}{\simeq} \psi(\mathbf{z}), \tag{85}$$

*if the recording function $q(\cdot)$ and the random walk $v_{[\cdot]}$ satisfies either of the following:*

- *$q(\cdot)$ uses anonymization, and $v_{[\cdot]}$ has traversed all edges of $G$.*

- *$q(\cdot)$ uses anonymization and named neighbors, and $v_{[\cdot]}$ has visited all vertices of $G$.*

---

[7]We remark that depth first search discovers all edges of a graph, while not necessarily visiting all of them.

*Proof.* Equation 84 is straightforward from the fact that the namespace $\mathrm{id}(\cdot)$ defines a bijection from $V(G_{\mathbf{z}})$ to $[|V(G_{\mathbf{z}})|]$, and the recording function uses names $\mathrm{id}(v_t)$ to record vertices and edges. Equation 85 is satisfied when all vertices and edges of $G$ have been recorded, i.e., $G_{\mathbf{z}} = G$, which is possible either when the random walk has traversed all edges of $G$, or when it has traversed all vertices of $G$ and named neighbors are used to record the induced subgraph $G[V(G)] = G$. $\quad\square$

We further remark Markov's inequality for any nonnegative random variable $T$ and $a > 0$:

$$\mathrm{Prob}[T \geq a] \leq \frac{\mathbb{E}[T]}{a}. \tag{86}$$

We are now ready to prove Theorem 3.2.

**Theorem 3.2.** *An RWNN $X_\theta(\cdot)$ with a sufficiently powerful $f_\theta$ is a universal approximator of graph-level functions in probability (Definition 3.1) if it satisfies either of the below:*

- *It uses anonymization to record random walks of lengths $l > C_E(G)/\delta$.*

- *It uses anonymization and named neighbors to record walks of lengths $l > C_V(G)/\delta$.*

*Proof.* Instead of directly approximating the target function $\phi : \mathbb{G}_n \to \mathbb{R}$, it is convenient to define a proxy target function on random walk records $\phi' : \{\mathbf{z}\} \to \mathbb{R}$ where $\{\mathbf{z}\} := \mathrm{Range}(q)$ as follows:

$$\phi' := \phi \circ \psi, \tag{87}$$

where $\psi : \{\mathbf{z}\} \to \mathbb{G}_n$ is the decoding function of walk records. Then, for a given $\mathbf{z}$, we have:

$$G \simeq \psi(\mathbf{z}) \quad \implies \quad \phi(G) = \phi'(\mathbf{z}), \tag{88}$$

which is because $\phi$ is an invariant function, so $\phi(G) = \phi(\psi(\mathbf{z})) = \phi \circ \psi(\mathbf{z}) = \phi'(\mathbf{z})$. Then we have:

$$\mathrm{Prob}[G \simeq \psi(\mathbf{z})] \leq \mathrm{Prob}[|\phi(G) - \phi'(\mathbf{z})| = 0]. \tag{89}$$

We now invoke universality of $f_\theta$ to approximate $\phi'$. If $f_\theta$ is a universal approximator of functions on its domain $\{\mathbf{z}\} := \mathrm{Range}(q)$, for any $\epsilon > 0$ there exists a choice of $\theta$ such that the below holds:

$$|\phi'(\mathbf{z}) - f_\theta(\mathbf{z})| < \epsilon, \quad \forall \mathbf{z} \in \mathrm{Range}(q). \tag{90}$$

Combining Equations 89 and 90, we have:

$$\mathrm{Prob}[G \simeq \psi(\mathbf{z})] \leq \mathrm{Prob}[|\phi(G) - \phi'(\mathbf{z})| + |\phi'(\mathbf{z}) - f_\theta(\mathbf{z})| < \epsilon]. \tag{91}$$

We remark triangle inequality of distances on $\mathbb{R}$, for a given $\mathbf{z}$:

$$|\phi(G) - f_\theta(\mathbf{z})| \leq |\phi(G) - \phi'(\mathbf{z})| + |\phi'(\mathbf{z}) - f_\theta(\mathbf{z})|, \tag{92}$$

which implies, for a given $\mathbf{z}$:

$$|\phi(G) - \phi'(\mathbf{z})| + |\phi'(\mathbf{z}) - f_\theta(\mathbf{z})| < \epsilon \quad \implies \quad |\phi(G) - f_\theta(\mathbf{z})| < \epsilon, \tag{93}$$

and hence:

$$\mathrm{Prob}[|\phi(G) - \phi'(\mathbf{z})| + |\phi'(\mathbf{z}) - f_\theta(\mathbf{z})| < \epsilon] \leq \mathrm{Prob}[|\phi(G) - f_\theta(\mathbf{z})| < \epsilon]. \tag{94}$$

Combining Equations 91 and 94, we have:

$$\mathrm{Prob}[|\phi(G) - f_\theta(\mathbf{z})| < \epsilon] \geq \mathrm{Prob}[G \simeq \psi(\mathbf{z})], \tag{95}$$

which can be written as follows:

$$\mathrm{Prob}[|\phi(G) - X_\theta(G)| < \epsilon] \geq \mathrm{Prob}[G \simeq \psi(\mathbf{z})]. \tag{96}$$

We now consider the probability of the event $G \simeq \psi(\mathbf{z})$ based on Lemma A.6. We first consider the case where the recording function $q(\cdot)$ uses anonymization. In this case, $G \simeq \psi(\mathbf{z})$ is achieved if the random walk of length $l$ has traversed all edges of $G$. Let $T_E(G, v_0)$ be the number of steps that a random walk starting at $v_0$ takes until traversing all edges of $G$. Since the edge cover time $C_E(G)$ is its expectation taken at the worst possible starting vertex (Equation 17), we have the following:

$$\mathbb{E}[T_E(G, v_0)] \leq C_E(G), \quad \forall v_0 \in V(G), \tag{97}$$

which leads to the following from Markov's inequality (Equation 86):

$$\text{Prob}[T_E(G, v_0) < l] \geq 1 - \frac{\mathbb{E}[T_E(G)]}{l} \geq 1 - \frac{C_E(G)}{l}. \tag{98}$$

For a given random walk $v_0 \to \cdots \to v_l$ and its record $\mathbf{z}$, the following holds:

$$T_E(G, v_0) < l \quad \implies \quad G \simeq \psi(\mathbf{z}), \tag{99}$$

which implies the following:

$$\text{Prob}[T_E(G, v_0) < l] \leq \text{Prob}[G \simeq \psi(\mathbf{z})]. \tag{100}$$

Combining Equations 96, 98, and 100, we have:

$$\text{Prob}[|\phi(G) - X_\theta(G)| < \epsilon] \geq 1 - \frac{C_E(G)}{l}. \tag{101}$$

Therefore, for any $\delta > 0$, if we choose $l > C_E(G)/\delta$ we would have the following:

$$\text{Prob}[|\phi(G) - X_\theta(G)| < \epsilon] > 1 - \delta. \tag{102}$$

This completes the proof for anonymization. The proof is identical for the recording function that uses anonymization and named neighbors, except that the edge cover time is changed to the vertex cover time $C_V(G)$ (Equation 16). This is because named neighbor recording automatically records the induced subgraph of visited vertices, thus visiting all vertices implies recording all edges, $G[V(G)] = G$. $\qquad \square$

### A.9.8 PROOF OF THEOREM 3.4 (SECTION 3.1)

**Theorem 3.4.** *An RWNN $X_\theta(\cdot)$ with a sufficiently powerful $f_\theta$ and any nonzero restart probability $\alpha$ or restart period $k \geq r + 1$ is a universal approximator of vertex-level functions in probability (Definition 3.3) if it satisfies either of the below for all $B_r(v) \in \mathbb{B}_r$:*

- *It uses anonymization to record random walks of lengths $l > C_E(B_r(v))/\delta$.*

- *It uses anonymization and named neighbors to record walks of lengths $l > C_V(B_r(v))/\delta$.*

*Proof.* The proof is almost identical to Theorem 3.2, except $G \in \mathbb{G}_n$ are substituted by $B_r(v) \in \mathbb{B}_r$, and the decoding function $\psi : \{\mathbf{z}\} \to \mathbb{B}_r$ is defined to ignore all recorded vertices $\text{id}(x)$ whose shortest path distance from the starting vertex $\text{id}(v) = \text{id}(v_0) = 1$ exceeds $r$. The latter is necessary to restrict the range of the decoding function $\psi$ to $\mathbb{B}_r$. In addition, any nonzero restart probability $\alpha$ or restart period $k \geq r + 1$ is sufficient to make the cover times $C_E(B_r(v))$ and $C_V(B_r(v))$ finite (Theorem 2.5), thereby guaranteeing the existence of a finite choice of $l$. $\qquad \square$

### A.9.9 PROOF OF THEOREM 3.5 (SECTION 3.2)

**Theorem 3.5.** *The simple RWNN outputs $\mathbf{h}^{(l)} \to \mathbf{x}^\top \boldsymbol{\pi}$ as $l \to \infty$.*

*Proof.* Since $G$ is connected and non-bipartite, the uniform random walk on it defines an ergodic Markov chain with a unique stationary distribution $\boldsymbol{\pi}$. The limiting frequency of visits on each vertex $v$ is precisely the stationary probability $\boldsymbol{\pi}_v$. Since the model reads $\mathbf{x}_{v_0} \to \cdots \to \mathbf{x}_{v_l}$ by average pooling, the output is given by weighted mean $\sum_v \boldsymbol{\pi}_v \mathbf{x}_v$ which is $\mathbf{x}^\top \boldsymbol{\pi}$. $\qquad \square$

### A.9.10 PROOF OF THEOREM 3.6 (SECTION 3.2)

**Theorem 3.6.** *Let $\mathbf{h}_u^{(l)}$ be output of the simple RWNN queried with $u$. Then:*

$$\mathbb{E}\left[\left\|\frac{\partial \mathbf{h}_u^{(l)}}{\partial \mathbf{x}_v}\right\|\right] = \frac{1}{l+1}\left[\sum_{t=0}^{l} P^t\right]_{uv} \to \boldsymbol{\pi}_v \quad as \quad l \to \infty. \tag{103}$$

*Proof.* Since the model reads $\mathbf{x}_{v_0} \to \cdots \to \mathbf{x}_{v_l}$ by average pooling, the feature Jacobian $|\partial \mathbf{h}_u^{(l)} / \partial \mathbf{x}_v|$ is given as number of visits to the vertex $v$ in the random walk $v_0 \to \cdots \to v_l$ starting at $v_0 = u$, divided by length $l + 1$. Let us denote the expected number of these visits by $J(u, v, l)$. Let $\mathbb{1}_{v_t = v}$ be the indicator function that equals 1 if $v_t = v$ and 0 otherwise. Then we can write $J(u, v, l)$ as:

$$
\begin{aligned}
J(u, v, l) &= \mathbb{E}\left[\sum_{t=0}^{l} \mathbb{1}_{v_t = v} | v_0 = u\right], \\
&= \sum_{t=0}^{l} \mathbb{E}[\mathbb{1}_{v_t = v} | v_0 = u].
\end{aligned}
\tag{104}
$$

We have used linearity of expectations for the second equality. $\mathbb{E}[\mathbb{1}_{v_t = v} | v_0 = u]$ is the probability of being at $v$ at step $t$ given that the walk started at $u$. This probability is precisely $[P^t]_{uv}$. Therefore:

$$
J(u, v, l) = \sum_{t=0}^{l} \left[P^t\right]_{uv} = \left[\sum_{t=0}^{l} P^t\right]_{uv},
\tag{105}
$$

which gives the equality in Equation 103. Furthermore, since $G$ is connected and non-bipartite, the uniform random walk on it defines an ergodic Markov chain with a unique stationary distribution $\boldsymbol{\pi}$. The limiting frequency of visits on vertex $v$ is precisely the stationary probability $\boldsymbol{\pi}_v$, which gives the convergence in Equation 103. This completes the proof. $\square$

A.10  EXTENDED RELATED WORK (SECTION 4)

**Anonymized random walks (Micali & Zhu, 2016)**   The initial work on anonymization by Micali & Zhu (2016) has stated an important result, that a sufficiently long anonymized walk starting from a vertex $v$ encodes sufficient information to reconstruct the local subgraph $B_r(v)$ up to isomorphism. While the focus of Micali & Zhu (2016) was a probabilistic graph reconstruction algorithm that uses a set of independent anonymized walks and accesses the oracle set of all possible anonymized walks, we adopt the idea in a neural processing context to acquire universality in probability (Section 3.1). In addition, the invariance property of anonymization has rarely been noticed formally in the literature, which is our key motivation for using it, on top of being able to recover the whole graph (Section 2).

**In-depth comparison with CRaWl (Tönshoff et al., 2023)**   Our approach is related to CRaWl in two key aspects: **(1)** identity and connectivity encodings of CRaWl contains analogous information to anonymization and named neighbor recording, respectively, and **(2)** CRaWl uses 1D CNNs as the reader NN. A key technical difference lies in the first part. The identity and connectivity encodings of CRaWl are defined within a fixed window size (denoted $s$ in (Tönshoff et al., 2023)), which puts a locality constraint on the recorded information. Precisely, the encodings at step $t$ can encode the information of the walk from step $t - s$ to $t$ (precisely, its induced subgraph), referred to as a walklet. The window size $s$ is a hyperparameter that controls the expressive power of CRaWl, and this dependency makes CRaWl non-universal (Section 3.2, (Tönshoff et al., 2023)). Our choice of anonymization and named neighbor recording are not under such a local constraint, and they encode the full information of a given walk globally. This property underlies our universality results in Section 3.1, which also naturally motivates our choice of universal reader NNs, e.g. a transformer language model, which were not explicitly considered in CRaWl.

**Random walks on graphs**   Our work builds upon theory of random walks on graphs, i.e., Markov chains on discrete spaces. Their statistical properties such as hitting and mixing times have been well-studied (Aleliunas et al., 1979; Lovász, 1993; Coppersmith et al., 1996; Feige, 1995; Oliveira, 2012; Peres & Sousi, 2015), and our Section 2 is related to vertex cover time (Aleliunas et al., 1979; Kahn et al., 1989; Ding et al., 2011; Abdullah, 2012), edge cover time (Zuckerman, 1991; Bussian, 1996; Panotopoulou, 2013; Georgakopoulos & Winkler, 2014), and improving them, using local degree information (Ikeda et al., 2009; Abdullah et al., 2015; David & Feige, 2018), non-backtracking (Alon et al., 2007; Kempton, 2016; Arrigo et al., 2019; Fasino et al., 2021), or restarts (Dumitriu et al., 2003; McNew, 2013; Janson & Peres, 2012). Our work is also inspired by graph algorithms based on random walks, such as anonymous observation (Micali & Zhu, 2016), sublinear algorithms (Dasgupta et al., 2014; Chiericetti et al., 2016; Ben-Hamou et al., 2018; Bera & Seshadhri, 2020), and personalized PageRank for search (Page et al., 1999) and message passing (Gasteiger et al., 2019). While we adopt their techniques to make our walks and their records well-behaved, the difference is that we use a non-linear deep neural network to process the records and directly make predictions. Our method is also related to label propagation algorithms (Zhu & Ghahramani, 2002; Zhu, 2005; Grady, 2006) that perform transductive learning on graphs based on random walks, as we have discussed in Section 5.3.

**Over-smoothing and over-squashing**   Prior work on over-smoothing and over-squashing of MPNNs (Barceló et al., 2020; Topping et al., 2022; Giovanni et al., 2023; Giraldo et al., 2023; Black et al., 2023; Nguyen et al., 2023; Park et al., 2023; Wu et al., 2023) often make use of structural properties of graphs, such as effective resistance (Doyle & Snell, 1984; Chandra et al., 1997), Laplacian eigen-spectrum (Lovász, 1993; Spielman, 2019), and discrete Ricci curvatures (Ollivier, 2009; Devriendt & Lambiotte, 2022). Interestingly, random walks and their statistical properties are often closely related to these properties, indicating some form of parallelism between MPNNs and RWNNs. This has motivated our analysis in Section 3.2, where we transfer the prior results on over-smoothing and over-squashing of MPNNs based on these properties into the results on our approach.

Unlike the MPNNs considered in the above prior work, AMP (Errica et al., 2023) mitigates over-smoothing and over-squashing by learning to prune messages at each step (among its other techniques). This brings it close to attention-based MPNNs, which can be understood as modifying the computation ($P$ in Section 3.2) over topology. Indeed, our description of over-smoothing and over-squashing in MPNNs in Section 3.2 mainly applies to message passing using fixed weights, and attention falls out of this framework, which is a shared limitation with prior work (Topping et al., 2022; Giraldo et al., 2023; Black et al., 2023; Nguyen et al., 2023). Formal propagation analysis for attention-based

MPNNs is currently in its infancy (Wu et al., 2023), but we believe it is interesting since, based on our parallelism between MPNNs and RWNNs, it may tell us about how to adapt the walk probabilities adaptively based on input data. We leave investigating this as future work.

**Language models for learning on graphs**   While not based on random walks, there have been prior attempts on applying language models for problems on graphs (Wang et al., 2023; Chen et al., 2023; Zhao et al., 2023; Fatemi et al., 2023; Ye et al., 2024), often focusing on prompting methods on problems involving simulations of graph algorithms. We take a more principle-oriented approach based on invariance and expressive power, and thus demonstrate our approach mainly on the related tasks, e.g. graph separation. We believe extending our work to simulating graph algorithms requires a careful treatment (Weiss et al., 2021; Delétang et al., 2023; de Luca & Fountoulakis, 2024; Sanford et al., 2024) and plan to investigate it as future work.

### A.11   LIMITATIONS AND FUTURE WORK

Our current main limitation is the cost of performing training and inference if using language models, especially on long walks. While random walk sampling and text recording can be done efficiently, the main computational bottleneck comes from the use of language models as the reader NN. For example, in Section 5.3, each test prediction of RWNN-Llama-3-70b takes around $0.410 \pm 0.058$ seconds on a $4\times$ H100 machine (RWNN-Llama-3-8b is faster, $0.116 \pm 0.064$ seconds per prediction). The inference time of standard MPNNs are much shorter, e.g., a 3-layer GAT with 16 hidden dimensions takes around one second for all test vertices of ogbn-arxiv on a V100 GPU (Wang et al., 2019). Overcoming this issue is an important future direction. We remark that, if language model embeddings are used and fine-tuned, the training cost of MPNNs can also be substantial, e.g., recent work has reported that prior methods for training an MPNN jointly with language model embeddings on arXiv take 25-46 hours on a $6\times$ RTX 3090 machine (Table 3 of Zhu et al. (2024)).

On the theory side, we focus on cover times of random walks in our analysis (Section 3.1 and Appendix A.8), which corresponds to the worst-case length bounds required for universal approximation. Yet, in practice, there is a substantial amount of useful information that can be already recovered from possibly non-covering walks. Examples include spectral properties (Benjamini et al., 2006), number of edges and mixing time (Ben-Hamou et al., 2018), and triangle counts (Bera & Seshadri, 2020), among others studied in the field of sublinear algorithms. These results suggest that shorter walks may suffice in practice for RWNNs, if the task at hand requires properties that can be estimated without seeing the whole graph, and the reader NN can learn to recover them (e.g., leveraging universality). Understanding this aspect better is an important direction for future work.

Another question for future work is whether we can train a language model on a large pseudocorpus of random walks (Klubicka et al., 2019; 2020) to build a foundation model for graphs which is language-compatible. We plan to investigate this direction in the future.

