# A   APPENDIX

## A.1   SECOND-ORDER RANDOM WALKS (SECTION 2)

In second-order random walks, the probability of choosing $v_{t+1}$ depends not only on $v_t$ but also on $v_{t-1}$. We consider non-backtracking (Alon et al., 2007; Fitzner & van der Hofstad, 2013) and node2vec (Grover & Leskovec, 2016) random walks as representatives.

A non-backtracking random walk is defined upon a first-order random walk algorithm, e.g. one in Equation 4, by enforcing $v_{t+1} \neq v_{t-1}$:

$$\text{Prob}[v_{t+1} = x | v_t = j, v_{t-1} = i] := \begin{cases} \dfrac{\text{Prob}[v_{t+1} = x | v_t = j]}{\sum_{y \in N(j) \setminus \{i\}} \text{Prob}[v_{t+1} = y | v_t = j]} & \text{for } x \neq i, \\ 0 & \text{for } x = i, \end{cases} \tag{13}$$

where $\text{Prob}[v_{t+1} = x | v_t = j]$ is the probability of walking $j \to x$ given by the underlying first-order random walk. Notice that the probabilities are renormalized over $N(j) \setminus \{i\}$. This is ill-defined in the case the walk traverses $i \to j$ and reaches a dangling vertex $j$ which has $i$ as its only neighbor, since $N(j) \setminus \{i\} = \emptyset$. In such cases, we allow the random walk to "begrudgingly" backtrack (Rappaport et al., 2017), $i \to j$ and then $j \to i$, given that it is the only possible choice due to the dangling of $j$.

In case of node2vec random walk (Grover & Leskovec, 2016), a weighting term $\alpha(v_{t-1}, v_{t+1})$ with two (hyper)parameters, return $p$ and in-out $q$, is introduced to modify the behavior of a first-order random walk:

$$\text{Prob}[v_{t+1} = x | v_t = j, v_{t-1} = i] := \frac{\alpha(i, x) \, \text{Prob}[v_{t+1} = x | v_t = j]}{\sum_{y \in N(j)} \alpha(i, y) \, \text{Prob}[v_{t+1} = y | v_t = j]}, \tag{14}$$

where the probability $\text{Prob}[v_{t+1} = x | v_t = j]$ is given by the underlying first-order random walk and $\alpha(v_{t-1}, v_{t+1})$ is defined as follows using the shortest path distance $d(v_{t-1}, v_{t+1})$:

$$\alpha(v_{t-1}, v_{t+1}) := \begin{cases} 1/p & \text{for } d(v_{t-1}, v_{t+1}) = 0, \\ 1 & \text{for } d(v_{t-1}, v_{t+1}) = 1, \\ 1/q & \text{for } d(v_{t-1}, v_{t+1}) = 2. \end{cases} \tag{15}$$

Choosing a large return parameter $p$ reduces backtracking since it decreases the probability of walking from $v_t$ to $v_{t-1}$. Choosing a small in-out parameter $q$ has a similar effect of avoiding $v_{t-1}$, with a slight difference that it avoids the neighbors of $v_{t-1}$ as well.

We now show an extension of Proposition 2.2 to the above second-order random walks:

**Proposition A.1.** *The non-backtracking random walk in Equation 13 and the node2vec random walk in Equation 14 are invariant if their underlying first-order random walk algorithm is invariant.*

*Proof.* The proof is given in Appendix A.9.3. $\qquad\square$

## A.2 MAIN ALGORITHM

We outline the main algorithms for the random walk and the recording function described in Section 2 and used in Section 5. Algorithm 1 shows the random walk algorithm, and Algorithms 2, 3, and 4 show the recording functions. For the random walk algorithm, we use non-backtracking described in Appendix A.1 and the minimum degree local rule conductance $c_G(\cdot)$ given in Equation 6 by default.

Based on the algorithms, in Figures 4, 5, and 6 we illustrate the task formats used for graph separation experiments in Section 5.2 by providing examples of input graphs, text records of random walks on them, and prediction targets for the language model that processes the records. Likewise, in Figures 7, 8, and 9 we show task formats for transductive classification on arXiv citation network in Section 5.3.

---

**Algorithm 1:** Random walk algorithm

---

**Data:** Input graph $G$, optional input vertex $v$, conductance function $c_G : E(G) \to \mathbb{R}_+$, walk length $l$, optional restart probability $\alpha \in (0, 1)$ or period $k > 1$
**Result:** Random walk $v_0 \to \cdots \to v_l$, restart flags $r_1, ..., r_l$

/* transition probabilities $p : E(G) \to \mathbb{R}_+$ */
1   **for** $(u, x) \in E(G)$ **do**
2    $\quad p(u, x) \leftarrow c_G(u, x) / \sum_{y \in N(u)} c_G(u, y)$        // Equation 4
    /* starting vertex $v_0$ */
3   **if** $v$ *is given* **then**
     /* query starting vertex */
4    $\quad v_0 \leftarrow v$
5   **else**
     /* sample starting vertex */
6    $\quad v_0 \sim \text{Uniform}(V(G))$
    /* random walk $v_1 \to \cdots \to v_l$, restart flags $r_1, ..., r_l$ */
7   $v_1 \sim \text{Categorical}(\{p(v_0, x) : x \in N(v_0)\})$
8   $r_1 \leftarrow 0$
9   **for** $t \leftarrow 2$ **to** $l$ **do**
     /* restart flag $r_t$ */
10   $\quad r_t \leftarrow 0$
11   $\quad$ **if** $\alpha$ *is given* **then**
       /* if restarted at $t-1$, do not restart */
12   $\quad\quad$ **if** $r_{t-1} = 0$ **then**
13   $\quad\quad\quad r_t \sim \text{Bernoulli}(\alpha)$
14   $\quad$ **else if** $k$ *is given* **then**
15   $\quad\quad$ **if** $t \equiv 0 \pmod{k}$ **then**
16   $\quad\quad\quad r_t \leftarrow 1$
     /* random walk $v_t$ */
17   $\quad$ **if** $r_t = 0$ **then**
18   $\quad\quad S \leftarrow N(v_{t-1}) \setminus \{v_{t-2}\}$
19   $\quad\quad$ **if** $S \neq \emptyset$ **then**
       /* non-backtracking */
20   $\quad\quad\quad v_t \sim \text{Categorical}(\{p(v_{t-1}, x) / \sum_{y \in S} p(v_{t-1}, y) : x \in S\})$    // Equation 13
21   $\quad\quad$ **else**
       /* begrudgingly backtracking */
22   $\quad\quad\quad v_t \leftarrow v_{t-2}$
23   $\quad$ **else**
     /* restart */