# OpenReview forum: "Revisiting Random Walks for Learning on Graphs"
_ICLR.cc/2025/Conference — ICLR 2025 Spotlight_

### Official Review · Reviewer_J6pj · 2024-10-30

**Soundness:** 3
**Presentation:** 4
**Contribution:** 4
**Rating:** 8
**Confidence:** 3

**Summary:**

The paper delivers a well-structured analysis, both theoretical and empirical, of the potential of random walk approaches as feature extractors for subsequent neural networks. The contribution is relevant both as a summary of previous random walk-based approaches and as a theoretical connection to the problems of oversmoothing, oversquashing, and underreaching. The paper first discusses the requirements for a random-walk feature extractor to be invariant to graph isomorphism, and later it shows that invariant random walks extractors can discriminate, in probability, any two graphs provided a sufficient length of the random walk is used. Random walks neural networks do not suffer from oversmoothing and the oversquashing is intimately related to probabilistic under-reaching (whereas in MPNNs one knows exactly how many steps are needed to reach a node).

**Strengths:**

This is one, if not the most, beautiful papers I have reviewed, and as such it deserves one of the most positive reviews I have ever written. The scope is well defined, the presentation is excellent and overall it is a pleasure to read. Coming from the GNN field, I found the connection of RWs and oversmoothing etc problems extremely fascinating.

Intuitive ideas have been well formalized into theorems (most of which seem ok, although I did not check all proofs carefully due to time shortage), some of which (Th. 3.2 and 3.4) are especially important to convince the GNN expressiveness community that it might be time to move beyond the WL-test arguments. The integration of GNNs and RWs seems a promising research direction in the future and this paper inspired me to think about it.

The empirical results support all theoretical findings (Figure 3 might be improved) and show some strong performance on discriminating regular graphs. The transformer architecture, and especially self-attention, seems to me a particularly adequate architecture for the tasks of Table 2, but it would be interesting to see how a simple MLP performs on top of the random walk features.

The authors also took care of efficiently implementing the core of their method so as to speed up the computation of random walks.

**Weaknesses:**

These are mostly suggestions for improvement, as there are no particular concerns on this reviewer’s side. One suggestion for the authors is to discuss more explicitly the empirical tasks, such as the one of Table 2; what is the target to be predicted? It is not enough to mention the relevant paper in this case, and the reader might get lost. Also, more details about the empirical procedure, such as hyper-parameters tried and model-selection/risk-assessment procedures, might contribute to improve replicability of the experiments.

I would also suggest to define the meaning of the symbols =d (define precisely what  means equal in probability) and ~ (the isomorphism relation). The authors use them a lot in the paper and they should be properly defined.

Another point that would need clarification is to specify how one can deal with named neighborhoods outside of the textual representation of the record. The authors could clarify possible options and hopefully run a demonstrative example, with an MLP for instance, in place of a language model.

In terms of experiments, I believe the authors might be trying to position RWNN-Llama3 too positively with respect to the literature. It is well-known that GraphSAGE, one of the simplest GNNs, already achieves 77% test accuracy on the ogbn-arxiv leaderboard. A proper model-selection was enough to significantly improve the original scores of the paper. So, while the paper deserves acceptance, it would do a service to the community to highlight this fact.

**Questions:**

These are mostly related to typos and requests for modification
  - Line  34: the authors cannot make a general claim that all GNNs are less expressive than 1-WL. The statement should be probably massaged to consider this
  - Line 53: should it be “Addressing such questions is important”?
  - Table 1: the authors claim that in principle the Reader NN of RWNN can be any universal function. I believe this might be true for many other methods the authors listed in the table. In light of this, I would suggest that the authors maintain consistency and replace “Any universal” with “Language Model”, since they have not shown how to combine RW+Neighbor Information records with models different from language models
  - Theorems 2.1, 2.2, 2.3 formalize rather intuitive concepts, and the proofs are mostly about stating the definition of invariance. Maybe the authors could “downgrade” them to Propositions to inform the reader that some results have more relevance than others?
  - Lines 347-349: recent architectures such as [1] show that it is possible to modify the computation over the topology to control oversmoothing and oversquashing, and these two concepts are not as entangled to the original topology and related to each other anymore. Perhaps the authors would like to rephrase their statement in light of this? [1] Errica F., et al. "Adaptive Message Passing: A General Framework to Mitigate Oversmoothing, Oversquashing, and Underreaching." arXiv preprint arXiv:2312.16560 (2023).
  - Figure 3: what are colors and lines? The authors should improve the figure a bit by adding a legend.

---

> ### Author Response · Authors · 2024-11-26
> **1/3**
>
> We thank the reviewer for the insightful comments and positive assessment of our work. We address the questions below.
>
> > **Q1.** For the tasks of Table 2, it would be interesting to see how a simple MLP performs on top of the random walk features.
>
> **A1.** Thank you for the suggestion, which could further support our theoretical results. We additionally tested a simple MLP on top of flattened walk features for Table 2. The MLP is a stack of 4 linear layers interleaved with ReLU, with the input dimension set as walk length times walk feature dimensions (set to 128), 128 hidden dimensions, and 1 output dimension (i.e. walk_len * 128 → 128 → 128 → 128 → 1). We denote the resulting model as RWNN-MLP.
>
> |  | Walk length | Clique (over-smoothing) | Barbell (over-squashing) |
> |---|---|---|---|
> | Baseline 1 | N/A | 30.94 ± 0.42 | 30.97 ± 0.42 |
> | Baseline 2 | N/A | 0.99 ± 0.08 | 1.00 ± 0.07 |
> | MLP | N/A | 1.10 ± 0.08 | 1.08 ± 0.07 |
> | GCN | N/A | 29.65 ± 0.34 | 1.05 ± 0.08 |
> | SAGE | N/A | 0.86 ± 0.10 | 0.90 ± 0.29 |
> | GAT | N/A | 20.97 ± 0.40 | 1.07 ± 0.09 |
> | BuNN | N/A | 0.03 ± 0.01 | 0.01 ± 0.07 |
> | RWNN-transformer | 100 | 0.047 ± 0.010 | 0.316 ± 0.127 |
> | RWNN-transformer | 1000 | **0.016 ± 0.006** | **0.005 ± 0.002** |
> | RWNN-MLP | 100 | 0.226 ± 0.103 | 0.409 ± 0.021 |
> | RWNN-MLP | 1000 | 0.725 ± 0.377 | 0.087 ± 0.045 |
>
> We see that RWNNs based on MLPs consistently perform better than GCN, SAGE, and GAT but are outperformed by RWNN-transformer. This supports our hypothesis that RWNNs can, in general, be resistant to over-smoothing and over-squashing (for the latter, if the walk is long), but also shows that proper choice of architecture is important to obtain the best performance. For example, the performance of RWNN-MLP for over-smoothing degrades when the walk length is increased from 100 to 1000, which is likely due to the information bottleneck caused by the fixed-size hidden features. RWNN-transformer does not experience this bottleneck as the size of its hidden feature scales with walk length, enabling it to achieve the best performance. The challenge of using MLPs directly for high-dimensional tasks is well-known (Bachman et al., 2023), but we think the MLP results are useful as a demonstration. We will include the results in the main text in the next revision.
>
> > **Q2.** Another point that would need clarification is to specify how one can deal with named neighborhoods outside of the textual representation of the record.
>
> **A2.** Thank you for raising this question. It is simple to use the neighborhood records outside textual representations. The simplest way is to encode it using a binary flag. Consider a 4-clique and a walk a → b → c → d. A textual record with neighbors would look like “1-2-3<1>-4<1,2>” (Line 195). The same information can be encoded in the following 2 x 7 matrix:
>
> 1 2 3 1 4 1 2
>
> 0 0 0 1 0 1 1
>
> where we use the binary flag in the second row to mark neighborhood recording. We will include this clarification in the main text.
>
> > **Q3.** One suggestion for the authors is to discuss more explicitly the empirical tasks, such as the one of Table 2; what is the target to be predicted? It is not enough to mention the relevant paper in this case, and the reader might get lost.
>
> **A3.** We agree that the task for Table 2 (Bamberger et al. 2024) should be described more clearly in the main text. In the task, each vertex is attributed with a binary flag and a random value. The task at each vertex is to regress the mean of values of all opposite-flag vertices. In Clique, the vertices of a 2N-clique are randomly partitioned into two sets of N vertices, and each set is marked with a binary flag. In Barbell, two N-cliques are linked to form a graph, and each clique is marked with a binary flag. In the next revision, we will clarify this detail and add a descriptive figure similar to Figure 3 of Bamberger et al. 2024 if the space constraint allows it.
>
> **References**
>
> Bachman et al. Scaling MLPs: A tale of inductive bias (2023)
>
> Bamberger et al. Bundle neural networks for message diffusion on graphs (2024)

---

> ### Author Response · Authors · 2024-11-26
> **2/3**
>
> > **Q4.** More details about the empirical procedure, such as hyper-parameters tried and model-selection/risk-assessment procedures, might contribute to improve replicability of the experiments.
>
> **A4.** We will make the model selection procedure more explicit in the next revision. For Section 5.1, we precisely followed the model depth, dimensionality, and training hyperparameters reported in Bamberger et al. 2024. For Section 5.2, the training hyperparameters were mainly taken from Kim et al. 2023 without modifications, except for the batch size, for which we searched for the least power of 2 stabilizing the training. The maximum number of tokens (512) was searched manually, and we found it to work robustly for all graph separation experiments. For Section 5.3, the restart probability $\alpha$ for random walks was the main hyperparameter for which we conducted a non-trivial search, and we used a 100-sample subset of validation data.
>
> > **Q5.** In terms of experiments, I believe the authors might be trying to position RWNN-Llama3 too positively with respect to the literature. It is well-known that GraphSAGE, one of the simplest GNNs, already achieves 77% test accuracy on the ogbn-arxiv leaderboard. A proper model-selection was enough to significantly improve the original scores of the paper. So, while the paper deserves acceptance, it would do a service to the community to highlight this fact.
>
> **A5.** We have revised the main text (e.g. Line 520-521) to better clarify that, while we obtain promising results, we do not currently claim the state-of-the-art on arXiv. We believe it is possible to improve our results by properly fine-tuning Llama-3-70b and we leave these engineering investigations for future work.
>
> > **Q6.** Line 34: the authors cannot make a general claim that all GNNs are less expressive than 1-WL. The statement should be probably massaged to consider this.
>
> **A6.** Thank you for pointing this out. At Line 34, by MPNNs, we were referring to the basic formulations in Gilmer et al. (2017) and Xu et al. (2019), but more recent and carefully designed variants indeed achieve, e.g., up to 3-WL expressive power (Maron et al. 2019; Frasca et al. 2022). We have revised “MPNNs can be viewed as…” in Line 33-34 to “MPNNs in their basic form can be viewed as…” to reflect this. We will add a discussion on recent 3-WL developments in the extended related works section (Appendix A.7) for further clarification in the next revision.
>
> > **Q7.** Table 1: the authors claim that in principle the Reader NN of RWNN can be any universal function. I believe this might be true for many other methods the authors listed in the table. In light of this, I would suggest that the authors maintain consistency and replace “Any universal” with “Language Model”, since they have not shown how to combine RW+Neighbor Information records with models different from language models.
>
> **A7.** We agree that any universal function can also replace the reader NN of prior methods. We think writing “any universal” would still be informative in understanding our contributions in Section 3.1 since no prior work in Table 1 showed universality or related the universality of RWNN to the universality of the reader NN. We hope the added experimental results in **A1** using an MLP as the reader NN make the expression “Any universal” more convincing, but we are happy to adjust it if the reviewer finds the result insufficient.
>
> **References**
>
> Kim et al. Learning probabilistic symmetrization for architecture agnostic equivariance (2023)
>
> Gilmer et al. Neural message passing for quantum chemistry (2017)
>
> Xu et al. How powerful are graph neural networks? (2019)
>
> Maron et al. Provably powerful graph networks (2019)
>
> Frasca et al. Understanding and extending subgraph GNNs by rethinking their symmetries (2022)

---

> ### Author Response · Authors · 2024-11-26
> **3/3**
>
> > **Q8.** Lines 347-349: recent architectures such as Errica et al. (2023) show that it is possible to modify the computation over the topology to control oversmoothing and oversquashing, and these two concepts are not as entangled to the original topology and related to each other anymore. Perhaps the authors would like to rephrase their statement in light of this?
>
> **A8.** Thank you for bringing up this reference. AMP (Errica et al. 2023) mitigates over-smoothing and over-squashing by learning to prune unnecessary messages at each step (among its other techniques). This brings it close to attention-based message-passing, which can be understood as modifying the computation (P) over topology. Indeed, our description of over-smoothing and over-squashing in MPNNs in Line 304-315 mainly applies to message-passing using fixed weights, and attention falls out of this framework, which is a shared limitation with related work (Topping et al. 2022; Giraldo et al. 2023; Black et al. 2023; Nguyen et al. 2023). We have revised several parts in Section 3.2 (e.g. Line 300-302) to clarify the scope of discussion to MPNNs using fixed weights. We will add the discussion on AMP in the extended related works section (Appendix A.7) in the next revision.
>
> Formal propagation analysis for attention-based message-passing is currently in infancy (Wu et al., 2023), but we believe it is interesting since, based on our parallelism between MPNNs and RWNNs, it may tell us something about how to adapt random walk probabilities adaptively based on input data. We leave investigating this aspect as future work.
>
> > **Q9.** Figure 3: what are colors and lines? The authors should improve the figure a bit by adding a legend.
>
> **A9.** Red, blue, and black dashed lines corresponds to Baseline 1, Baseline 2, and BuNN (previous best method), respectively. We have revised the figure.
>
> > **Q10.** Minor typos
>
> > I would also suggest to define the meaning of the symbols =d (define precisely what means equal in probability) and ~ (the isomorphism relation). The authors use them a lot in the paper and they should be properly defined.
>
> > Line 53: should it be “Addressing such questions is important”?
>
> > Theorems 2.1, 2.2, 2.3 formalize rather intuitive concepts, and the proofs are mostly about stating the definition of invariance. Maybe the authors could “downgrade” them to Propositions to inform the reader that some results have more relevance than others?
>
> **A10.** Thank you for pointing out the typos. We have revised them directly. Precise definitions for equality in probability and isomorphisms are provided in Appendix A.5 due to space restrictions, and we added a pointer in Line 129.
>
> **References**
>
> Errica et al. Adaptive message passing: A general framework to mitigate oversmoothing, oversquashing, and underreaching (2023)
>
> Topping et al. Understanding over-squashing and bottlenecks on graphs via curvature (2022)
>
> Black et al. Understanding oversquashing in GNNs through the lens of effective resistance (2023)
>
> Giraldo et al. On the trade-off between over-smoothing and over-squashing in deep graph neural networks (2023)
>
> Nguyen et al. Revisiting over-smoothing and over-squashing using Ollivier-Ricci curvature (2023)
>
> Wu et al. Demystifying oversmoothing in attention-based graph neural networks (2023)

---

> > ### Comment · Reviewer_J6pj · 2024-11-28
> > **Thank you for your clarifications**
> >
> > I thank the reviewers for addressing my concerns. I am quite happy with the rebuttal and will keep my score. I will also support the paper in the next phases.

---

> ### Author Response · Authors · 2024-11-28
>
> Thank you very much for your warm support!

---

### Official Review · Reviewer_RBkC · 2024-10-31

**Soundness:** 3
**Presentation:** 2
**Contribution:** 3
**Rating:** 8
**Confidence:** 4

**Summary:**

The paper introduces RWNNs as a unified framework of graph learning approaches that sample random walks from an input graph and then process a suitable representation of those walks with a reader neural network. This general concept covers a range of recent approaches.

The work provides a theoretical analysis of sufficient conditions on the walks and walk records that ensure permutation invariance regardless of the choice of the reader architecture. The paper further proves that RWNNs are probabilistically universal if the random walks are sufficiently likely to cover a graph and that RWNNs inherently avoid the phenomenon of over-smoothing.

An empirical study further studies how RWNNs with transformer-based reader functions perform on various tasks. Here, a training-free approach that passes a text-based walk representation to a Llama model is explored to solve classification tasks in citation networks with in-context learning. The overall performance of this method is reported to be competitive.

**Strengths:**

* The work provides multiple novel theoretical insights on random-walk-based GNNs and a general framework for modeling and studying such approaches further.
* Applying pre-trained language models to text-based representations of random walks to perform in-context learning is a novel idea, and the presented results seem promising.

**Weaknesses:**

While I think the theoretical contributions are significant, I think the experiment section lacks some details that would improve the paper:

* What is the runtime of the compared methods during inference? While the RWNN-Llama model requires no training, I would expect its computational cost during inference to be substantially larger than that of a standard GNN. This by itself is not a problem as LLMs are expected to be costly, and RWNN also allows for more efficient choices for the reader model. However, this tradeoff should be discussed more clearly, and providing some runtime measurements would be very helpful to the reader.

* The cover time of different walk distributions is examined in section 5.1. However, no further comparison of different walks on the predictive tasks of 5.2 or 5.3 is performed. It is therefore not clear whether or not the difference in cover time translates to a significant difference in downstream tasks in practice.

*  The main walk hyperparameters (length and restart probability) are not specified in Sections 5.2 and 5.3.

There are also some grammatical mistakes that would benefit from a spell checker.

**Questions:**

* What is the inference cost for RWNN-Llama in section 5.3? How does it compare to inductive GNN-based approaches?
* How are the results in section 5.2 and 5.3 affected by the choice of random walks? Would uniform random walks with or without backtracking yield significant performance differences on the downstream tasks?
* What walk length and restart probabilities are used in sections 5.2 and 5.3?
* Why is a DeBERTa model that was pretrained on natural language used in the experiments for isomorphism learning? The used walk records do not resemble written language and it is not clear which information learned during language pretaining is useful here. Is there a significant benefit to using this model as opposed to a randomly initialized one?

---

> ### Author Response · Authors · 2024-11-26
> **1/3**
>
> We thank the reviewer for the insightful comments and positive assessment of our work. We address the questions below.
>
> > **Q1.** What is the runtime of the compared methods during inference? While the RWNN-Llama model requires no training, I would expect its computational cost during inference to be substantially larger than that of a standard GNN. This by itself is not a problem as LLMs are expected to be costly, and RWNN also allows for more efficient choices for the reader model. However, this tradeoff should be discussed more clearly, and providing some runtime measurements would be very helpful to the reader. (...) What is the inference cost for RWNN-Llama in section 5.3? How does it compare to inductive GNN-based approaches?
>
> **A1.** We agree that while RWNN-Llama in Section 5.3 can perform inference without training, the inference cost is substantially larger compared to standard GNNs for transductive classification. While random walk sampling and text recording can be done efficiently, the main computational bottleneck comes from the use of Llama 3, a large language model, as the reader model. For example, each test prediction of RWNN-Llama-3-70b takes around 0.410 ± 0.058 seconds on a 4x H100 machine (RWNN-Llama-3-8b is faster, 0.116 ± 0.064 seconds per prediction). While we haven’t measured the precise inference time of standard GNNs in the same environment, we expect it to take much shorter, possibly less than one minute for 48,603 test vertices. We will include the runtime measurement of RevGAT in the next revision. We also remark that if language model embeddings are used and fine-tuned, the training cost of GNN methods can also be substantial, e.g., recent work has reported that prior methods for training a GNN jointly with language model embeddings on arXiv take 25-46 hours on a 6x RTX 3090 machine (Table 3 of Zhu et al. 2024).
>
> > **Q2.** The cover time of different walk distributions is examined in section 5.1. However, no further comparison of different walks on the predictive tasks of 5.2 or 5.3 is performed. It is therefore not clear whether or not the difference in cover time translates to a significant difference in downstream tasks in practice. (...) How are the results in section 5.2 and 5.3 affected by the choice of random walks? Would uniform random walks with or without backtracking yield significant performance differences on the downstream tasks?
>
> **A2 (1/2).** Our results on cover times provide the worst-case upper bounds of walk length required to achieve universality, which is in line with Markov chains literature on the cover times (Feige, 1995). We have experimented with two examples where performing interventions that change cover times leads to substantial differences in task performances (but please also see the discussion provided at the end of **A2 (2/2)**). We show the results below.
>
> First, in arXiv transductive classification, due to the high homophily of the dataset (Platonov et al. 2024), we may assume that the cover time of local neighborhoods ($B_r(v)$ for small $r$) has an important influence on classification performance. Based on this assumption, we set $r = 1$ and checked if the random walk strategy that lowers the cover time of $B_r(v)$ leads to a better performance. To experiment with different cover times, we test two choices, 0.7 and 0.3, for the random restart probability $\alpha$. Since we use neighborhood recording, we measure vertex cover times. Due to the large size of the dataset, we estimate the cover times by fixing a random set of 100 test vertices and running 100 random walks from each of them. The result is given below.
>
> |  | Test accuracy | Cover time $C_V(B_r(v)), r=1$ | Unique vertices observed in a walk |
> |---|---|---|---|
> | $\alpha = 0.3$ | 74.40% | 161.2 | **45.28** |
> | $\alpha = 0.7$ | **74.81%** | **96.84** | 31.13 |
>
> We can see that increasing the restart probability $\alpha$ (0.3 → 0.7) simultaneously results in lower cover time (161.2 → 96.84) and higher test accuracy (74.40% → 74.81%), even though the average number of observed vertices decreases (45.28 → 31.13). This implies that lower local cover time is correlated with better performance, while the number of observed vertices itself is not directly correlated with better performance.
>
> **References**
>
> Zhu et al. Efficient tuning and Inference for Large Language Models on Textual Graphs (2024)
>
> Feige, A tight upper bound on the cover time for random walks on graphs (1995)
>
> Platonov et al. Characterizing graph datasets for node classification: Homophily-heterophily dichotomy and beyond (2024)

---

> ### Author Response · Authors · 2024-11-26
> **2/3**
>
> **A2 (2/2).** Second, in SR16 graph separation, we experiment with whether or not to use the neighborhood recording, which changes the required cover time from vertex cover time $C_V(G)$ (when used) to edge cover time $C_E(G)$ (when not used). The result is given below.
>
> |  | Test accuracy | Cover time |
> |---|---|---|
> | w/ neighborhood recording | **100%** | **48.25 ($C_V(G)$)** |
> | w/o neighborhood recording | 50% | 490.00 ($C_E(G)$) |
>
> We can see that not using neighborhood recording has a drastic effect on the required cover time (vertex cover time 48.25 → edge cover time 490.00), and the test performance deteriorates from perfect to random accuracy (100% → 50%).
>
> The above results are the cases where interventions in cover times lead to differences in task performances. In general, in practice, we think there are several factors that make it not always trivial to equate task performances and cover times. While cover times provide the worst-case upper bounds of walk length required for perfect reconstruction of input graph, in practice, there is a substantial amount of useful information that can be recovered from possibly non-covering walks. Examples include spectral properties (Banjamini et al., 2003), number of edges and mixing time (Ben-Hamou et al., 2019), and triangle counts (Bera & Seshadhri, 2020), among others studied in the field of sublinear algorithms. These results suggest that shorter walks may suffice in practice for RWNNs if the task at hand requires properties that can be estimated without seeing the whole graph, and the reader NN can learn to recover them (e.g., leveraging universality). Understanding this aspect better is important but would require significant additional research, and we leave it for future work. We will add this discussion to the main text in the next revision.
>
> > **Q3.** The main walk hyperparameters (length and restart probability) are not specified in Sections 5.2 and 5.3. (...) What walk length and restart probabilities are used in sections 5.2 and 5.3?
>
> **A3.** For Sections 5.2 and 5.3, the effective lengths of walks are affected by the language model tokenizers since text records exceeding the maximum number of tokens are truncated. The maximum number of tokens is a hyperparameter; for the DeBERTa tokenizer used in Section 5.2, we set it to 512, and for the Llama 3 tokenizer used in Section 5.3, we set it to 4,096. As a result, the effective lengths of random walks for each dataset are measured as follows:
>
> |  | Effective random walk length |
> |---|---|
> | CSL (Section 5.2) | 214.03 ± 2.48 |
> | SR16 (Section 5.2) | 222.00 ± 0.00 |
> | SR25 (Section 5.2) | 129.54 ± 2.26 |
> | arXiv (Section 5.3) | 194.52 ± 10.59 |
>
> The effective length is shorter (129.54 on average) for SR25 graphs compared to CSL and SR16 graphs (214.03 and 222.00 on average) although using the same tokenizer configurations, which is because the higher average degrees of SR25 graphs compared to CSL and SR16 graphs (Figures 4-6) has led to a higher number of text characters invested in neighborhood records (“#” symbol in Figures 4-6).
>
> In addition, for Section 5.3, where we used restarts, we chose restart probability α = 0.7 by hyperparameter search on a randomized 100-sample subset of validation data.
>
> > **Q4.** There are also some grammatical mistakes that would benefit from a spell checker.
>
> **A4.** Thank you for pointing out the errors. We have fixed several grammatical mistakes, and will run grammar-checking across the paper in our next revision.
>
> **References**
>
> Benjamini et al. Waiting for a bat to fly by (in polynomial time) (2003)
>
> Ben-Hamou et al. Estimating graph parameters with random walks (2019)
>
> Bera & Seshadhri, How to count triangles, without seeing the whole graph (2020)
>
> Lu et al. Pretrained transformers as universal computation engines (2021)
>
> Rothermel et al. Don’t sweep your learning rate under the rug: A closer look at cross-modal transfer of pretrained transformers (2021)

---

> ### Author Response · Authors · 2024-11-26
> **3/3**
>
> > **Q5.** Why is a DeBERTa model that was pretrained on natural language used in the experiments for isomorphism learning? The used walk records do not resemble written language and it is not clear which information learned during language pretaining is useful here. Is there a significant benefit to using this model as opposed to a randomly initialized one?
>
> **A5.** Thank you for the suggestion. On CSL and SR16 isomorphism learning datasets (Section 5.1), we ran the same experiment but with random parameters initialization instead of language pre-training. We indeed find that language pre-training has significant benefits over random initialization for isomorphism learning tasks. We have added the results in Appendix A.4.1 and Figure 14.
>
> We conjecture that certain representations or computational circuits acquired in language pre-training are useful for processing text records of walks to recognize graphs. For example, we could imagine that some circuits specialized in recognizing certain syntactic structures in natural language could be repurposed during fine-tuning to detect skip links of CSL graphs from text records of walks.
>
> Our results are also consistent with prior observations on cross-model transfer of pre-trained language models in different domains, such as image recognition (Lu et al. 2021; Rothermel et al. 2021).
>
> **References**
>
> Lu et al. Pretrained transformers as universal computation engines (2021)
>
> Rothermel et al. Don’t sweep your learning rate under the rug: A closer look at cross-modal transfer of pretrained transformers (2021)

---

> ### Author Response · Authors · 2024-12-02
>
> Dear reviewer RBkC, thank you for taking time to review our paper. Since there are now around 29 hours left for reviewers to post a message to the authors, we would like to check if our responses have addressed your concerns and would like to provide additional clarification if there are any follow-up questions.

---

> > ### Comment · Reviewer_RBkC · 2024-12-02
> >
> > I thank the reviewers for the thorough response and paper adjustments. My main concerns have been adequately addressed and I have increased my rating to 8.

---

> > > ### Author Response · Authors · 2024-12-03
> > >
> > > Thank you very much for taking our response and revisions into consideration! We will make sure to incorporate all discussions and new results to the next version of the paper.

---

### Official Review · Reviewer_8Hx8 · 2024-11-04

**Soundness:** 2
**Presentation:** 2
**Contribution:** 2
**Rating:** 6
**Confidence:** 4

**Summary:**

The paper revisits the usage of random walks for learning on graphs. In particular, they consider the setting where the model consists of three components: a random walk algorithm, a recording function producing a machine-readable record of the random walk, and a reader neural network that processes the record and produces an output. The authors show that probabilistic invariance can be achieved by requiring invariance conditions on the random walk algorithm and the recording function. They also show connections with over-smoothing and over-squashing.

**Strengths:**

The problem is interesting, and using llms to process outputs from the recording function makes the results more practical and significant.

**Weaknesses:**

The readability of the paper can be improved. The paper is quite dense and has many theoretical results, while the experiments appear only at the end. Perhaps move the corresponding experiments closer to the relative sections would improve readability.

Generally speaking, I find it confusing that the authors emphasize the invariance of the random walk algorithm, while instead this is a probabilistic invariance, that is, an invariance of the distribution. While the authors discuss that they accept the probabilistic notion of invariance (eq. 2), they sometimes only refer to it as invariance (eq. 3). I think this point should be better clarified throughout the paper. Moreover, since it is well known that one can average over all random walks to obtain the invariance of permutations (Srinivasan and Ribeiro, 2020), I encourage the authors to connect the probabilistic invariance to the notion of invariance more commonly used in graph learning.


Srinivasan and Ribeiro, 2020. On the Equivalence between Positional Node Embeddings and Structural Graph Representations.

**Questions:**

1. Can you include results on real-world graph-level datasets? I think only evaluating graph-level synthetic datasets is a bit restrictive.

2. What is the impact of voting in the node-level task? I think the fact that you need to average across multiple voters reflects the difference between invariance and probabilistic invariance.

---

> ### Author Response · Authors · 2024-11-26
> **1/2**
>
> We thank the reviewer for the insightful and constructive suggestions on improvements. We address the questions below.
>
> > **Q1.** The readability of the paper can be improved. The paper is quite dense and has many theoretical results, while the experiments appear only at the end. Perhaps move the corresponding experiments closer to the relative sections would improve readability.
>
> **A1.** Thank you for the constructive suggestion. We think the submitted version's Section 3.1 was a bit verbose and had room for improvement. We have improved the readability by making the section shorter, and invested the obtained space to improve the readability of Section 3.2. We have also improved the readability of the experiments section by adjusting the writing as well as spacings of tables and figures.
>
> > **Q2.** Can you include results on real-world graph-level datasets? I think only evaluating graph-level synthetic datasets is a bit restrictive.
>
> **A2.** Thank you for raising this point. We ran a preliminary experiment on the Peptides-func dataset used in the CRaWl paper and obtained promising initial results. We selected this dataset because it involves a real-world graph-level classification task, and its moderate size made it feasible to complete the experiment within the rebuttal period.
>
> Our model is a pre-trained DeBERTa-base (identical to Section 5.1) fine-tuned on text records of non-backtracking MDLR random walks with anonymization and neighborhood recording. The recording function is designed to properly incorporate the vertex and edge attributes provided in the dataset, including the atom and bond types.
>
> In the table below, we report the test average precision (AP) at best validation accuracy, with 40 random predictions ensembled at test time. We report the mean and standard deviation of the test performances for five repeated tests. The baseline performances, including CRaWl, are from Toenshoff et al. (2023).
>
> |  | Peptides-func (test AP) |
> |---|---|
> | GCN | 0.5930 |
> | GINE | 0.5498 |
> | GatedGCN | 0.6069 |
> | Transformer | 0.6326 |
> | SAN | 0.6439 |
> | CRaWl | 0.7074 |
> | RWNN-DeBERTa (Ours) | **0.7124 ± 0.0016** |
>
> We see that RWNN-DeBERTa, which has been successful in graph isomorphism learning (Section 5.2), also shows a promising result in real-world protein graph classification despite its simple approach of recording random walks in text and processing them with a fine-tuned language model. We added this result in Appendix A.4.2, along with an example of text record in Figure 15.
>
> > **Q3.** Generally speaking, I find it confusing that the authors emphasize the invariance of the random walk algorithm, while instead, this is a probabilistic invariance, that is, an invariance of the distribution. While the authors discuss that they accept the probabilistic notion of invariance (eq. 2), they sometimes only refer to it as invariance (eq. 3). I think this point should be better clarified throughout the paper. Moreover, since it is well known that one can average over all random walks to obtain the invariance of permutations (Srinivasan and Ribeiro, 2020), I encourage the authors to connect the probabilistic invariance to the notion of invariance more commonly used in graph learning. (...) I think the fact that you need to average across multiple voters reflects the difference between invariance and probabilistic invariance.
>
> **A3 (1/2).** We apologize for the confusion. We indeed meant probabilistic invariance when describing random walks or the entire RWNN outputs, but sometimes wrote invariance due to space restrictions. We have revised the main text overall to clearly distinguish between probabilistic invariance and invariance. We note that invariances of conductance (Equation 5) and recording function (Equation 7) follow the common notion of invariance since they are deterministic functions.
>
> **References**
>
> Toenshoff et al. Walking out of the Weisfeiler Leman hierarchy: Graph learning beyond message passing (2023)
>
> Srinivasan & Ribeiro, On the equivalence between positional node embeddings and structural graph representations (2020)

---

> ### Author Response · Authors · 2024-11-26
> **2/2**
>
> **A3 (2/2).** And thank you for raising the question on the connection between probabilistic invariance and invariance. As the reviewer pointed out, our model is a probabilistic invariant predictor $p_\theta(\mathbf{y}|\mathbf{x})$. To make a prediction from this model, we obtain a set of samples $\mathbf{y}\_i \sim p_\theta(\mathbf{y}|\mathbf{x})$ and ensemble them by averaging* (Sections 5.1 and 5.2). This procedure can be understood as an unbiased Monte Carlo (MC) estimation of the underlying "mean predictor" $F_\theta(\mathbf{x}) \coloneqq \mathbb{E}\_{\mathbf{y} \sim p_θ(\mathbf{y}|\mathbf{x})}[\mathbf{y}]$. Importantly, since $p_\theta$ is invariant in probability, the mean predictor $F_\theta$ is guaranteed to be an invariant function (in the standard sense). This clarifies the connection between probabilistic invariance and invariance.
>
> In the terminology of Srinivasan & Ribeiro (2020), if we have a randomized predictor of positional node embeddings, we can estimate structural graph representations by sample averaging (Theorem 2 of Srinivasan & Ribeiro, 2020). What we are doing with RWNNs in our work can be understood similarly, estimating the underlying invariant predictor ($F_\theta$) using samples from a probabilistic invariant predictor ($p_\theta$). We have revised Line 125-129 and 140-141 to make this interpretation clear.
>
> Probabilistic invariant predictors are recently actively being investigated since they can be relatively easily designed to have high expressive powers compared to deterministic invariant counterparts (Yarotsky et al., 2019; Murphy et al., 2019a, 2019b; Abboud et al., 2021; Cotta et al., 2023; Kim et al., 2023), and their sampling-based predictions still retain the fundamental generalization benefits of invariant predictors (e.g., Proposition 8 of Lyle et al., 2020). Our work has similar underlying motivations. We are happy to elaborate on these aspects if the reviewer is interested.
>
> \* note: in Section 5.3, we ensemble samples $\mathbf{y}\_i \sim p_\theta(\mathbf{y}|\mathbf{x})$ by voting, not averaging, since we cannot directly obtain continuous logits from Llama 3 generated text. Similarly to MC estimation, this can be understood as estimating the invariant limiting classifier $F_\theta(\mathbf{x})$ (Cotta et al., 2023).
>
> **Q4.** What is the impact of voting in the node-level task?
>
> **A4.** As discussed in **A3**, averaging or voting can be viewed as an estimation process of the underlying invariant predictor. From another perspective, we can also interpret them as a type of test-time scaling with repeated sampling, which has been shown beneficial in modern language models (Brown et al. 2024) and randomized algorithms (classically called amplification; Cotta et al. 2023). To see if similar perspectives can be held for RWNNs, we have measured model performance with an increased number of samples for averaging or voting. The results are given below:
>
> | RWNN-DeBERTa test sample size (averaging) | Peptides-func test AP |
> |---|---|
> | 1 | 0.5300 ± 0.0092 |
> | 10 | 0.6937 ± 0.0039 |
> | 20 | 0.7064 ± 0.0027 |
> | 40 | 0.7124 ± 0.0016 |
> | 60 | 0.7145 ± 0.0013 |
> | 80 | 0.7155 ± 0.0014 |
> | 100 | 0.7170 ± 0.0009 |
> | 160 | 0.7176 ± 0.0013 |
> | 320 | **0.7177 ± N/A*** |
>
> | RWNN-Llama-3-70b test sample size (voting) | arXiv classification test accuracy |
> |---|---|
> | 1 | 74.45 ± 0.049% |
> | 3 | 74.75 ± 0.035% |
> | 5 | 74.83 ± 0.028% |
> | 7 | 74.86% ± N/A* |
> | 10 | **74.87% ± N/A*** |
>
> \* we were unable to run repeated experiments due to sampling costs.
>
> Indeed, we see that increasing sample size consistently leads to better performances as well as reduced variances. The point of plateau seems to differ per application and model architecture, and we leave obtaining a better understanding of the scaling behavior for future work.
>
> **References**
>
> Cotta et al. Probabilistic invariant learning with randomized linear classifiers (2023)
>
> Yarotsky et al. Universal approximations of invariant maps by neural networks (2019)
>
> Murphy et al. Relational pooling for graph representation (2019)
>
> Murphy et al. Janossy pooling: Learning deep permutation-invariant functions for variable-size inputs (2019)
>
> Abboud et al. The surprising power of graph neural networks with random node identification (2021)
>
> Kim et al. Learning probabilistic symmetrization for architecture agnostic equivariance (2023)
>
> Lyle et al. On the benefits of invariance in neural networks (2020)
>
> Brown et al. Large language monkeys: Scaling inference compute with repeated sampling (2024)

---

> ### Author Response · Authors · 2024-12-02
>
> Dear reviewer 8Hx8, thank you for taking time to review our paper. Since there are now around 29 hours left for reviewers to post a message to the authors, we would like to check if our responses have addressed your concerns and would like to provide additional clarification if there are any follow-up questions.

---

> > ### Comment · Reviewer_8Hx8 · 2024-12-02
> >
> > Thank you for the rebuttal. I believe the paper makes an interesting contribution, though **there is room for improvement in the presentation**. Additionally, **the inclusion of only one graph-level real-world dataset feels somewhat limited**. However, I understand the constraints of the rebuttal period, and therefore I have increased my score to 6.

---

> > > ### Author Response · Authors · 2024-12-03
> > >
> > > Thank you very much for your willingness to reassess our work and kind understanding of the constraints of the rebuttal period! We will do our best to improve the readability of the paper in our next revision.

---

### Official Review · Reviewer_R47c · 2024-11-04

**Soundness:** 3
**Presentation:** 3
**Contribution:** 3
**Rating:** 8
**Confidence:** 4

**Summary:**

The paper considers random-walk based techniques for graph learning. It introduces a framework splitting the random walk function, recording function (drawing features from the graph), and reader NN (actually processing those features). They then show that for invariance one needs to have it on the first two levels but not the third. In terms of expressiveness, everything that is sufficiently powerful suffices, which is again made formal. Also on the theoretical side, over-smoothing does not happen for RW-based methods.

On the practical side, the paper combines efficient non-backtracking walks while always recording the neighborhood, outputting the result as a string which is then fed into an LLM. This architecture works very well for oversmoothing, over-squashing, isomorphism learning and node classification.

**Strengths:**

The whole part about random walk strategies and cover times is novel and interesting. The suggested method of compiling the walk information (including anonymization and neighborhood information) into a string to be fed into an LLM is also new, even though building upon exisitng work (this could be more explicit when describing the model).

The theoretical results seem sound too.

**Weaknesses:**

A lot of the theory has been implicitly used e.g. in CraWl, e.g. by using 1D CNNs as reader NN. Also the anonymization and neighborhood recording is already done there (with the difference that it outputs a matrix instead of a string). Also there is quite a bit of literature on anonymization and e.g. that anonymization alone is enough to recover the graph which I would have expected to be discussed in that context.

The expressive power section is quite lengthy while not saying much - by now we should all know that there are universal function approximators so if no information is lost during preprocessing (message passing or random walks), any such model is universal. While writing down the exact details on what it means to be universal/expressive, I consider most of section 3.1 entirely unsurprising and somewhat trivial.

There should definitely be more experiments. Currently the experiments cover two types of toy datasets and one real-world one. Graph classification that the main competitor Crawl exclusively covers is left out in the real-world setting which is odd and I believe should be added. Also having more than one node-classification dataset would be nice.

Code is unavailable. This makes reproducing the results a lot harder (even though the description of walks and feature extraction is detailed, the exact use of the LLM and the finetuning procedures are not well-enough described to confidently repeat the experiments).

Details:
- Grammar/Structure in 87ff is off. especially: "given that..." and "The result also..."
- Table 3/4: maybe just put the link to the reference on the method name to avoid the cluttered look of the table.

**Questions:**

Is there a simple intuitive reason why RWNN also outputs something based on the stationary vector in Thm 3.5?

would it be possible to include Crawl and WalkLM in Table2? As those seem to be the only two methods to compare to..

I would be very interested in graph learning performance, especially since Crawl was surprisingly good there.

Would it be possible to include SimTeG methods in Table 4? They also combine LLM and GNN, just in the opposite order. I also believe that using SimTeG features would further improve your results and make them actually SOTA. Currently the table is misleading.

Is there anything to say about cases where walks do not cover the whole graph?

---

> ### Author Response · Authors · 2024-11-26
> **1/3**
>
> We thank the reviewer for the insightful comments and positive assessment of our work. We address the questions below.
>
> > **Q1.**  A lot of the theory has been implicitly used, e.g., in CraWl e.g., by using 1D CNNs as reader NN. Also, the anonymization and neighborhood recording is already done there (with the difference that it outputs a matrix instead of a string).
>
> **A1.** As the reviewer mentioned, our approach is related to CRaWl in two key aspects:
>
> - CRaWl’s binary identity and connectivity matrix encodings can be viewed as analogous to anonymization and neighborhood recording in our work, respectively.
> - CRaWl uses 1D CNNs as the reader NN.
>
> A key technical difference lies in the first part. While CRaWl’s identity and connectivity encodings contain analogous information to anonymization and neighborhood recording, they are defined within a fixed window size (denoted $s$ in Toenshoff et al., 2023), which puts a locality constraint on the recorded information. Precisely, the encodings at step t can encode the information of the walk from step $t - s$ to $t$ (precisely, its induced subgraph), referred to as a walklet. The window size $s$ is a hyperparameter that controls the expressive power of CRaWl, and this dependency makes CRaWl non-universal (Section 3.2, Toenshoff et al. 2023). Our choice of anonymization and neighborhood recording are not under such a local constraint, and they encode the full information of a given walk globally. This property underlies our universality results in Section 3.1, which also naturally motivates our choice of universal reader NNs (e.g., a transformer LM), which were not explicitly considered in CRaWl.
>
> We have added this discussion in Appendix A.6 and left a pointer in Section 4 to make the relationship between our work and CRaWl more explicit.
>
> > **Q2.** There is quite a bit of literature on anonymization and, e.g., that anonymization alone is enough to recover the graph, which I would have expected to be discussed in that context.
>
> **A2.** Indeed, the first anonymization paper by Micali & Zhu (2015) has stated an important result that a sufficiently long anonymized walk starting from a vertex $v$ encodes sufficient information to reconstruct the local subgraph $B_r(v)$ up to isomorphism. While the authors’ focus was a probabilistic graph reconstruction algorithm that uses a set of independent anonymized walks and accesses the oracle set of all possible anonymized walks, we adopt the idea in a neural processing context to acquire universality in probability. We also remark that the invariance property of anonymization has rarely been noticed formally in the literature, which is our key motivation for using it, on top of being able to recover the whole graph.
>
> The influence of Micali & Zhu (2015) on our work is important. We have added above discussion in Appendix A.6 to highlight it and left a pointer in Line 201. We are happy to discuss additional related work if the reviewer suggests.
>
> > **Q3.** The expressive power section is quite lengthy while not saying much - by now, we should all know that there are universal function approximators so if no information is lost during preprocessing (message passing or random walks), any such model is universal. While writing down the exact details on what it means to be universal/expressive, I consider most of section 3.1 entirely unsurprising and somewhat trivial.
>
> **A3.** We agree that having built up the results on cover time in Section 2, the results in Section 3.1 can be derived straightforwardly by applying the standard argument (e.g., Zaheer et al., 2017) that lossless pre-processing postfixed by a universal processor leads to universality. (Our proofs for them are indeed quite compact compared to cover time results.) While we think there is a necessity to show these results in the main text, we agree that an experienced reader can find them expected. We have revised Section 3.1 to be more compact overall and invested the obtained space to make Section 3.2 more readable.
>
> **References**
>
> Toenshoff et al. Walking out of the Weisfeiler Leman hierarchy: Graph learning beyond message passing (2023)
>
> Micali & Zhu, Reconstructing Markov processes from independent and anonymous experiments (2015)
>
> Ivanov & Burnaev, Anonymous walk embeddings (2018)
>
> Zaheer et al. Deep sets (2017)

---

> ### Author Response · Authors · 2024-11-26
> **2/3**
>
> > **Q4.** There should definitely be more experiments. Currently the experiments cover two types of toy datasets and one real-world one. Graph classification that the main competitor Crawl exclusively covers is left out in the real-world setting which is odd and I believe should be added. (...) I would be very interested in graph learning performance, especially since Crawl was surprisingly good there.
>
> **A4.** Thank you for raising this point. In response to the reviewer’s request, we conducted a preliminary experiment on the Peptides-func dataset used in CRaWl and obtained promising initial results. We selected this dataset because it involves a real-world graph-level classification task, and its moderate size made it feasible to complete the experiment within the rebuttal period.
>
> Our model is a pre-trained DeBERTa-base (identical to Section 5.1) fine-tuned on text records of non-backtracking MDLR random walks with anonymization and neighborhood recording. The recording function is designed to properly incorporate the vertex and edge attributes provided in the dataset, including the atom and bond types.
>
> In the table below, we report the test average precision (AP) at best validation accuracy, with 40 random predictions ensembled at test time. We report the mean and standard deviation of the test performances for five repeated tests. The baseline performances, including CRaWl, are from Toenshoff et al. (2023).
>
> |  | Peptides-func (test AP) |
> |---|---|
> | GCN | 0.5930 |
> | GINE | 0.5498 |
> | GatedGCN | 0.6069 |
> | Transformer | 0.6326 |
> | SAN | 0.6439 |
> | CRaWl | 0.7074 |
> | RWNN-DeBERTa (Ours) | **0.7124 ± 0.0016** |
>
> We see that RWNN-DeBERTa, which has been successful in graph isomorphism learning (Section 5.2), also shows a promising result in real-world protein graph classification despite its simple approach of recording random walks in text and processing them with a fine-tuned language model. We added this result in Appendix A.4.2, along with an example of text record in Figure 15.
>
> > **Q5.** Having more than one node classification dataset would be nice.
>
> **A5.** We would like to kindly point out that Appendix A.4.4 (Table 7) includes preliminary results on other vertex-level transductive classification datasets (Cora, Citeseer, and Amazon Ratings), where we mainly compare against existing language-based graph learning methods and training-free GNNs (Sato, 2024). We have obtained reasonably good performances, especially for the heterophilic Amazon Ratings dataset when the model is fine-tuned. The result supports the idea that RWNNs avoid over-smoothing, as avoiding it is known to be important for handling heterophily (Yan et al., 2022; Platonov et al., 2023). We will revise the main text so that this result is more visible.
>
> > **Q6.** The code is unavailable. This makes reproducing the results a lot harder (even though the description of walks and feature extraction is detailed, the exact use of the LLM and the finetuning procedures are not well enough described to confidently repeat the experiments).
>
> **A6.** We have made the code for experiments in Sections 5.1-5.3 available in the supplementary material, as well as the following anonymized link: https://anonymous.4open.science/r/rwnn-iclr2025-52F3. We plan to open-source the full code of our experiments, checkpoints for DeBERTa fine-tuning experiments, and text records of sampled walks for arXiv experiments upon acceptance.
>
> > **Q7.** Grammar/Structure in 87ff is off. especially: "given that..." and "The result also..."; Table 3/4: maybe just put the link to the reference on the method name to avoid the cluttered look of the table.
>
> **A7.** Thank you for pointing out the typos. We have fixed them.
>
> > **Q8.** Is there a simple, intuitive reason why RWNN also outputs something based on the stationary vector in Thm 3.5?
>
> **A8.** Intuitively, the RWNN output in Theorem 3.5 encodes the stationary vector because it appears in the long-term frequency of a walk visiting each vertex. Let us consider a random walk running for a very long time $T$. At a given step, the probability of the walk being at some vertex $v$ is the stationary probability $\boldsymbol{\pi}_v$ from the fundamental theorem of Markov chains. This means, in the long run, the number of visits to some vertex $v$ converges to $T\boldsymbol{\pi}_v$. Since the RWNN in Theorem 3.5 uses average pooling, its output converges to a sum of vertex attributes $\mathbf{x}_v$ weighted by: the number of visits $T\boldsymbol{\pi}_v$ divided by walk length $T$.
>
> **References**
>
> Sato, Training-free graph neural networks and the power of labels as features (2024)
>
> Platonov et al. A critical look at the evaluation of GNNs under heterophily: Are we really making progress? (2023)
>
> Yan et al. Two sides of the same coin: Heterophily and oversmoothing in graph convolutional neural networks (2022)

---

> ### Author Response · Authors · 2024-11-26
> **3/3**
>
> > **Q9.** Would it be possible to include Crawl and WalkLM in Table 2? As those seem to be the only two methods to compare to…
>
> **A9.** Thank you for the suggestion. We additionally tested CRaWl and WalkLM for Table 2. For CRaWl, we use the official implementation and only remove batch normalization since it led to severe overfitting for tasks in Table 2. Since CRaWl uses vertex-level average pooling, while we have used walk-level pooling for RWNNs in Table 2, we also tested a variant of CRaWl that uses walk-level pooling and denote it by CRaWl*. For WalkLM, while the work entails two ideas (representation of walks and fine-tuning by masked auto-encoding), we focus on testing its representation of walks since the fine-tuning method is orthogonal to our approach. We do this by modifying our RWNN-transformer to use uniform walks and only record vertex attributes, and denote the resulting model by WalkLM-transformer. The results are shown below. All models have 1 layer and 128 hidden dimensions.
>
> |  | Clique (over-smoothing) | Barbell (over-squashing) |
> |---|---|---|
> | Baseline 1 | 30.94 ± 0.42 | 30.97 ± 0.42 |
> | Baseline 2 | 0.99 ± 0.08 | 1.00 ± 0.07 |
> | MLP | 1.10 ± 0.08 | 1.08 ± 0.07 |
> | GCN | 29.65 ± 0.34 | 1.05 ± 0.08 |
> | GAT | 20.97 ± 0.40 | 1.07 ± 0.09 |
> | BuNN | 0.03 ± 0.01 | 0.01 ± 0.07 |
> | RWNN-transformer (Ours) | **0.016 ± 0.006** | **0.005 ± 0.002** |
> | CRaWl | 0.079 ± 0.049 | 0.177 ± 0.096 |
> | CRaWl* | 0.099 ± 0.027 | 0.096 ± 0.057 |
> | WalkLM-transformer | 0.018 ± 0.005 | 0.037 ± 0.015 |
>
> We can see that RWNN-transformer shows the best result, and WalkLM-transformer is better than CRaWl as well as CRaWl*. We think the limited window size of identity and connectivity encodings and the limited kernel size of 1D CNN used in CRaWl (as discussed in **A1**) make reliable information propagation challenging, which is important in this task. We will add this result to the main text in the next revision.
>
> > **Q10.** Would it be possible to include SimTeG methods in Table 4? They also combine LLM and GNN, just in the opposite order. I also believe that using SimTeG features would further improve your results and make them actually SOTA. Currently the table is misleading.
>
> **A10.** We have attempted combining SimTeG features with our model but could not complete it in time since it involves non-trivial engineering work of integrating text and embedding vector representations of random walks while maintaining high throughput. We have revised Line 520-521 to make it clear that, while we obtain promising results, we do not currently claim the state-of-the-art on arXiv. We believe it is possible to improve our results by fine-tuning Llama-3-70b or by integrating SimTeG features as suggested by the reviewer and leave these engineering investigations for future work.
>
> > **Q11.** Is there anything to say about cases where walks do not cover the whole graph?
>
> **A11.** We thank the reviewer for raising such an interesting question. Running a possibly non-covering random walk to recover certain information about the whole graph with a high probability is a well-studied topic, especially in the field of sublinear algorithms, and we believe these results have implications for RWNNs. Examples of recoverable information include spectral properties (Banjamini et al., 2003), number of edges and mixing time (Ben-Hamou et al., 2019), and triangle counts (Bera & Seshadhri, 2020), among others. In the context of RWNNs, while our results on cover times provide the worst-case guarantees on the required lengths of walks for RWNNs to work, these results suggest that shorter lengths may suffice in practice if a task at hand requires properties that can be estimated without seeing the whole graph and the reader neural network can learn to recover them (e.g., leveraging universality). This is an important direction for future work. We will add this discussion to the main text in the next revision.
>
> **References**
>
> Benjamini et al. Waiting for a bat to fly by (in polynomial time) (2003)
>
> Ben-Hamou et al. Estimating graph parameters with random walks (2019)
>
> Bera & Seshadhri, How to count triangles, without seeing the whole graph (2020)

---

> > ### Comment · Reviewer_R47c · 2024-11-29
> >
> > Thanks a lot for the detailed responses, not only to the questions, but also carefully addressing all weaknesses.
> >
> > The additional experiments look really nice! For the given one (peptides-func) I would suggest the updated baseline results for the message-passing model from another Toenshoff et al paper ("Where did the gap go") as these are much more representative of the classes than those reported in LRGB (and also in crawl which cites the LRGB values).
> > I also highly appreciate the work that went into getting Crawl and WalkLM to run on Clique and Barbell, I believe this allows readers to fully set the present work into context.
> >
> > Also thanks for explaining the difficulty with additional SimTeG features! I agree that this might require a significant amount of engineering and thus might be out of scope for the current paper.
> >
> > Overall, I think it is really a nice paper and I have raised my score.

---

> ### Author Response · Authors · 2024-11-29
>
> Thank you very much for your willingness to interact with our rebuttal and updating the evaluation! We will make sure to incorporate all the updated results and discussions in the next version. And thank you for reminding us the important empirical work of Toenshoff et al. (2023b). We will update the MPNNs baselines for Peptides-func using the scores in the paper.
>
> Toenshoff et al. Where did the gap go? Reassessing the long-range graph benchmark (2023)

---

### Comment · Area_Chair_npDL · 2024-11-28

I would like to encourage the reviewers to engage with the author's replies if they have not already done so. At the very least, please
acknowledge that you have read the rebuttal.

---

### Meta-Review · Area_Chair_npDL · 2024-12-19

**Metareview:**

The paper proposes a generic random walk-based framework for graph learning that combines random walks, a recording function, and a reader neural network (e.g., a pre-trained language model). The authors derive theoretical results on probabilistic invariance and universality, and show how their method can avoid oversmoothing and oversquashing under specific configurations. The discussion on cover times is interesting. The empirical results look promising, but the breath of the evaluation is limited (primarily synthetic and a few real-world datasets).

**Additional Comments On Reviewer Discussion:**

Some of the concerns raised by the reviewers included: lack of real-world datasets, ambiguity in distinguishing invariance concepts, and missing runtime and scalability analyses. In response the authors added (promising) results on the Peptides-func dataset), and revised the paper for clarity and compactness. The reviewers acknowledged the improvements (included a raised score). One reviewer gave high praise "This is one, if not the most, beautiful papers I have reviewed.".

---

### Decision · Program_Chairs · 2025-01-22

Accept (Spotlight)